# Lithium diffusion-controlled Li-Al alloy negative electrode for all-solid-state battery

Yuju Jeon [1], Dong Ju Lee [1], Hongkui Zheng[2], Sesha Sai Behara [3], Jung-Pil Lee[4], Junlin Wu[5], Feng Li [1], Wei Tang[1], Lanshuang Zhang [5], Yu-Ting Chen[5], Dapeng Xu[1], Jiyoung Kim[4], Min-Sang Song[4], Anton Van der Ven [3] ✉, Kai He [2] ✉ & Zheng Chen [1,5] ✉

Metal alloy negative electrodes are promising candidates for lithium all-solid-state batteries due to their high specific capacity and low cost. However, chemo-mechanical degradation and atomic transport limitations in the solid state remain unresolved challenges. Herein, we demonstrate a lithium-aluminum alloy negative electrode design ($Li_xAl_1$, $x$ = molar ratio of lithium to aluminum) based on a comprehensive understanding of the underlying diffusion mechanisms within the lithium-poor α ($0 \leq x \leq 0.05$) and lithium-rich β phases ($0.95 \leq x \leq 1$). The lithium-aluminum alloy negative electrodes with a higher lithium to aluminum ratio facilitate lithium migration through the β-LiAl phases, which serve as highly lithium-conductive channels with a lithium diffusion coefficient that is ten orders of magnitude higher than that of the α phase. In addition, a bulk dense negative electrode and an intimate negative electrode-electrolyte interface is demonstrated in the cross-sections of the lithium-aluminum alloy negative electrodes. Consequently, a high-rate capability of 7 mA cm$^{-2}$ is attained in $LiNi_{0.8}Co_{0.1}Mn_{0.1}O_2$-based full-cell operation. The optimal cell configuration of $Li_{0.5}Al_1 || LiNi_{0.8}Co_{0.1}Mn_{0.1}O_2$ shows stable lithium reversibility during 2000 cycles with a capacity retention of 83% at 4 mA cm$^{-2}$ with a $LiNi_{0.8}Co_{0.1}Mn_{0.1}O_2$ loading of 5 mAh cm$^{-2}$.

All-solid-state batteries (ASSBs) with inherent safety and high energy density are promising for electric vehicles and other energy storage applications[1,2]. Nevertheless, innovations in high-capacity ASSB negative electrode design are crucial to achieve higher energy density (> 900 Wh L$^{-1}$) and specific energy (> 450 Wh kg$^{-1}$), alongside advancements in the positive electrode and solid-electrolyte (SE) layers[2]. Lithium (Li) metal has been considered as an ideal negative electrode due to its high specific capacity (3860 mAh g$^{-1}$) and low reduction potential (−3.04 V vs. standard hydrogen electrode). However, new anode designs are needed to prevent chemo-mechanical degradation and to eliminate the susceptibility to battery short-circuiting, which plague lithium metal anodes[3,4].

To improve the reversibility of Li negative electrodes, a variety of host materials have been adopted that enable stable Li storage and fast diffusion while mitigating Li dendrite growth. In contrast to intercalation (graphite and $Li_4Ti_5O_{12}$)[5–8] and conversion (SnS, $LiAlH_6$ and AgF)[9–12], alloying reactions (In, Si, Al, Sn and Mg)[13–17] provide higher Li storage capacity and faster Li transport, which are crucial for increasing the energy density of ASSBs[4]. In particular, the high Li diffusion coefficients of some alloy enables negative electrodes with a

[1]Aiiso Yufeng Li Family Department of Chemical and Nano Engineering, University of California, San Diego, La Jolla, CA, USA. [2]Department of Materials Science and Engineering, University of California, Irvine, Irvine, CA, USA. [3]Materials Department, University of California, Santa Barbara, Santa Barbara, CA, USA. [4]LG Energy Solution, Ltd., LG Science Park, Seoul, Republic of South Korea. [5]Program of Materials Science and Engineering, University of California, San Diego, La Jolla, CA, USA. ✉e-mail: avdv@ucsb.edu; kai.he@uci.edu; zhc199@ucsd.edu

high active material ratio without requiring SE to assist Li transport. For example, a recent study of micron-silicon (μSi) demonstrated a high specific capacity (2890 mAh g⁻¹) with a high Li diffusivity in the lithiated state ($\sim 10^{-9}$ cm² s⁻¹) in the absence of SEs in the negative electrode, leading to high-capacity performance at room temperature[14]. This prelithiation strategy was applied to alloy negative electrodes to facilitate uniform Li diffusion, mitigate chemo-mechanical degradation and suppress Li dendrite formation, using Li-Si and Li-Ag alloys, respectively[18,19]. To further increase specific energy (Wh kg⁻¹), an anode-less design was developed with a thin Mg film as an effective seeding layer for reversible lithium plating and stripping, supported by a $Ti_3C_2T_x$ MXene mechanical buffer layer underneath[20]. For practical applications, however, recent studies have focused on improving chemo-mechanical stability within large-scale pouch cells operating under low stack pressure to overcome the challenges that accompany the large volume changes in alloy negative electrodes during Li (de)alloying reactions[21,22].

Among prospective alloying elements, aluminum (Al) exhibits favorable electrochemical properties and is relatively easy to manufacture (Table S1). At room temperature, an Al negative electrode is able to store Li up to 993 mAh g⁻¹ through the formation of stoichiometric LiAl by means of a two-phase reaction from Li-poor α-Al to Li-rich β-LiAl[23]. The resulting β-LiAl compound exhibits a high Li diffusion coefficient of $\sim 10^{-7}$ cm² s⁻¹ and its formation is accompanied by a relatively small volume change of 96% (vs. 280% for Si)[14,24,25]. The high ductilty of Al (Vickers hardness (HV) is 35 for Al and 1130 for Si) is expected to improve interfacial contact with the SE layer and increase the density of the bulk electrode[26,27]. Al is the third most abundant element in the earth's crust (8.1%)[28] with its low cost ($2.45 kg⁻¹ as of June 2025) making it an attractive negative electrode material for widespread use. Recent studies of Al in the context of ASSB systems have focused on the effect of different Li to Al starting ratios[29–31], on the impact of bulk and surface engineering of Al electrodes[15,32,33], on Li diffusion in Al-based electrodes[34,35], on Li metal interface engineering utilizing $Li_9Al_4$[36], and on the effect of porosity on Li-Al alloy electrode stability[37]. Nevertheless, the relationship between phase fraction and microstructure development in the Al electrode and their effect on electrochemical properties has not been thoroughly elucidated.

Herein, we present a comprehensive study of the correlation between β-LiAl phase distribution and Li kinetics in Li-Al negative electrodes by controlling the Li to Al ratio via in situ prelithiation of Al during cell assembly. We found that the reversibility of the Li-Al alloy negative electrode improved significantly at Li to Al ratios above 0.5 as a result of fine microstructures with increased phase fractions of β-LiAl. Density functional theory calculations predict a ten orders of magnitude difference in the Li diffusion coefficients of the Li-poor α and Li-rich β phases. The exceptionally high Li diffusion coefficient in β-LiAl provides fast diffusion pathways to unreacted Al (Li-poor α phase) in the microstructures of $Li_{x \geq 0.5}Al_1$ alloy negative electrodes, thereby enabling significant improvements in rate capability compared to the pure Al negative electrode (7 mA cm⁻² for Li-Al alloy vs. 4 mA cm⁻² for pure Al). The improved charge/discharge reversibility of a $Li_{0.5}Al_1$ alloy negative electrode extended the cycle life to 2000 cycles with capacity retention of 83% in $LiNi_{0.8}Co_{0.1}Mn_{0.1}O_2$ (NCM811)-based solid-state cells at high areal capacity loading of 5 mAh cm⁻². Cross-sectional characterization revealed the changes in negative electrode microstructures with increasing Li to Al ratios, with the β-phase predominantly developing in the inner core of the Al particles with a thinner α-phase surface layer at higher Li ratios. In addition, a dense and intact structure of Li-Al alloy negative electrode independent of volume change after Li alloying and dealloying was also confirmed. Consequently, this study is believed to provide an in-depth understanding of the phase-dependent Li diffusion behavior for future alloy negative electrode design.

## Results

### Li diffusion kinetics in Li-Al alloy

The Li-Al alloy system has a complex phase diagram and contains a large number of stable intermetallic compounds (Supplementary Fig. 1). The first intermetallic that forms upon alloying pure Al (α-phase, which has an FCC lattice) with Li is β-LiAl, a zintl phase also known as B32-LiAl. β-LiAl is a superlattice ordering of the BCC parent crystal structure and can be described as consisting of two interpenetrating diamond cubic sublattices of Al and Li. The solubility of Li in α-Al is low ($x \leq 0.05$ in $Li_xAl_1$), while the β-LiAl compound can tolerate a slight deficiency of Li ($0.95 \leq x$ in $Li_xAl_1$). A $Li_xAl_1$ alloy with a Li composition between $0.05 \leq x \leq 0.95$ forms a two-phase mixture between $\alpha\text{-}Li_{0.05}Al_1$ and $\beta\text{-}Li_{0.95}Al_1$. The addition of Li to a two-phase mixture of α and β only modifies the relative fractions of the two phases. The conversion of $\alpha\text{-}Li_{0.05}Al_1$ to $\beta\text{-}Li_{0.95}Al_1$ upon alloying with Li results in a 96% volume expansion (Fig. 1a)[4,15]. A remarkable property of $\beta\text{-}Li_xAl_1$ is its exceptionally high Li tracer diffusion coefficient. A first-principles statistical mechanics study estimates a lithium tracer diffuson coefficient of $10^{-7}$ cm² s⁻¹. While a very large value for any intermetallic or substitutional alloy, it is ten orders of magnitude higher than the Li tracer diffusion coefficient of α-Al[38], which is estimated from first principles to be $2.6 \times 10^{-17}$ cm² s⁻¹. The high lithium tracer diffusion coefficient in β-LiAl arises from two factors: the migration barriers for lithium hops into adjacent vacancies are very low, with values around 100 meV (Fig. 1c), and β-LiAl has an unusually high concentration of vacancies[39]. In contrast, Li diffusion in the FCC crystal of Al (α-Al) is sluggish because the migration barriers for Li hops into an adjacent vacancy are high (Fig. 1b) and the equilibrium vacancy concentration is exceedingly low. Beyond a Li concentration of $x = 1$, the Li tracer diffusion coefficient is also lowered as Li diffusion is only mediated by thermally generated vacancies (Supplementary Fig. 2).

To investigate the kinetics of the alloying reaction of Al anodes with Li, galvanostatic intermittent titration technique (GITT) measurements were conducted on a pure Al electrode in a Li metal half cell (Fig. 1d, e). Due to the low Li diffusion coefficient in α-Al, diffusional trapping of Li can occur in microstructures dominated by α-Al. This results in a reversibility of delithiation to lithiation that is as low as 44% (Fig. 1d). As Al is being lithiated, the voltage polarization, which begins at 0.1 V, gradually decreases to 0.03 V at 92% of the lithiated state and rapidly increases up to the very end of lithiation (Fig. 1e).

This is interpreted as arising from the complete transformation of the original Al anode into a Li deficient β phase, which, with its high Li diffusion coefficient, then provides high mobility pathways for Li to unreacted Al. The increase in polarization as 100% of the Li state is achieved occurs when the β phase attains a $Li_1Al_1$ stoichiometry, where the vacancy concentration is significantly reduced[39]. This diffusion-dependent kinetics was also confirmed through the delithiation where the voltage polarization decreased back to 68% of Li after the vacancy formation, followed by a rapid increase due to the development of α-dominant phase at the Al surface. It should be noted that diffusion coefficients can only be extracted with GITT when applied to single phase electrodes. Nevertheless, meaningful insights about the kinetics can still be inferred from the polarization measured in GITT experiments[40,41].

### Characterization of pristine Li-Al alloy negative electrodes

Li-Al alloy negative electrodes were in situ prepared during cell assembly (Supplementary Fig. 3). An alloying reaction was induced by applying pressure to a mixed powder composite with a desired atomic ratio of Al and Li. For the pristine electrode characterization only, the powder composite was pressed without the positive electrode and SE layers. The X-ray diffraction (XRD) pattern of the Li-Al alloy negative electrodes clearly showed the development of LiAl peaks, of which the ratio to pure Al increased as the Li atomic ratio increased from 0.25 to 0.75 (Fig. 2a). In addition, increasing the Li proportion in Al improves

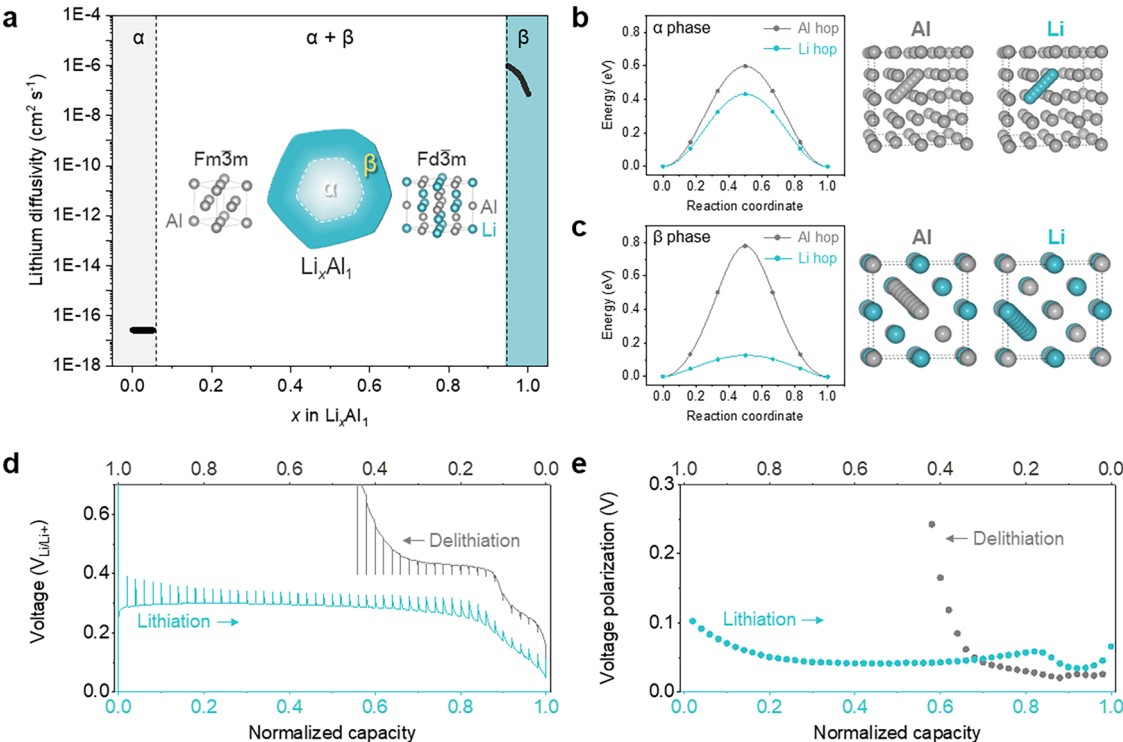

**Fig. 1 | Li diffusion kinetics in Li-Al alloy. a** Density functional theory (DFT)-based calculation on Li diffusivity in α and β phases. **b, c** Calculated migration energy of Al and Li. **b** α phase. **c** β phase. **d, e** Galvanostatic intermittent titration technique on lithiation and delithiation of pure Al in solid-state Li metal half cell (Li|LPSCl|Al). **d** Voltage profiles of galvanostatic pulses and open-circuited state. **e** Voltage polarization at each open-circuited state. Galvanostatic titration at 0.1 mA cm⁻² for 30 min; Relaxation at open circuit for 2 h; Stack pressure: 50 MPa; Testing temperature: 25 ± 1 °C.

the electronic conductivity of the Li-Al electrodes (Supplementary Fig. 4). Although Al has a higher intrinsic electronic conductivity than Li ($3.8 \times 10^5$ S cm⁻¹ for Al vs. $1.1 \times 10^5$ S cm⁻¹ for Li), the native aluminum oxide ($Al_2O_3$) on the Al particle surface significantly affects the actual conductivity. In Fig. 2b, the coexistence of β-LiAl and Al phases was confirmed by cryogenic transmission electron microscopy (cryo-TEM) characterization on a cluster of Li-Al alloy particles. The selected area electron diffraction (SAED) pattern clearly shows the main crystal lattices consistent with the XRD peaks: (111) and (220) for LiAl; (111) and (200) for Al (Inset in Fig. 2b). The high-resolution TEM image processed by fast Fourier transform (FFT) filtering further confirms the nano-domains of distributed LiAl via the lattice fringe of the (111) plane with d-spacing of 3.6 Å (Fig. 2c).

The lithiated state of Li-Al alloy negative electrode was estimated by X-ray photoelectron spectroscopy (XPS) depth profiling on Al 2*p* (Fig. 2d). The deconvoluted XPS Al 2*p* spectra show $Al_2O_3$ at 75.7 eV, $Li_xAlO_y$ at 74.8 eV, Al(OH)₃ at 73.6 eV, Al at 72.6 eV and LiAl at 71.7 eV[42]. The $Li_xAlO_y$ and LiAl peaks elucidate that the lithiation of native $Al_2O_3$ layer as well as pure Al weakens the binding ability of Al, resulting in the 0.9 eV shift to the lower-energy state. In terms of depth distribution, the ratio increase of LiAl to $Li_xAlO_y$ with the development of Al peak infers the decreasing native oxide layer toward the inner part where LiAl and Al coexist. Moreover, Fig. 2e visualizes the cross-section of particles of $Li_{0.5}Al_1$ negative electrode using cryogenic high-angle annular dark-field scanning transmission electron microscopy (cryo-HAADF-STEM), in which heavier elements are represented in brighter grayscale due to the atomic number contrast. The micron-sized $Li_{0.5}Al_1$ particles show the structure with a bright core surrounded by darker layers of < 500 nm. The energy-dispersive X-ray spectroscopic (EDS) mapping indicates that the core regions are Al while the surface regions contain O, which is not only attributed to the native oxide layer but also arises from the oxidation of the alloyed Li that was shortly exposed to the air during sample preparation and characterization

(Fig. 2f). Through the electron energy-loss spectroscopy (EELS) characterization on the spots across the interface of dark and bright areas (turquoise region 1 and gray region 2 in Fig. 2e), the signals of Li K-edge (59.5 eV and 63.5 eV) and Al L-edge (72 eV in region 1 and 75 eV in region 2) were detected (Fig. 2g). Interestingly, the Al L-edge peak shows a shift of 3 eV to the higher energy, indicating the binding ability of less lithiated Al which is consistent with the XPS result in Fig. 2d. Therefore, the combined physicochemical characterization could reveal the Li-Al alloy structure in which the Li concentration gradient is distributed from the surficial native oxide layer to the inner pure Al core.

## Electrochemical kinetics of Al-based electrodes

The Li diffusion kinetics was investigated in symmetric cells employing Li-Al alloy electrodes with different Li to Al ratios of 0.25, 0.5, and 0.75 (Fig. 3a–c). Overpotentials were compared during delithiation and lithiation at 0.25 mA cm⁻² to 0.25 mAh cm⁻² (Fig. 3a). While the voltage of $Li_{0.25}Al_1$ already spiked dramatically prior to the first lithiation, Li-Al alloy electrodes with a higher Li ratio experienced less overpotential (0.26 V for $Li_{0.5}Al_1$ and 0.23 V for $Li_{0.75}Al_1$). In the critical current density (CCD) tests, the Li-Al alloy electrodes exhibited consistent kinetic behavior, with $Li_{0.25}Al_1$ and $Li_{0.5}Al_1$ failing at a current density of 0.1 mA cm⁻² and 1.6 mA cm⁻², respectively, whereas $Li_{0.75}Al_1$ can be operated up to 12 mA cm⁻² (Supplementary Fig. 5). On the other hand, the Li symmetric cell short-circuited at 0.7 mA cm⁻² resulting from Li dendrite growth, which was not observed in the Li-Al alloy cells[43]. The significant overpotential development in the $Li_{0.25}Al_1$ and $Li_{0.5}Al_1$ without short-circuiting implies that insufficient β-phase channels result in limited Li diffusion through the α-phase grown at the surface when the alloyed Li ratio is low in both the working and counter electrodes (Supplementary Fig. 6). Therefore, $Li_{0.75}Al_1$ demonstrated stable long-term cycling performance compared to $Li_{0.5}Al_1$ which failed after a few cycles with large overpotential during lithiation and

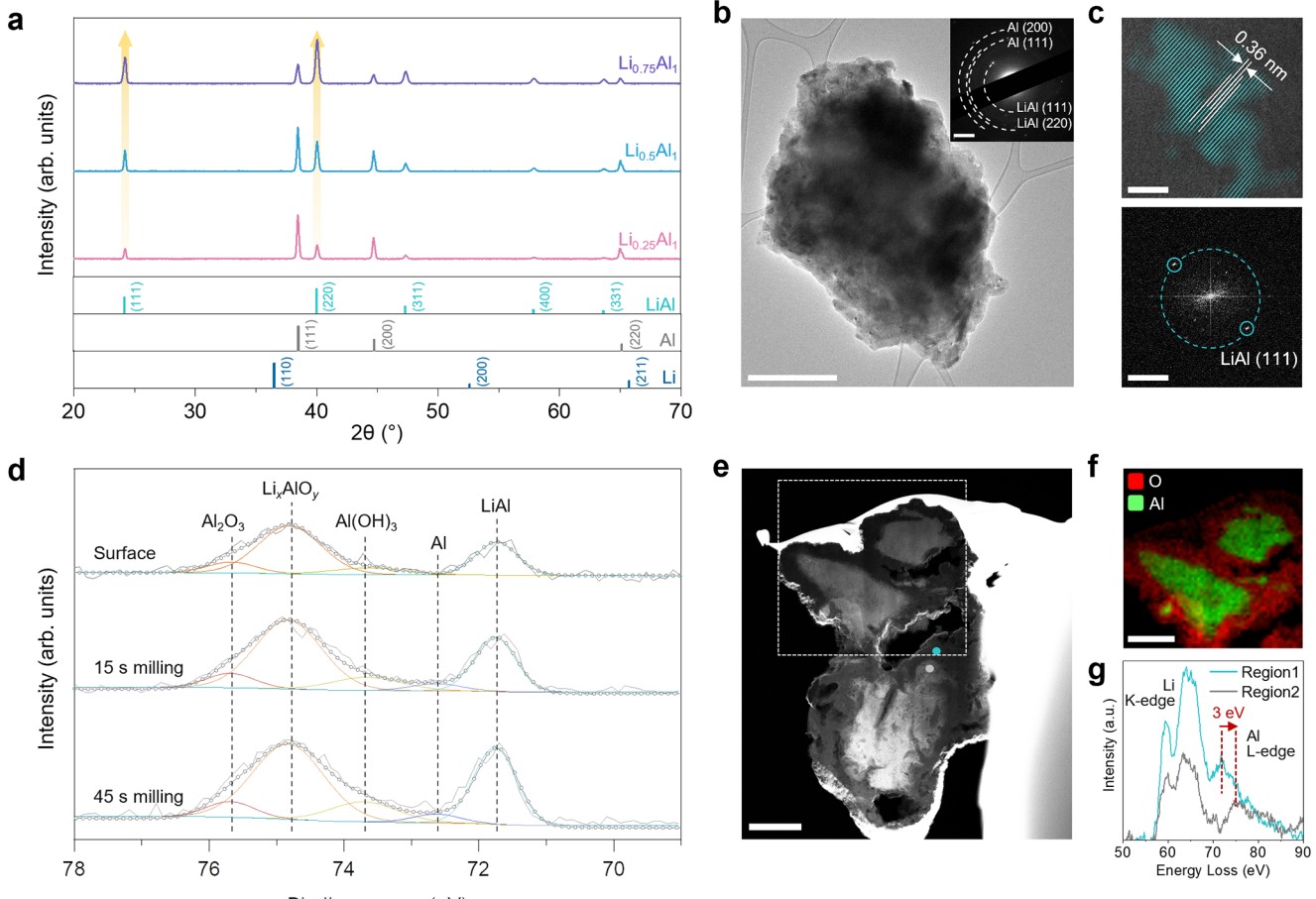

**Fig. 2 | Physicochemical characterization of pristine Li-Al alloy negative electrode. a** XRD pattern of $Li_{0.25}Al_1$, $Li_{0.5}Al_1$ and $Li_{0.75}Al_1$ negative electrodes. Yellow arrows indicate the increase of LiAl peaks at 24.2° and 40°. **b, c** Cryo-TEM characterization on a particle of $Li_{0.5}Al_1$ negative electrode. **b** TEM image at low magnification. The inset is the SAED pattern of the particles. **c** FFT-filtered high-resolution TEM image (top) and the corresponding FFT pattern (bottom) of a LiAl grain. **d** Al 2*p* X-ray photoelectron spectra (XPS) of $Li_{0.5}Al_1$ negative electrode according to the depth. **e** Cross-section cryo-HAADF-STEM image of particles of $Li_{0.5}Al_1$ negative electrode. **f** Cryo-STEM-EDS mapping of the particle in the white-dotted box in (**e**). Color indication: red = oxygen; green = aluminum. **g** EELS spectra of Li K-edge and Al L-edge obtained from region 1 (turquoise dot) and region 2 (gray dot) in (**e**). Scale bars: 1 μm for (**b**, **e**, **f**), 2 nm⁻¹ for (**b** (inset) and **c** (bottom)), and 5 nm for (**c** (top)).

delithiation at $1\,mA\,cm^{-2}$ up to $0.5\,mAh\,cm^{-2}$ (Supplementary Fig. 7). The changes in impedance elements over the cycles corroborate the voltage development as a reflection of the reaction kinetics (Fig. 3b, c and Supplementary Figs. 8 and 9). The charge transfer resistance of pristine negative electrodes ($R_n$) decreases significantly from $29.4\,\Omega$ for $Li_{0.25}Al_1$ to $12.6\,\Omega$ for both $Li_{0.5}Al_1$ and $Li_{0.75}Al_1$. The enhancement of the charge transfer before the reaction is largely related to the electronically conductive environment, which is consistent with the previous results showing that $Li_{0.5}Al_1$ and $Li_{0.75}Al_1$ provide improved electronic conductivity than $Li_{0.25}Al_1$ (Supplementary Fig. 4). After the first delithiation of $Li_{0.25}Al_1$, a huge semicircle was observed, primarily originating from the delithiated working electrode, in which the α-phase is highly developed (Supplementary Fig. 8). On the other hand, Li-Al alloy electrodes with higher Li ratio alleviated the increase in charge transfer resistance after repeated cycles: $27.5\,\Omega$ for $Li_{0.5}Al_1$ vs. $17.5\,\Omega$ for $Li_{0.75}Al_1$.

The resistances were further specified to gain an in-depth understanding of Li diffusion by converting the impedance domain from frequency to relaxation time distribution (Fig. 3c). The relaxation time window could be compartmentalized by the resistances associated with $Li^+$ transport through the SE grain boundary ($R_{gb}$) and SE interphase ($R_{SEI}$), (de)lithiation of the host materials ($R_{ct}$) and solid diffusion within the host matrix ($R_{diff}$)[44,45]. Compared to the pristine

case, the cycled $Li_{0.5}Al_1$ and $Li_{0.75}Al_1$ showed the averaged $R_{ct}$ with significant decrease at $10^{-1} - 1\,s$ and increase in resistance at lower relaxation time and significantly suppressed $R_{diff}$ (Top and middle in Fig. 3c). This is most possibly resulted from the homogenization of the β-dominant phase due to the Li interdiffusion between the separated α and β phases, and the particle size reduction by pulverization during cycling, which will be further discussed in the following section. On the contrary, a large increase of both $R_{ct}$ and $R_{diff}$ in the $Li_{0.25}Al_1$ case was confirmed, indicating dominant growth of α-phase, which prevents Li from moving out in the first delithiation step (Bottom in Fig. 3c). The GITT and in situ EIS analysis across the lithiated state further confirmed the gradual change of Li diffusion kinetics in Li-Al alloy negative electrodes compared to pure Al (Supplementary Fig. 10). It was consistently observed that the Li-Al alloy negative electrodes with predominant β phase, developed at higher Li ratio, exhibited lower voltage polarization and resistances during lithiation.

The rate capability of the Al-based electrodes was then investigated in an NCM811-based full-cell configuration, where the performance is limited by the Al-based negative electrode, in contrast to the symmetric cell system limited by both the working and counter electrodes (Fig. 3d–f). The Li-Al alloy with higher Li ratio exhibited superiority in rate capability tests up to $7\,mA\,cm^{-2}$ (Fig. 3d and Supplementary Fig. 11). However, the Li||NCM811 cell short-circuited at

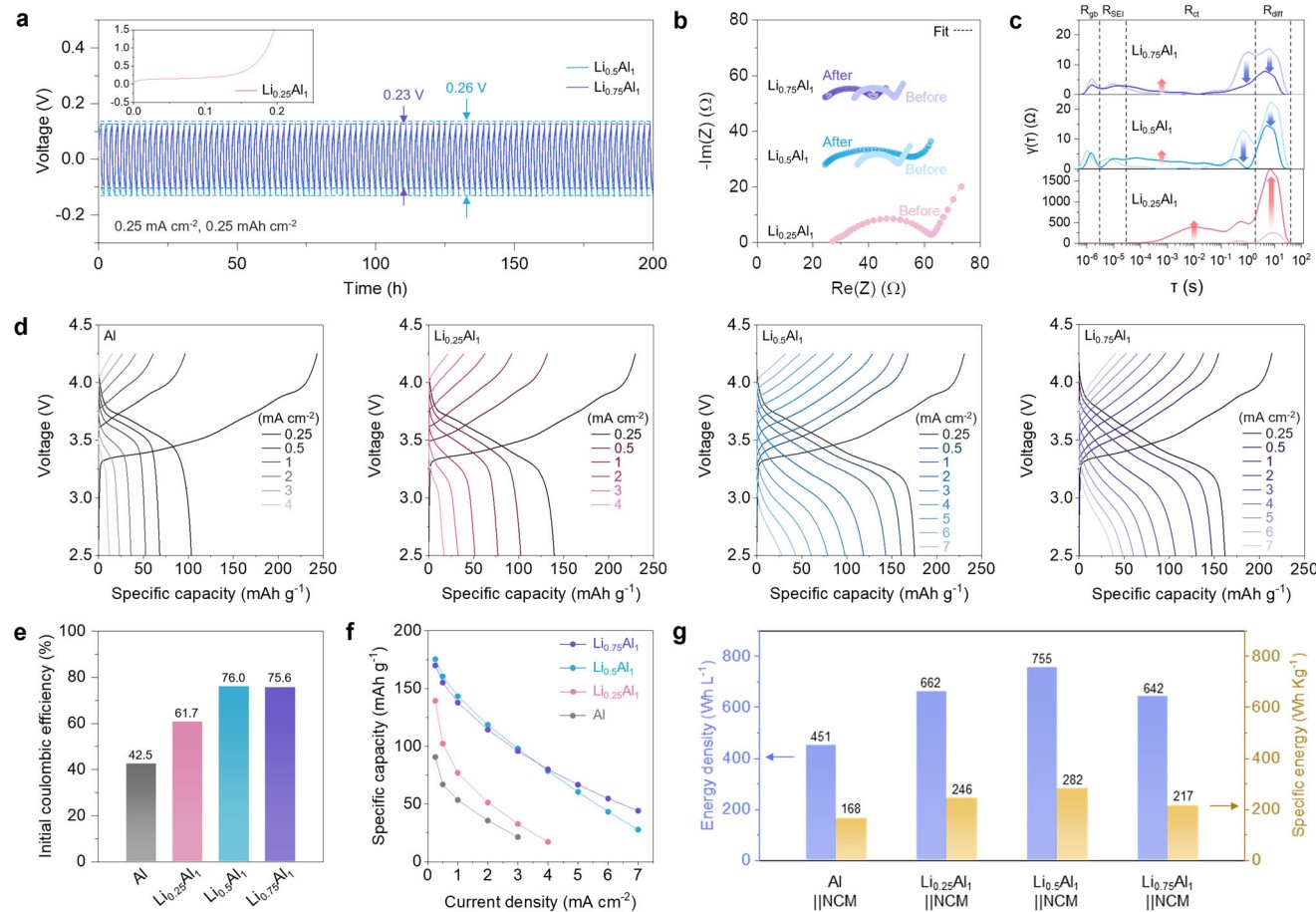

**Fig. 3 | Electrochemical kinetics behaviors of Li-Al alloy electrodes. a** Long-term cyclability in symmetric cell with $Li_{0.25}Al_1$, $Li_{0.5}Al_1$ and $Li_{0.75}Al_1$ negative electrodes at 0.25 mA cm$^{-2}$ to 0.25 mAh cm$^{-2}$. **b, c** Electrochemical impedance measurements on the pristine and 200 h-cycled Li-Al electrodes. **b** Nyquist plot of impedance. **c** Distribution of relaxation time (DRT) spectra. **d** Rate capability test of NCM811-based full cells with pure Al, $Li_{0.25}Al_1$, $Li_{0.5}Al_1$ and $Li_{0.75}Al_1$ negative electrodes. **e** ICE of pure Al, $Li_{0.25}Al_1$, $Li_{0.5}Al_1$ and $Li_{0.75}Al_1$ negative electrodes. **f** Specific capacities comparison at each current density. **g** Energy density (Wh L$^{-1}$) and specific energy (Wh kg$^{-1}$) of ASSB with the configuration of pure Al and Li-Al alloy-based negative electrodes and NCM811 positive electrode. Stack pressure for the Li-Al symmetric cell and Li-Al||NCM811 full-cell tests: 75 MPa. N/P ratio for full cells: 2. Testing temperature: 25 ± 1 °C.

a current density of 1 mA cm$^{-2}$, since the CCD of Li is 0.7 mA cm$^{-2}$ as confirmed in the symmetric cell tests (Supplementary Fig. 12). In Fig. 3e, it was readily seen that the initial coulombic efficiency (ICE) of charging and discharging at 0.25 mA cm$^{-2}$ to 2.5 mAh cm$^{-2}$ was largely improved for the Li-Al alloy negative electrodes compared to the pure Al negative electrode (e.g. 42.5% for pure Al vs. 76% for $Li_{0.5}Al_1$). Above the Li ratio of 0.5, ICE was saturated to 76% due to the interface degradation of the NCM positive electrode, which was also observed in LiCoO$_2$ (LCO)-based full-cell system with 91% of ICE saturation (Supplementary Fig. 13)[46]. At higher current densities, a relatively higher proportion of Li ($x \geq 0.5$) in the Li-Al alloy negative electrode led to higher specific capacity through further improved Li diffusion (Fig. 3f). It is noted that the stack pressure of 75 MPa was applied in both symmetric and full-cell tests to maintain intact particle contacts, particularly in the positive electrode composite as well as the Li-Al alloy negative electrode, to solely investigate the β-phase-dependent Li diffusion mechanism[35]. Exploring advanced interface strategies will be key to enabling practical large-scale solid-state batteries with Li-Al alloy negative electrodes operating under low stack pressure.

Based on the electrochemical studies in symmetric and full cells, the Li ratio was identified as a key parameter for the phase control. Additionally, N/P ratio is considered as another critical parameter in controlling phase development of Li-Al negative electrode in full cells (Supplementary Figs. 14 and 15). The N/P ratio would be defined as the ratio of the accessible capacity of the negative electrode to that of the positive electrode. Decreasing the N/P ratio then reduces the capacity of Li-poor α-phase Al, which promotes more utilization of β-phase of the Li-Al alloy negative electrode with a fixed positive electrode loading capacity. The rate capability tests on various N/P ratios revealed that the kinetics of $Li_{0.25}Al_1$ negative electrode and pure Al at N/P ratio of 1.2 were much superior to other N/P ratios of 2 and 1.5 as shown in the improvement of ICE and discharged specific capacities (Supplementary Fig. 15a, c, d). $Li_{0.5}Al_1$ negative electrode, which already exhibited abundant β phase, did not show a noticeable improvement (Supplementary Fig. 15a, b). Through the systematic parameter study on the phase control, it can be concluded that the capacity utilization is highly dependent on the Li diffusion kinetics in the Li-Al alloy. In order to construct a practical cell-level design, the effective capacity of the Al-based full cells should thus be considered rather than only using the theoretical capacity of the active materials (Fig. 3f). Consequently, using the obtained effective full-cell capacities, the $Li_{0.5}Al_1$-based full-cell system was projected to provide optimal energy density and specific energy (755 Wh L$^{-1}$ and 282 Wh kg$^{-1}$) compared to other Al-based negative electrodes (Fig. 3g, Supplementary Fig. 16 and Table S2).

## Demonstration of Li-Al alloy negative electrode-based high-loading ASSB

NCM811-based full cells with a high positive electrode loading of 5 mAh cm$^{-2}$ were tested to demonstrate the feasibility of the $Li_{0.5}Al_1$

negative electrode. The $Li_{0.5}Al_1$||NCM811 cell showed superior rate performance compared to the pure Al counterpart (Supplementary Fig. 17). The pure Al exhibited capacity utilization of only 33% even at 0.05C due to the Li trapped in the α-dominant phase. To investigate the maintenance of the utilized capacity, long-term cycling of $Li_{0.5}Al_1$ negative electrode was conducted at 0.33C (1.65 mA cm$^{-2}$) and 0.8C (4 mA cm$^{-2}$) at which pure Al is not expected to deliver meaningful capacities (Fig. 4a–d). At 0.33C, $Li_{0.5}Al_1$ negative electrode realized the high capacity of 3.5 mAh cm$^{-2}$ and capacity retention of 80% after 400 cycles (Fig. 4a, b). Even at higher current density of 0.8C, cycles of up to 2000 were achieved without significant changes in resistances (Fig. 4c, d and Supplementary Figs. 18–20).

The cyclability could be attributed to the morphological stability as well as the β-dominant phase of the $Li_{0.5}Al_1$ negative electrode during repetitive charging and discharging. The cross-sectional morphology and phase development of the $Li_{0.5}Al_1$ negative electrode at pristine, charged, and discharged states were then visualized in secondary electron (SE) and backscattered electron (BSE) modes using plasma focused ion beam scanning electron microscopy (PFIB-SEM) (Fig. 4e–h). The $Li_{0.5}Al_1$ negative electrode maintained its dense structure with very few voids despite the relatively large volume change during the charging (39 μm to 59 μm) and discharging (59 μm to 47 μm) (Left in Fig. 4e–g). The good electrode integrity is resulted from the intimate contact between the ductile Li and Al particles after fabrication, which helps their morphology to fit the surrounding structures (Middle in Fig. 4e–g). Therefore, close and intimate contact between the expanded Li-Al alloy particles was observed after charging, (Middle in Fig. 4f) which was further enhanced by the particles pulverized to a few micrometers in size during discharge (Middle in Fig. 4g). The pulverized particles, which could densify the electrode structure, would influence the improvement in interface and charge transfer resistances during the early cycles (Fig. 4c). Additional cross-sectional characterization of the pure Al and other Li-Al alloy negative electrodes also confirmed that the electrode density could be improved by increasing the Li ratio, due to enhanced particle pulverization (Supplementary Fig. 21). Correspondingly, the porosity measured in the 1$^{st}$ discharged state was significantly reduced in the Li-Al alloy negative electrodes compared to pure Al (Supplementary Fig. 22). Interestingly, even the 2000-cycled $Li_{0.5}Al_1$ negative electrode maintained a highly dense structure by the pulverized particles continuing to reduce to sub-micrometer size during repetitive volume changes (Supplementary Fig. 23). In the BSE mode, the phase distribution across the electrode could be distinguished by contrast: light gray = α-Al; dark gray = β-LiAl; darkest gray = Li (Right in Fig. 4e–g). The pristine $Li_{0.5}Al_1$ electrode showed localized $Li_1Al_1$ even though the electrode was prepared by pressing a thoroughly mixed powder composite of Li and Al (dashed orange box on the Right in Fig. 4e). This is interpreted as the Li-Al alloying reaction spreading from the darkest spot where pristine Li powder was initially located. After the first charge, interestingly, the contrast of cross-section turned to be homogeneous with a few nanometer-thick α layers, representing uniform distribution of the β-LiAl and complete consumption of Li metal (Right in Fig. 4f). The charged $Li_{0.5}Al_1$ is regarded as a Li conductor with highly developed Li transport channels throughout the electrode. After discharging, the electrode consisted of pulverized particles surrounded by thin α-phase layer (Right in Fig. 4g). XRD characterization of the $Li_{0.5}Al_1$ negative electrode clearly revealed this phase development in which the α phase significantly decreased after charging and fully recovered after discharging (Supplementary Fig. 24). More notably, the β-LiAl was observed to be relatively concentrated on the SE side far from the copper current collector. When Li moves toward the SE during discharging, it could be inferred that the Li concentration gradient, generated at the interface between the Li-absent vacancy and the Li-present β-LiAl lattice, may possibly drive Li diffusion

toward the SE layer[34]. Thus, highly developed β-LiAl phase in the Li-Al alloy negative electrodes with a higher Li ratio could facilitate Li diffusion to the SE side, serving as efficient Li transport channels. The insight could be further supported by observing the cross-sections of discharged pure Al and other Li-Al alloy negative electrodes in BSE mode (Supplementary Figs. 21 and 25). The $Li_{0.5}Al_1$ and $Li_{0.75}Al_1$ negative electrodes with a higher Li ratio showed that pulverized particles composed of small β phase domains ($Li_{0.5}Al_1$: 2.326 μm; $Li_{0.75}Al_1$: 1.607 μm) surrounded by thin α-phase layer, and the dark β-LiAl layer is mostly distributed at the SE-negative electrode interface. Meanwhile, the light gray α-Al layer was clearly observed at the interface of the pure Al and $Li_{0.25}Al_1$ negative electrodes with β-phase island (pure Al: 4.877 μm; $Li_{0.25}Al_1$: 2.783 μm) enclosed by thick α-phase layer.

## Phase-dependent Li diffusion in Li-Al negative electrodes

Building on insights from DFT studies and physicochemical, electrochemical, and morphological characterizations, we propose a mechanism for phase-dependent Li diffusion in Li-Al negative electrodes within the ASSB system (Fig. 5a). In pure Al, the intrinsic Li-trapping problem is inevitable due to the significantly disparate Li diffusion kinetics between the Li-poor α phase and Li-rich β phase (Fig. 5b)[47–49]. During the initial alloying process, the β phase grows on the surface and gradually penetrates into the intact α phase in the inner region (Fig. 5b-i). In a subsequent dealloying step, Li diffuses out from the outermost surface where the α phase begins to grow on the surface and act as a diffusion barrier that prevents the residual Li dealloying from inside, leading to irreversible Li trapping (Fig. 5b-ii). In the following alloying step, the Li sources are limited in reconnecting the β-phase due to the trapped Li during the previous dealloying step (Fig. 5b-iii). Contrarily, the Li-Al alloy with pre-developed β phase is expected to facilitate the diffusion when accepting Li from the positive electrode (Fig. 5c-i). During dealloying, the Li diffusion is facilitated through the reduced α phase distance with a higher driving force induced by the predominantly developed internal β phase[26,35]. Then, the reconnected β phase between the interior and the surface during the next alloying process serves as good conductive channels for the stored Li (Fig. 5c-iii). This study on the relationship between predeveloped β-LiAl phase and the Li diffusion behavior emphasizes the importance of phase design in Al-based negative electrodes and demonstrates a mechanism consistent with the previous studies on the pure Al negative electrodes, offering guidance for the development of Al negative electrode-based ASSB[34,49].

In this work, we designed the kinetics-controlled Li-Al alloy negative electrode based on the in-depth study of phase-dependent Li diffusion. DFT-based simulation showed the superiority of the Li-rich β phase with Li diffusivity (~10$^{-7}$ cm$^2$ s$^{-1}$) ten-orders-of magnitude higher than that of Li-poor α phase (2.6 × 10$^{-17}$ cm$^2$ s$^{-1}$). It was confirmed that the enhanced Li diffusion kinetics of Li-Al alloy negative electrodes through increasing the ratio of alloyed Li resulted in reduced voltage polarization and resistance in symmetric cells. This improvement also led to enhanced ICE and higher capacities in the rate capability tests of full cells incorporated with NCM811 positive electrode. Based on the practical perspective of the phase-dependent Li-Al alloy negative electrode, the $Li_{0.5}Al_1$ negative electrode configured of NCM811 offers the optimal energy density (755 Wh L$^{-1}$) and specific energy (282 Wh kg$^{-1}$) among the Al-based full cells. In addition, the $Li_{0.5}Al_1$ negative electrode demonstrated stable cycling up to 2000 cycles at the high current density of 4 mA cm$^{-2}$ with the high positive electrode loading of 5 mAh cm$^{-2}$. This is attributed to the dense structure of $Li_{0.5}Al_1$ negative electrode and β-dominant phase retained toward the SE side during charging and discharging. This work provides insight and design strategy of low-cost metal alloy negative electrode for ASSBs.

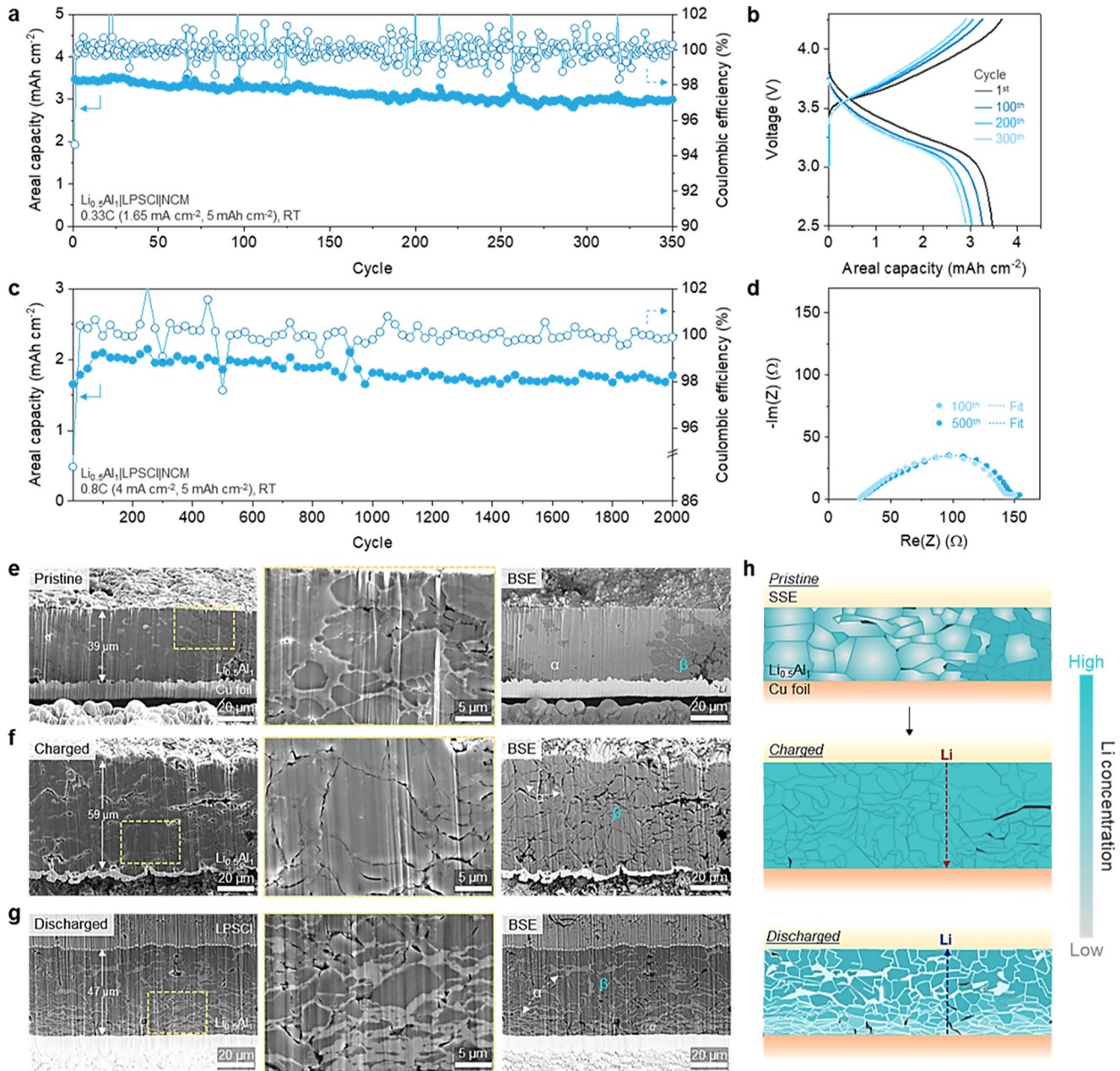

**Fig. 4 | Long-term cycling performance and morphological change of $Li_{0.5}Al_1$ negative electrode with high-loading NCM811 positive electrode. a–d** Long-term capacity retention test and voltage profiles at corresponding cycle. **a** Long-term cyclability at 0.33C (1C = 5 mA cm⁻²). **b** Voltage profiles at 1st, 100th, 200th and 300th cycle. **c** Long-term cyclability at 0.8C (1C = 5 mA cm⁻²). All the data points were selected at every 25 cycles. **d** EIS analysis at 100th and 500th cycle. Stack pressure: 75 MPa. N/P ratio: 2. Testing temperature: 25 ± 1 °C. **e–g** PFIB-prepared cross-sections of $Li_{0.5}Al_1$ negative electrode. **e** Pristine. **f** After 1st charge. **g** After 1st discharge. The area of yellow box was enlarged to high-magnification images at the middle row. The first-row images in backscattered mode were shown at the right-most row. **h** Schematics of $Li_{0.5}Al_1$ negative electrode representing morphological change, Li diffusion pathway and phase development and during charging and discharging.

## Methods

### Materials preparation

Al powder (≤ 30 µm, ≥ 99%, Sigma-Aldrich), Li powder (SLMP®, ≥ 97%, Livent), lithium nickel cobalt manganese oxide (NCM811; LG Energy Solution), lithium cobalt oxide (LCO; MSE Supplies LLC) with 1 wt. % of lithium niobium oxide ($LiNbO_3$) coating layer, $Li_6PS_5Cl$ (LPSCl; NEI Corporation), vapor-grown carbon fiber (VGCF; > 98%, Sigma-Aldrich), polytetrafluoroethylene (PTFE; < 300 nm, Chemours), poly(styrene-ethylene/butylene-styrene) block copolymer (SEBS; 99.97%, Sigma-Aldrich) and xylene (≥ 98.5%, Fisher Scientific) were used as received. LPSCl was further ball-milled at 400 rpm for 2 h with 1 min rest time

every 1 h, using high-energy ball-milling machine (PBM-0.4 A, Xiamen Tmax Battery Equipments Ltd.) and 50 mL air-tight zirconia ball-milling jars for the positive electrode composite. The ball-to-LPSCl mass ratio was 36:1, using 5 mm zirconia balls. All the ball-milling preparation was progressed in the glovebox (Vigor; $O_2 ≤ 0.05$ ppm, $H_2O ≤ 0.02$ ppm, Argon atmosphere).

### Electrodes preparation

**Pure Al electrode.** The pure Al electrode slurry was prepared by mixing Al with 0.1 wt. % of SEBS in xylene solvent for 15 min. The slurry was cast on the non-polished side of Cu foil by a doctor blade and dried

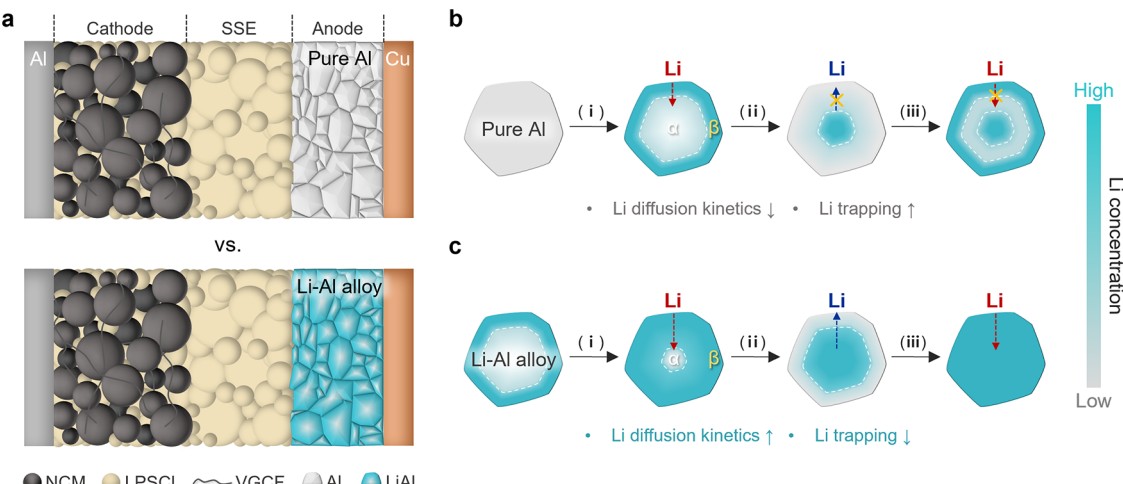

**Fig. 5 | Phase development of Li-Al negative electrodes in ASSB system. a** ASSB configuration based on pure Al and Li-Al alloy negative electrodes with NCM811-based positive electrode and LPSCl SSE layer. **b, c** Mechanism of Li diffusion and phase development in pure Al and Li-Al alloy negative electrode during Li alloying and dealloying. **b** Pure Al. **c** Li-Al alloy. (i) = first Li alloying; (ii) = first Li dealloying; (iii) = second Li alloying. The schematics represent the average lithiation state of the electrode at each stage.

at 80 °C for 2 h. The Al loading density was 2.5 mg cm⁻² equivalent to 2.5 mAh cm⁻².

**Li-Al powder composite.** The Li and Al powder was mixed to prepare Li-Al composite at the atomic ratio of $x$ to 1 in $Li_xAl_1$ ($x$ = 0.25, 0.4, 0.5, 0.75, 0.9 and 0.95) for 10 min at 1900 rpm using a vortex mixer (MI0101002D, Four E's Scientific). The pristine Li-Al alloy electrodes used for XRD, cryo-TEM and XPS characterizations were prepared in the Ar-filled glovebox by pressing the Li-Al powder composite at a corresponding Li atomic ratio under 125 MPa for 10 s, followed by 6 h resting at 75 MPa.

**Positive electrode powder composite.** The NCM powder composite was prepared by hand mixing NCM811, ball-milled LPSCl and VGCF in mortar and pestle at 66 to 31 to 3 in mass ratio. NCM811 and ball-milled LPSCl powder were first mixed for 20 min, and VGCF was added to the composite, followed by 10 min mixing. The LCO powder composite was prepared using the same process and composition as NCM811. All positive electrode powder composites were prepared in the Ar-filled glovebox.

**NCM dry film electrode.** In the Ar-filled glovebox, the NCM powder composite was prepared first by hand mixing NCM811, ball-milled LPSCl, VGCF and PTFE in mortar and pestle at 66 to 31 to 3 to 0.5 in mass ratio. NCM811 and ball-milled LPSCl powder were mixed first for 20 min, and then VGCF was mixed with the composite for further 10 min followed by PTFE addition. The prepared powder composite was kneaded in a heated mortar to obtain a dough for the film fabrication. The dough was rolled repeatedly at 90 °C until the target loading density (1C = 5 mAh cm⁻²) was achieved by gradually reducing the gap between the stainless-steel rods of hot roller (Xiamen Tmax Battery Equipments Ltd.).

**Solid-state cell assembly**
The solid-state cells were assembled with two titanium plungers and one polyetheretherketone (PEEK) holder with diameter of 10 mm in the glove box.

**Half cell.** Li metal half-cells with pure Al and Li-Al alloy negative electrodes were used for GITT and in situ EIS analyses. For a pure Al half cell, a pure Al electrode disc (radius = 5 mm) was prepared as

the working electrode, and a Li metal foil (radius = 5 mm, thickness = 20 μm) was used as the counter electrode. A total of 70 mg of LPSCl was spread on a titanium plunger in a PEEK holder and pressed with the other plunger at 310 MPa for 10 s to first prepare a separator layer. The pure Al electrode disc was then placed on one side of the separator layer and pressed at 375 MPa for 3 min, followed by insertion of the Li metal foil on the opposite side. For half cells with Li-Al alloy negative electrodes, 70 mg of LPSCl was spread and pressed between two titanium plungers at 375 MPa for 3 min. The Li-Al powder composite for $Li_xAl_1$ negative electrode ($x$ = 0.25, 0.5 and 0.75) was spread on one side of the separator layer and pressed at 125 MPa for 10 s, followed by insertion of the Li metal foil on the opposite side. The assembled half-cells were rested under a stack pressure of 55 MPa for 6 h before testing.

**Symmetric cell.** A total of 70 mg of LPSCl was spread and pressed between two titanium plungers at 375 MPa for 3 min. The Li-Al powder composite for $Li_xAl_1$ negative electrode ($x$ = 0.25, 0.5 and 0.75) was spread on both sides of the separator layer and pressed at 125 MPa for 10 s. The assembled cells were rested under a stack pressure of 75 MPa for 6 h before testing.

**Full cell.** A SE layer was first prepared by pressing 70 mg of LPSCl at 310 MPa for 10 s. NCM powder composite or NCM dry film was placed on one side of the layer and pressed at 375 MPa for 3 min, followed by $Li_xAl_1$ ($x$ = 0.25, 0.4, 0.5, 0.75, 0.9 and 0.95) negative electrode preparation on the opposite side by pressing at 125 MPa for 10 s. The assembled cells were rested under a stack pressure of 75 MPa for 6 h before testing. The used loading density of the positive electrode was 9.57 mg$_{ncm}$ cm⁻² (2.5 mAh cm⁻²) for the NCM positive electrode composite and 25.1 mg$_{ncm}$ cm⁻² (5 mAh cm⁻²) for the NCM dry film. The desired amount of Li-Al composite powder was used according to the designed NP ratio of 1.2, 1.5 or 2 based on the theoretical capacity of $Li_xAl_1$.

**Electrochemical measurements**
All battery tests were performed on a galvanostat (BTS4000, Neware) inside the glove box at 25 ± 1 °C. Galvanostatic intermittent titration technique (GITT) was used to measure the voltage polarization development in open circuit for 2 h after galvanostatic titration at 0.1 mA cm⁻² for 30 min during lithiation to 0.01 $V_{Li/Li+}$ and delithiation

to 1.5 $V_{Li/Li+}$ of pure Al electrode at stack pressure of 50 MPa. The long-term cycling in the symmetric-cell configuration ($Li_{0.25}Al_1$, $Li_{0.5}Al_1$ and $Li_{0.75}Al_1$) was tested with galvanostatic lithiation and delithiation at 0.25 mA cm$^{-2}$ to 0.25 mAh cm$^{-2}$ and at 1 mA cm$^{-2}$ to 0.5 mAh cm$^{-2}$, under stack pressure of 75 MPa. The critical current density test in the symmetric cells (Li, $Li_{0.25}Al_1$, $Li_{0.5}Al_1$ and $Li_{0.75}Al_1$) was conducted with galvanostatic lithiation and delithiation up to 0.25 mAh cm$^{-2}$, with current densities from 0.05 to 12 mA cm$^{-2}$. NCM811-based full cells with pure Al and $Li_xAl_1$ ($x$ = 0.25, 0.4, 0.5, 0.75, 0.9 and 0.95) were galvanostatically charged up to 4.25 V and then discharged to 2.5 V. Constant-voltage step at 4.25 V was not included in the charging. For the rate capability test, the current density was ramped up from 0.25 mA cm$^{-2}$ to 1 mA cm$^{-2}$ at the stack pressures from 75 MPa to 5 MPa with NP ratios of 1.2, 1.5 and 2. Used conditions of the stack pressure and NP ratio were noted in corresponding figures. The high-loading NCM811-based full cells were precycled by gradually ramping up the C-rate to the long-term cycling rate in the order of 0.05C, 0.1C, 0.2C, 0.4C, and 0.6C. The applied stack pressure was measured and monitored by load sensor calibrated with Instron Loadframe. The potentiostatic electrochemical impedance spectra (PEIS) of symmetric and full cells was measured using a potentiostat (VSP-300, BioLogic) immediately after cell assembly and cycling. PEIS circuits were obtained under the following conditions: at open-circuit voltage; frequency range from 7 MHz to 0.05 Hz; sinusoidal amplitude of 10 mV; and 6 data points per decade in Logarithmic spacing. Distribution of relaxation time (DRT) spectra of symmetric cells was converted from PEIS circuit using MATLAB.

## Material characterization

The X-ray diffraction (XRD) patterns of the $Li_xAl_1$ alloy electrodes were obtained at 1° min$^{-1}$ (XRD; SmartLab, RIGAKU with Cu-Ka1 radiation). All electrode samples were prepared in heat-sealed bags inside the Ar-filled glovebox prior to transfer to the XRD chamber. The lithiated state of $Li_{0.5}Al_1$ was investigated in Al 2$p$ by X-ray photoelectron spectroscopy (XPS; AXIS Supra, Kratos Analytical) with an Al monochromatic anode source at 15 kV with a 5.0 × 10$^{-8}$ Torr vacuum level. The spot area for XPS measurements is 300 μm × 700 μm. All electrode samples were transferred in a heat-sealed bag inside the Ar-filled glovebox before they were transferred to the XPS glovebox. The XPS sample transfer is conducted under N$_2$ environment to avoid air-exposure. Survey scans were collected with a 1.0 eV step size, followed by high-resolution scans with a step size of 0.1 eV for Al 2$p$. Etching using Argon1000+ gas cluster ion source (GCIS) accelerated at 10 keV was carried out to clean the surface and mill the electrode over a raster area of 2 mm × 2 mm with energy of 1000 keV for 15 s. The cross-sectional morphology of electrode was visualized by scanning electron microscopy (SEM; Helios 5 PFIB Dual-Beam, Thermo Fischer Scientific) equipped with Xenon plasma focused ion beam (PFIB) column. All electrode samples were prepared in heat-sealed bags inside the Ar-filled glovebox and immediately transferred to the SEM chamber after opening.

## Cryogenic transmission electron microscopy (cryo-TEM) characterization

The $Li_{0.5}Al_1$ particles for cryo-TEM characterization were prepared by hand grinding the pressed $Li_{0.5}Al_1$ pellet into particles using a mortar and pestle. The ground $Li_{0.5}Al_1$ alloy particle clusters were directly dispersed onto a Cu grid and loaded to a Gatan 636 cooling holder for cryo-TEM analysis at liquid-nitrogen temperatures. All sample preparation and loading were conducted inside a glovebox. To minimize side reactions, the sample was sealed with an airtight cover prior to brief air exposure during insertion into the TEM column. TEM images and electron diffraction patterns were captured with a JEOL JEM-2100F TEM at 200 kV, equipped with a Gatan K3 direct electron camera and a Gatan OneView camera. To minimize electron beam damage to the Li-Al alloy crystal structure, the electron dose was kept below 150 e$^-$/Å$^2$ s.

High-resolution transmission electron microscopy (HRTEM) images along with fast Fourier transform (FFT) pattern and FFT-filtering were analyzed using Gatan Microscopy Suite software. For further elemental analysis, Li-Al alloy particles were thinned using a Tescan GAIA3 SEM-FIB on a Cu FIB grid. The characterization of electron-dispersive X-ray spectroscopy (EDS) or electron energy-loss spectroscopy (EELS) in scanning transmission electron microscopy (STEM) mode was conducted using a JEOL JEM ARM300F Grand ARM TEM operated at 300 kV and equipped with aberration correctors, Gatan K2 Summit camera, and dual 100 mm² Si drift detectors. A probe convergence semi-angle of 25.7 mrad, a spectrometer collection semi-angle of approximately 35 mrad, and an energy dispersion of 0.05 eV per channel were used for acquiring high-angle annular dark-field (HAADF)-STEM images, Al and O EDS elemental mapping, and Li K-edge and Al L-edge EELS spectra. All TEM and STEM imaging as well as EDS and EELS analyses were conducted at cryogenic conditions.

## Computational details

Density functional theory (DFT) calculations performed in this work used the Vienna ab initio Simulation package (VASP)[50–52] within the GGA-PBE approximation[53]. All the calculations used the projector-augmented-wave (PAW) pseudopotentials[54] with valence electronic configurations of $1s^2 2s^1$ for Li and $3s^2 3p^1$ for Al respectively. Convergence tests were performed for the plane-wave basis set energy cutoff (ENCUT) and the $\Gamma$-centered k-point mesh spacing to ensure the error from DFT is within 1 meV/ atom. A value of 650 eV for ENCUT and $\frac{2\pi}{64}$ Å$^{-1}$ for the k-point spacing was used for all the calculations. Within the electronic self-consistent loop, the energies were converged to a tolerance of 10$^{-5}$ eV. While performing the geometric structure optimization within DFT, the forces were converged to a tolerance of 0.02 eV/Å.

All the Li and Al hops to a nearest neighbor vacant site in the α- and β-phase were enumerated using the CASM software package[55–57]. The migration barriers for the enumerated hops were calculated in a 108-atom supercell for the α-phase and 128-atom supercell for the β-phase using Nudged Elastic Band (NEB) calculations as implemented in the Transition State Tools (TST) for VASP[58]. A climbing image method[59] with a quick-min optimizer[60] was used for all the NEB calculations to determine the migration barrier for the hop. The energies and forces in the NEB calculations were also converged to a tolerance of 10$^{-5}$ eV and 0.02 eV/Å, respectively. The NEB calculation parameters and the corresponding initial and final configurations of the α- and β-phases are provided in Supplementary Data 1 and 2.

To accurately determine the diffusion coefficients in an alloy, rigorous kinetic Monte Carlo (kMC) simulations, which thoroughly accounts for the short- and long- range order effects around a vacancy, are warranted[57,61–63]. This is usually achieved by constructing a surrogate cluster expansion model based on first-principles calculations[57,61]. Nevertheless, an approximate estimate of the tracer diffusion coefficients can be calculated using the following relation,

$$D^* = gr^2 f \nu^* x_{va} e^{\frac{-E^*}{k_B T}} \tag{1}$$

where $g$ is the geometric prefactor which can be approximated by $\frac{\Gamma}{6}$ ($\Gamma$ is the number of nearest neighbors available for the hopping atom)[64], $r$ is the nearest neighbor distance, $f$ is the correlation factor, $\nu^*$ is the vibrational prefactor, $x_{va}$ is the vacancy concentration, $E^*$ is the energy barrier for migration, $k_B$ is the Boltzmann constant and $T$ is the temperature.

The tracer diffusion coefficient of Li in the FCC α-phase at 25 ± 1 °C was calculated using Eq. (1) with $g$ = 2 ($\Gamma$ = 12 for FCC), $f$ = 0.78 (FCC)[65], $r$ = 2.85A, $\nu^*$ = 7 × 10$^{13}$ Hz[31], $x_{va}$ = 5.29 × 10$^{-9}$ and $E^*$ = 0.43 eV. The equilibrium vacancy concentration ($x_{va}$) in the α-phase is calculated by obtaining the DFT vacancy formation energy ($\Delta\Omega$) within the grand-canonical ensemble[32,66] in a 128-atom supercell of the FCC structure using $e^{\frac{-\Delta\Omega}{k_B T}}$.

The crystal structure of the BCC $\beta$-phase can be viewed as two interpenetrating diamond cubic (DC) sublattices of Li and Al[32]. Since Al is predicted to form a very rigid sublattice in the $\beta$-phase[32], only diffusion of Li on its own sublattice was considered. Using Eq. (1), the tracer diffusion coefficient at $25 \pm 1\,°C$ in the $\beta$-phase was calculated with $g = \frac{4}{6}$ ($\Gamma = 4$ for DC), $f = 0.5$ (DC)[65], $r = 2.75\text{\AA}$, $\nu^* = 10^{13}$ Hz and $E^* = 0.13\,eV$. For compositions ($x$) between 0.95 and 1 in Li$_x$Al, vacancies are predicted to be thermodynamically stable[32] and a vacancy concentration ($x_{va}$) of $1 - x$ is used for calculating the tracer diffusion coefficient of Li. While this is an approximation, accurate equilibrium vacancy concentrations, along with the corresponding tracer diffusion coefficients in the β-phase, can be obtained through rigorous (kinetic) Monte Carlo simulations[67]. For compositions between 1 and ~1.15, the vacancy concentration ($x_{va} = e^{\frac{-\Delta\Omega}{k_B T}}$) is interpolated smoothly using the DFT vacancy formation energies ($\Delta\Omega$) of four structures at various compositions calculated within the grand-canonical ensemble.'

## Data availability

Source data are provided with this paper.

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

## Acknowledgements

This work was supported by LG Energy Solution—U.C. San Diego Frontier Research Laboratory (FRL) via the Open Innovation program (Y.J., D.J.L., J.-P.L., J.K., M.-S.S. and Z.C.). This work was partially performed at the San Diego Nanotechnology Infrastructure (SDNI) of U.C. San Diego, a member of the National Nanotechnology Coordinated Infrastructure, which is supported by the National Science Foundation (Grant ECCS-1542148, Y.J., D.J.L., J.-P.L., J.W., F.L., W.T., L.Z., Y.-T.C., D.X., J.K., M.-S.S. and Z.C.). The authors (H.Z. and K.H.) acknowledge the support by National Science Foundation (DMR-2239598) and the use of facilities and instrumentation at the UC Irvine Materials Research Institute (IMRI) supported in part by National Science Foundation Materials Research Science and Engineering Center program through the U.C. Irvine Center for Complex and Active Materials (DMR-2011967).

## Author contributions

Y.J. and Z.C. conceived the idea and conceptualization. Y.J. contributed experiments, methodology, data curation, data analysis and investigation. D.J.L. performed FIB-SEM visualization and investigation. H.Z., K.H., and L.Z. contributed cryo-TEM characterization and investigation. S.S.B. and A.V.V. contributed computational calculations and proposed mechanisms. J.-P.L. contributed conceptualization, methodology, investigation and validation. J.W. performed XPS characterization. Y.-T.C., F.L., W.T., D.X., J.K. and M.-S.S. supported data analysis. Y.J. and Z.C. wrote the manuscript. Z.C. supervised the project. All authors discussed the results and commented on the paper.

## Competing interests

Z.C., Y.J., D.J.L., J.-P.L., J.K. and M.-S.S. declare that patents were filed for this work through UC San Diego's Office of Innovation and Commercialization and LG Energy Solution, Ltd. The remaining authors declare no competing interests.
