## [Peer Review File · Nature Communications]

Lithium Diffusion-Controlled Li-Al Alloy Negative Electrode for All-Solid-State Battery

Corresponding Author: Professor Zheng Chen

Version 0:

Reviewer comments:

Reviewer #1

(Remarks to the Author)

In this work, the authors demonstrated Li-Al alloy anode design to realize high performance solid-state lithium battery. The overall design concept and proposed mechanism of Li-Al alloy is reasonable and the electrochemical performance is satisfactory. I think this discovery promotes the understanding of the diffusion mechanism in alloy anode. However, there are still many areas worthy of enhancement and refinement for this work. A major revision is suggested before publishing on Nature Communications.

1. In the introduction section, the authors declared the necessity to develop aluminum-based alloy anode, but the advantage of aluminum compared with other alloy elements was not highlighted. Meanwhile, the as-prepared Li-Al anode was applied in solid-state battery, so it's also necessary to discuss the challenge of applying alloy anode in solid state battery.
2. For the GITT test in Figure 1c and 1d, the authors should specifically explain which kind of half cell (solid state cell or liquid cell) were adopted and its selection basis. Besides, I suggest the diffusion efficiency of lithium ion could be calculated and displayed in Figure 1d as well and compared that with the DFT calculation result to support the author's conclusion. And another minor suggestion is that the lithium atoms, aluminum atoms and hopping atoms could be distinguished by different colors (Figure 1b).
3. Further kinetic characterization of Al-Li anode should be conducted to determine the optimum Li-Al ratio, including the GITT of Al-Li/LPSC/Li half-cells with various ratios of Li and Al. It's better to conduct in-situ EIS and DRT analysis of Al-Li/LPSC/Li half-cells if possible.
4. The electrochemical performance of Al-Li/LPSC/Li half-cells with various ratios of Li and Al need to be comprehensively evaluate, including discharge-charge profile, rate performance and cyclic performance.
5. In the CCD test (Figure 4a), the authors should explain the reason for the large overpotential of Li_{0.25}Al₁.
6. The NCM/LPSC/Li cell and Li/LPSC/Li symmetric cell need to be assembled and tested as control group to highlight the improvement of as-designed Li-Al anode
7. From the Al-Li/LPSC/Al-Li symmetric cell, Li_{0.75}Al₁ seems to be a better experimental group with smaller impedance and change, but why the Li_{0.5}Al₁ was selected as optimum group for the following full-cell test? How did the author determine the optimum ratio of Li and Al?
8. In Figure 3, the lithiation behavior and process were assumed, which seems to be reasonable, but I still suggest to provide some experimental evidence if possible (such as TOF-SIMS, cryo-TEM or XRD to determine the inner Li-Al component).
9. For the LPSC at anode-side interface, the reduction decomposition to form SEI is a crucial aspect. Can the Li-Al make impact on the component of SEI compared with the LPSC/Li-derived SEI? XPS characterization on the LPSC/Al-Li interface need to be conducted.
10. The authors characterized the volume expansion of Li_{0.5}Al₁ anode and what about other Li-Al anodes? The morphology change of various Li-Al anodes should be observed by AFM and SEM.

Reviewer #2

(Remarks to the Author)

This paper presents the impact of prelithiation on the behavior of aluminum anodes in solid state batteries with sulfide electrolytes at high stack pressure. The findings are generally interesting and the battery performance is good.

The authors should carefully consider the clarity of the manuscript – the main point of the paper (that the prelithiation helps diffusivity and enables improved capacity and reversibility) is clear after reading the paper, but the first few figures seem to provide a lot of excess information that delay the reader from getting to this point.

Other comments which should be addressed:

- Delithiation and lithiation labels are switched on Fig 1d
- Fig 3 – the schematics seem to show concentration gradients between the beta LiAl phase and alpha. This should be a sharp phase boundary - please correct this
- The schematics and description of the governing mechanism in fig 3 are confusing. It is stated that “wide distribution of the internal β phase facilitates the Li⁺ move out...”. This is overly vague – why exactly does partial prelithiation help reversibility? Trapping should still be active based on the proposed mechanism, since the delithiation and alpha phase nucleation should start at the particle surface.
- Please specify stack pressure in figure captions, since stack pressure is arguably as important as current density in solid-state battery measurements.
- In reference to the point above, it should be made clear to the reader in a general-interest publication such as Nature Communications that such high stack pressures are commonly used for testing in small scale cells but are impractical for practical cells
- Fig 4b – which spectra are before and after?
- The impedance feature in Fig 4b is referred to as charge transfer resistance in the text. This seems like an oversimplification. Proper fitting (if appropriate) would be helpful
- The authors use the terminology CCD to refer to the full cell tests at different currents. This is inconsistent with the generally accepted use of this terminology, which is for Li/Li symmetric cells. It is suggested the authors use a more specific term to avoid confusion here.
- In reference to the point above, at high current densities, the rate capability of cathodes could limit the critical current density for Li-Al alloys. The reviewer suggests the critical-current-density tests in symmetrical cells for the Li-Al alloys.
- It is noted that data availability will be at reasonable request. It is suggested that the authors make all raw data available in the SI or as an included spreadsheet, as this is the trend in the field and is helpful for the community.
- The SEM images of the cross sections need to be described as FIB-prepared cross sections in the main text
- The Li-Al alloys were fabricated by mixing Al and Li powders and pressing the composite under 125 MPa. Figures 2 and 3 show that the Li-Al alloy particle has a Li concentration gradient from the surficial Li-rich layer to the inner pure Al core. The reviewer thinks that it is hard to fabricate Li-Al alloys with this structure by the direct pressing method, since the Al particle size is ~15 μm and Li particle size is ~80 μm (5 times larger, as shown in Supplementary Figure 3). Indeed, the pristine Li-Al anode in Fig. 5e shows Li-rich, Li-poor, and pristine Li powder regions, rather than the Li-rich surficial layer and pure Al core for each Li-Al particle. Thus, the mechanism proposed in Fig. 3 should be revised based on Fig. 5e.
- How were the Li-Al particles for cryo-TEM were prepared from the pressed Li-Al composite? The details should be clarified in Methods, because the Li-Al alloy (Fig. 5e) clearly has different compositional regions (Li-rich, Li-poor, and pristine Li powder).
- In line 145, “(Supplementary Fig. 4)” should be “(Supplementary Fig. 5)”.
- In line 327, Fig. 5b caption shows “voltage profiles at 1st, 100th, 200th, and 300th cycles”, and Fig. 5b legend shows voltage profiles at 1st, 100th, 200th, and 400th cycles, but Fig. 5b only has three sets of voltage curves.
- It looks that the initial coulombic efficiency of the cycling test in Fig. 5a-b is higher than 90%, which is contradictory to the results in Fig. 4e. This should be explained. And the voltage profiles for the cycling test in Fig. 5c should be included in supplementary information.
- In line 316, “After discharging, the β -LiAl layer was formed on the SE side far from the copper current collector”, but the BSE image in Fig. 5g does not show a clear darker layer of β -LiAl near the SSE or a clear brighter layer of α -Al near the copper current collector. Further evidence needs to be added to support the proposed mechanism in Fig. 5h.

Reviewer #3

(Remarks to the Author)

Reviewer #4

(Remarks to the Author)

This paper explores a Li-Al alloy anode and investigates variations in Li diffusivity depending on the Li content within the alloy. It highlights a high-performance all-solid-state battery that leverages the Li-rich β phase, known for its exceptional diffusivity. However, the purported advantages of superior diffusivity and the in situ pressurized prelithiation method discussed in the paper have already been addressed in several prior studies. Furthermore, several shortcomings in formatting and detailed analysis prevent us from endorsing the paper in its current form. A summary of the necessary revisions is provided below.

1) The superior conductivity of the Li-Al beta phase has been established in several previous studies(doi.org/10.1038/s41467-023-39685-x, doi.org/10.1038/s41467-024-48585-7, doi.org/10.1126/sciadv.abn4372). The authors need to clearly specify their novel findings, rather than merely restating known information.

- 2) The authors should discuss whether in-situ prelithiation using high-pressure conditions, as opposed to electrochemical charging, leads to the formation of a single Li-Al phase rather than a two-phase system. The paper should include a comparison of the phase differences between electrochemical charging of Al foils and physical Al-Li alloying under pressure.
- 3) The concept of in-situ prelithiation under pressure for all-solid-state battery fabrication has already been extensively applied, not only for Al (doi.org/10.1002/adma.202407128) but also for Si (doi.org/10.1038/s41467-024-47352-y), reducing the novelty of this approach.
- 4) Since GITT analysis is based on Fick's law, it is technically challenging to calculate the exact diffusion coefficient for a two-phase reaction. This limitation should be clearly stated in the paper, or alternative methods should be employed to obtain quantitative values.
- 5) In addition to the cross-sectional image of the Li_{0.5}Al electrode after the first charge-discharge cycle, additional cross-sectional analysis of the following additional samples is also required.
 - Pure Al foil after charge and discharge
 - Li_{0.5}Al anode and its interface after 100+ cycles
- 6) The following errors in the paper must be addressed, and a thorough review is necessary:
 - The overall readability of the figures, including Fig. 5b is compromised. The visual distinction of the lines should be enhanced by modifying the colors and line thicknesses for better reader comprehension.
 - In the case of Li half-cell, cycling begins with the lithiation of Li-Al alloy. An explanation is needed for how the delithiation capacity is higher than lithiation capacity in Fig. 1d, e.
 - In Fig. 5b, although four datasets are listed in the caption, only three are shown on the graph.
 - In Fig. 5c, while the cycle life extends to 2000 cycles, the number of plotted data points is insufficient. Please plot all 2000 data points accordingly.
 - Fig. 4b impedance graphs require clear internal labeling to identify each component, as done in Fig. 5d.
 - In Fig. 5c, the capacity continues to increase for approximately 100 cycles; an explanation for this prolonged initial capacity increase is needed.
 - Please cite and introduce recent research trends in alloy-based anode strategies in the introduction. doi.org/10.1021/acseenergylett.4c00915, doi.org/10.1002/aenm.202301508, doi.org/10.1002/advs.202301381, doi.org/10.1021/acscami.2c09013
 - The cell assembly used in Supplementary Figure 7 should be added to the caption, as indicated in Supplementary Figure 8.

Reviewer #5

(Remarks to the Author)

In this manuscript, the authors present a Li-Al alloy as an anode for sulfide-based all-solid-state batteries. This anode demonstrates a high critical current density, extended cycling life, and stable structural integrity during cycling. While the use of a Li-Al alloy as an anode is not a novel concept (references: *Sci. Adv.* 2022, 8, eabn4372; *J. Mater. Chem. A* 2022, 10, 12350; *Small* 2022, 18, e2204037; *Nat. Commun.* 2023, 14, 3975; *Adv. Mater.* 2024, 2407128), previous studies have also confirmed its compatibility with sulfide solid electrolytes. However, this report introduces distinctive features, particularly in its comprehensive analysis of the correlation between crystal phase and Li⁺ kinetics. After significant revision, this manuscript could be suitable for publication in *Nature Communications*.

1. In Fig. 2a, the XRD patterns show that the peak of the Li-Al alloy shifts compared to the standard reference, indicating lattice distortion in the prepared Li-Al alloy, suggesting that Li or Al atoms are integrating into the lattice. However, the authors state that "there is no intermediate Li-Al alloy compound due to the immiscibility of the two phases (Line 89)." The authors should clarify the interpretation of this peak shift.
2. In Fig. 2d, the fitted curve does not align well with the experimental data and should be re-fitted.
3. The Li⁺ diffusion coefficient should be calculated using the GITT method.
4. In Figs. 3b and 3c, after the dealloying of the alloyed Al/Li-Al (step ii), it is unclear why the α -phase Al core transitions to a β -phase LiAl core. The authors should provide additional characterization data to substantiate this observation.
5. The current density and areal capacity used in the cycling tests of symmetric cells are too low to adequately demonstrate the high-rate performance of Li-Al alloy anodes.
6. The authors should consider conducting a critical current density test in symmetric cells to determine the maximum current density the anode can tolerate. Evaluating this in a full battery would also be influenced by the dynamics of the positive electrode.
7. Regarding the NP ratio, there are several concerns: Is NP calculated based on the theoretical specific capacity of the Li-Al alloy? The theoretical specific capacity of the Li-Al alloy in Fig. S11 is based on Li₁Al₁, but Li₁Al₁ can be further lithiated to Li₃Al₂ and Li₉Al₄. The theoretical NP ratio may not be appropriate due to the different rate capabilities of the anode and cathode; that is, at the same current density, the delivered capacity/theoretical capacity ratio of the anode and cathode differs. Additionally, in Line 260, the statement "Decreasing the N/P ratio reduces the capacity of Li-poor α -phase Al" needs further clarification.
8. The cutoff voltage of the full cells is questionable, as the polarization of the charging curves sharply increases above 4.0 V. In previous Li-Al alloy-based studies, the cutoff voltage for full cells does not exceed 4.0 V.
9. In Fig. S13, the intermediate frequency region of the EIS spectra is attributed to the resistance of the cathode interface, but the authors have designated it as the anode interface resistance. This should be corrected.
10. How was the Li-Al electrode prepared? Was it prepared similarly to the Al electrode using the casting method? If not, why?
11. Were the cells tested under 75 MPa? This pressure is significantly higher than what has been reported in the literature.

The authors should justify this choice.

12. Other minor errors include:

Line 20: "Lithium" should be lowercase as "lithium".

Line 145: "Supplementary Fig. 4" should be corrected to "Supplementary Fig. 5".

The preparation process of the LCO cathode should be provided.

The voltage profiles of the high-loading full cell under different rates and cycles should also be included.

Version 1:

Reviewer comments:

Reviewer #1

(Remarks to the Author)

The revised manuscript has addressed all my concerns and is acceptable for publication.

Reviewer #2

(Remarks to the Author)

I have reviewed the revised manuscript and response letter and I believe the paper is acceptable for publication.

Reviewer #3

(Remarks to the Author)

Reviewer #4

(Remarks to the Author)

The authors have responded appropriately to the review, and I recommend publication in Nature Communications as is.

Reviewer #5

(Remarks to the Author)

This version can be accepted as it is.

We are grateful to the reviewers for their valuable comments and feedback to improve our manuscript. We believe that the comments and questions raised by the reviewers have helped us clarify our arguments and show the novelty of our work. We respond to each question raised by the reviewers in blue, and highlight the changes made to our manuscript in yellow.

Reviewer 1

In this work, the authors demonstrated Li-Al alloy anode design to realize high performance solid-state lithium battery. The overall design concept and proposed mechanism of Li-Al alloy is reasonable and the electrochemical performance is satisfactory. I think this discovery promotes the understanding of the diffusion mechanism in alloy anode. However, there are still many areas worthy of enhancement and refinement for this work. A major revision is suggested before publishing on Nature Communications.

Response.

The authors appreciate reviewer's thoughtful comments. We have carefully addressed each point in our responses below and have incorporated the corresponding discussions into the main text. For your reference, we've reordered the main figures, moving original Fig. 3 to Fig. 5 to make the proposed mechanism clearer. We believe that these revisions have significantly improved the quality of this work.

1. In the introduction section, the authors declared the necessity to develop aluminum-based alloy anode, but the advantage of aluminum compared with other alloy elements was not highlighted. Meanwhile, the as-prepared Li-Al anode was applied in solid-state battery, so it's also necessary to discuss the challenge of applying alloy anode in solid state battery.

Response.

The authors thank the reviewer for the detailed comments in the introduction. To highlight the necessity of aluminum (Al), the basic properties of Al have been compared to various other alloy-type materials in terms of electrochemical properties, volume change, diffusion coefficient, hardness, abundance of elements, and cost, as summarized in **Table S1**. Additionally, the large volume change of alloy materials was discussed through the introduction of recent challenging studies on the large-scale pouch cell operation under low stack pressure alongside previous strategies applied to alloy anodes in solid-state batteries.

The introduction section has been revised as follows:

(Main text, page 3)

“To improve the reversibility of Li anode, host materials have been adopted to enable stable Li storage and diffusion while mitigating Li dendritic growth. Among the host matrices with Li reaction mechanisms of intercalation (graphite and $\text{Li}_4\text{Ti}_5\text{O}_{12}$)⁵⁻⁸, conversion (SnS , LiAlH_6 and AgF)⁹⁻¹², and alloying (In , Si , Al , Sn and Mg)¹³⁻¹⁷, the alloy-type anode materials provide relatively higher Li storage capacity and diffusivity, which are crucial for increasing energy density of ASSBs⁴. In particular, their high Li^+ diffusivity enables an anode with high active material ratio without requiring SE to assist Li^+ transport. For example, a recent work on micron-silicon (μSi) demonstrated a high specific capacity (2890 mAh g^{-1}) with a high Li^+ diffusivity in the lithiated state ($\sim 10^{-9} \text{ cm}^2 \text{ s}^{-1}$) without adding SEs in the anode, leading to outstanding high-capacity performance at room temperature¹⁴. The prelithiation strategy was applied to alloy anodes to facilitate uniform ion diffusion for mitigating chemo-mechanical degradation and suppressing Li dendrite formation, using Li-Si and Li-Ag alloys, respectively.^{18, 19} To further achieve specific energy density (Wh kg^{-1}) to the ultimate limit, an anode-less design was developed with a thin Mg film as an effective seeding layer for reversible lithium plating and stripping, supported by a $\text{Ti}_3\text{C}_2\text{T}_x$ MXene mechanical buffer layer underneath.²⁰ For practical applications, however, recent studies at the forefront have been challenging the large volume changes in alloy anodes during Li (de)alloying reactions by improving chemo-mechanical stability within large-scale pouch cells operating under low stack pressure.^{21, 22}

Among prospective alloy-type materials, aluminum (Al) could be considered versatile, offering decent properties in terms of electrochemical performance and manufacturability (Table S1). Al stores Li up to 993 mAh g^{-1} (Li_3Al) via phase transformation from Li-poor $\alpha\text{-Al}$ to Li-rich $\beta\text{-LiAl}$, which is electrochemically achievable range at room temperature²³. The $\beta\text{-LiAl}$ can provide high Li^+ diffusivity of $\sim 10^{-7} \text{ cm}^2 \text{ s}^{-1}$ and a relatively small volume change of 96% (vs. 280% for Si)^{14, 24, 25}. In addition, the ductile property of Al (Vickers hardness (HV) is 35 for Al and 1130 for Si) is expected to improve interfacial contact with the SE layer as well as density of bulk electrode along with low porosity^{26, 27}. Furthermore, Al is the third most abundant element in the earth's crust (8.1%)²⁸ and is cost-effective ($\$2.45 \text{ kg}^{-1}$ as of Jun 2025), which further shows its potential as a practical anode material for widespread applications....”

Supplementary Table 1. Electrochemical, volume change, diffusion coefficient, hardness, abundance of elements and cost properties of various alloy materials.

	Specific capacity (mAh g ⁻¹)	Lithiated phase	Li diffusion coefficient (cm ² s ⁻¹)	Volume change (%)	Vickers hardness ^f (HV)	Elemental abundance in the earth's crust (%)	Cost (\$ kg ⁻¹)
Al	993 ^l	LiAl ^l	10 ⁻⁷ – 10 ⁻⁹ 2-4, a	96 ⁵	35 ⁶	8.1 ⁷	2.6 ⁸
Si	3579 ^l	Li ₁₅ Si ₄ ^l	10 ⁻⁹ – 10 ⁻¹² 8-10	280 ^l	1130 ^{ll}	29.5 ⁷	1.4 ^h
Mg	2150 ^l	Li _{1.95} Mg ^l	~10 ⁻⁷ 12, b	125 ^l	21 ^{l3}	1.9 ⁷	2.2 ^h
In	1012 ^{l4}	Li ₁₃ In ₃ ^{l4}	~10 ⁻⁶ 4, c	105 ^{l5, c}	10 ^{l6}	2.5×10 ⁻⁵ 7	349.5 ^h
Sn	993 ^l	Li _{4.4} Sn ^l	~10 ⁻⁷ 17, d	244 ^l	7 ⁶	2.5×10 ⁻⁴ 7	35.1 ⁸
Ag	670 ^l	Li _{2.7} Ag ^l	~10 ⁻⁸ 18, e	236 ^l	91 ^{l6}	0.7×10 ⁻⁵ 7	1098.1 ⁱ

^a Includes the Li diffusion coefficient calculated in this study

^b Measured for Li_xMg (1.34 ≤ x ≤ 3.65)

^c Measured for LiIn

^d Measured for Li_{2.33}Sn

^e Measured for Li_xAg (4.73 ≤ x ≤ 5.18)

^f Measured for pure materials

^g Element price as of March 2025 from London Metal Exchange

^h Element price as of March 2025 from Shanghai Metals market

ⁱ Element price as of March 2025 from Bloomberg

References

- [1] Obrovac, M. N., Chevrier, V. L. Alloy negative electrodes for Li-ion batteries. *Chem. Rev.* **114**, 11444-11502 (2014).
- [2] Jow, T. R., Liang, C. C. Lithium-aluminum electrodes at ambient temperatures. *J. Electrochem. Soc.* **129**, 1429 (1982).
- [3] Armstrong, R. D., Brown, O. R., Ram, R. P., Tuck, C. D. Lithium electrodes based upon aluminium and alloy substrates I. Impedance measurements on aluminium. *J. Power Sources* **28**, 259-267 (1989).

- [4] Tarczon, J. C., Halperin, W. P., Chen, S. C., Brittain, J.O. Vacancy antistructure defect interaction diffusion in beta-LiAl and beta-LiIn. *Mat. Sci. Eng. A-Struct.* **101**, 99-108 (1988).
- [5] Liu, Y., *et al.* Aluminum foil negative electrodes with multiphase microstructure for all-solid-state Li-ion batteries. *Nat. Commun.* **14**, 3975 (2023).
- [6] Li, H., *et al.* Circumventing huge volume strain in alloy anodes of lithium batteries. *Nat. Commun.* **11**, 1584 (2020).
- [7] Yaroshevsky, A. A. Abundances of chemical elements in the Earth's crust. *Geochem. Int.* **44**, 48-55 (2006).
- [8] Ding, N., *et al.* Determination of the diffusion coefficient of lithium ions in nano-Si. *Solid State Ion.* **180**, 222-225 (2009).
- [9] Pharr, M., Zhao, K., Wang, X., Suo, Z., Vlassak, J. J. Kinetics of initial lithiation of crystalline silicon electrodes of lithium-ion batteries. *Nano Lett.* **12**, 5039-5047 (2012).
- [10] Tan, D. H. S., *et al.* Carbon-free high-loading silicon anodes enabled by sulfide solid electrolytes. *Science* **373**, 1494-1499 (2021).
- [11] Walls, M. G., Chaudhri, M. M., Tang, T. B. STM profilometry of low-load vickers indentations in a silicon crystal. *J. Phys. D Appl. Phys.* **25**, 500-507 (1992).
- [12] Shi, Z., Liu, M., Naik, D., Gole, J. L. Electrochemical properties of Li-Mg alloy electrodes for lithium batteries. *J. Power Sources* **92**, 70-80 (2001).
- [13] Lu, Y., *et al.* Effects of minor gadolinium addition and T4 heat treatment on microstructure and properties of magnesium. *Adv. Eng. Mater.* **24**, 2200966 (2022).
- [14] Songster, J., Pelton, A. The In-Li (indium-lithium) system. *J. Phase Equilib.* **12**, 37-41 (1991).
- [15] Zhang, W., *et al.* (Electro)chemical expansion during cycling: monitoring the pressure changes in operating solid-state lithium batteries. *J. Mater. Chem. A* **5**, 9929-9936 (2017).
- [16] Negm, S. E., Mady, H., Bahgat, A. A. Influence of the addition of indium on the mechanical creep of Sn-3.5%Ag alloy. *J. Alloys Compd.* **503**, 65-70 (2010).
- [17] Wang, J., Raistrick, I. D., Huggins, R.A. Behavior of some binary lithium alloys as negative electrodes in organic solvent-based electrolytes. *J. Electrochem. Soc.* **133**, 457 (1986).
- [18] Jin, S., *et al.* Solid-solution-based metal alloy phase for highly reversible lithium metal anode. *J. Am. Chem. Soc.* **142**, 8818-8826 (2020).

2. For the GITT test in Figure 1c and 1d, the authors should specifically explain which kind of half cell (solid state cell or liquid cell) were adopted and its selection basis. Besides, I suggest the diffusion efficiency of lithium ion could be calculated and displayed in Figure 1d as well and compared that with the DFT calculation result to support the author's conclusion. And another minor suggestion is that the lithium atoms, aluminum atoms and hopping atoms could be distinguished by different colors (Figure 1b).

Response.

The authors appreciate the reviewer for detailed comments.

First, regarding cell configurations for the GITT test, a solid-state half cell (Li|LPSCI|Al), in which the solid electrolyte is excluded from the anode, was used to study the Li diffusion behavior in the pure Al only. To avoid confusion, the cell configuration has been added to the caption of **Fig. 1** and the corresponding experimental methods have been revised. Second, regarding the color of Li and Al atoms, the atom colors of the α and β phases in **Fig. 1b** and **c** have been modified to enhance readability. The revised **Fig. 1** and Methods section are as follows:

(Main text, page 6)

Fig. 1 | Li⁺ diffusion kinetics in Li-Al alloy. **a** Density functional theory (DFT)-based calculation on Li⁺ diffusivity in α and β phases. **b and c** Calculated migration energy of Al and Li. **b** α phase. **c** β phase. **d and e** Galvanostatic intermittent titration technique on lithiation and delithiation of pure Al in solid-state Li metal half cell (Li|LPSCI|Al). **d** Voltage profiles of galvanostatic pulses and open-circuited state. **e** Voltage polarization at each open-circuited state. Galvanostatic titration at 0.1 mA cm⁻² for 30 mins; Relaxation at open circuit for 2 hours; Stack pressure at 50 MPa; Room temperature.

(Main text, page 17)

Solid-state cell assembly

“The solid-state cells were assembled with two titanium plungers and one polyetheretherketone (PEEK) holder with diameter of 10 mm in the glove box.

Half cell. Li metal half cells with pure Al and Li-Al alloy anodes were used for GITT and *in situ* EIS analyses. For a pure Al half cell, A pure Al electrode disc (radius = 5 mm) was prepared as the working electrode, and a Li metal foil (radius = 5 mm) was used as the counter electrode. 70 mg of LPSCl was spread on a titanium plunger in a PEEK holder and pressed with the other plunger at 310 MPa for 10 s to first prepare a separator layer. The pure Al electrode disc was then placed on one side of the separator layer and pressed at 375 MPa for 3 min, followed by insertion of the Li metal foil on the opposite side. For half cells with Li-Al alloy anodes, 70 mg of LPSCl was spread and pressed between two titanium plungers at 375 MPa for 3 min. The Li-Al powder composite for Li_xAl_1 anode ($x = 0.25, 0.5$ and 0.75) was spread on one side of the separator layer and pressed at 125 MPa for 10 s, followed by insertion of the Li metal foil on the opposite side. The assembled half cells were rested under a stack pressure of 55 MPa for 6 h before testing.”

Third, for the calculation of diffusion coefficients from the GITT test, only single-phase reactions align with the theoretical conditions of GITT, which is based on Fick’s law, considering only diffusion. Therefore, materials with two-phase reaction behavior, such as aluminum, are not suitable for GITT analysis, since the additional factor of phase boundary propagation should be considered.¹ The limitation of GITT has been added to the main text, as follows:

(Main text, page 5)

“...This diffusion-dependent kinetics was also confirmed through the delithiation where the voltage polarization decreased back to 68% of Li after the vacancy formation, followed by rapid increase due to the development of α -dominant phase at the Al surface. It should be noted that GITT provides Li diffusivity at each lithiated state of a single-phase reaction material based on Fick’s law. In other words, a two-phase reaction material, such as Al, is not an ideal model for Li diffusivity calculations via GITT, which was therefore not applied in this study.^{42, 44}”

[1] Zhu, Y. J., Wang, C. S. Galvanostatic Intermittent Titration Technique for Phase-Transformation Electrodes. *Journal of Physical Chemistry C* **114**, 2830-2841 (2010).

3. Further kinetic characterization of Al-Li anode should be conducted to determine the optimum Li-Al ratio, including the GITT of Al-Li/LPSC/Li half-cells with various ratios of Li and Al. It's better to conduct in-situ EIS and DRT analysis of Al-Li/LPSC/Li half-cells if possible.

Response.

The authors appreciate the suggestion. Li⁺ diffusion kinetics of Li-Al alloy and pure Al anodes across the lithiated states were further investigated through GITT, *in-situ* EIS, and DRT characterizations in a Li half-cell configuration (**Supplementary Fig. 10**). During lithiation, the Li-Al alloy anodes at higher Li ratios showed lower voltage polarization and resistances, consistent with other kinetic characterizations in **Fig. 3**. The corresponding discussion and figures have been added as follows:

(Main text, page 9)

“The resistances were further specified to gain an in-depth understanding of Li⁺ diffusion by converting the impedance domain from frequency to relaxation time distribution (**Fig. 3c**). The relaxation time window could be compartmentalized by the resistances associated with Li⁺ transport through the SE grain boundary (R_{gb}) and SE interphase (R_{SEI}), (de)lithiation of the host materials (R_{ct}) and solid diffusion within the host matrix (R_{diff})^{47, 48}. Compared to the pristine case, the cycled Li_{0.5}Al₁ and Li_{0.75}Al₁ showed the averaged R_{ct} with significant decrease at 10⁻¹ – 1 s and increase in resistance at lower relaxation time and significantly suppressed R_{diff} (**Top and middle in Fig. 3c**). This is most possibly resulted from the homogenization of the β -dominant phase due to the Li⁺ interdiffusion between the separated α and β phases, and the particle size reduction by pulverization during cycling, which will be further discussed in the following section. On the contrary, a large increase of both R_{ct} and R_{diff} in the Li_{0.25}Al₁ case was confirmed, indicating dominant growth of α -phase, which prevents Li⁺ from moving out in the first delithiation step (**Bottom in Fig. 3c**). The GITT and *in-situ* EIS analysis across the lithiated state further confirmed the gradual change of Li⁺ diffusion kinetics in Li-Al alloy anodes compared to pure Al (**Supplementary Figure 10**). It was consistently observed that the Li-Al alloy anodes with predominant β phase, developed at higher Li ratio, exhibited lower voltage polarization and resistances during lithiation.”

Supplementary Figure 10. Analysis of GITT, *in situ* EIS, and DRT on lithiation of pure Al and Li-Al alloy anodes in Li metal half cell. a - c Pure Al. d - f $\text{Li}_{0.25}\text{Al}$. g - i $\text{Li}_{0.5}\text{Al}$. j - l $\text{Li}_{0.75}\text{Al}$. GITT was conducted with galvanostatic titration at 0.1 mA cm^{-2} for 2 hours, followed by relaxation at open circuit for 1 hour. EIS was measured immediately after each titration within a frequency range from 5 MHz to 0.5 MHz. The EIS corresponding to the red, orange, yellow and green circles were used for b, c, e, f, h, I, k and l, excluding the measurements in the unstable range of the normalized capacity below 0.1 and above 0.9. Stack pressure: 50 MPa. Room temperature.

4. The electrochemical performance of Al-Li/LPSC/Li half-cells with various ratios of Li and Al need to be comprehensively evaluate, including discharge-charge profile, rate performance and cyclic performance.

Response.

The authors appreciate the comment. Due to the stability limitation of lithium metal as a counter electrode, which has a critical current density (CCD) of 0.7 mA cm^{-2} (**Supplementary Fig. 5**), a symmetric cell configuration was utilized for additional rate performance test and cycling test at 1 mA cm^{-2} (**Supplementary Fig. 5 and 7**). $\text{Li}_{0.75}\text{Al}_1$ showed superior CCD and long-term cycling to $\text{Li}_{0.25}\text{Al}_1$ and $\text{Li}_{0.5}\text{Al}_1$, which has been added to the main text as follows:

(Main text, page 9)

“The Li^+ diffusion kinetics was investigated in symmetric cells employing Li-Al alloy electrodes with different Li to Al ratios of 0.25, 0.5, and 0.75 (**Fig. 3a-c**). Overpotentials were compared during delithiation and lithiation at 0.25 mA cm^{-2} to 0.25 mAh cm^{-2} (**Fig. 3a**). While the voltage of $\text{Li}_{0.25}\text{Al}_1$ already spiked dramatically prior to the first lithiation, Li-Al alloy electrodes with higher Li ratio experienced less overpotential (0.26 V for $\text{Li}_{0.5}\text{Al}_1$ and 0.23 V for $\text{Li}_{0.75}\text{Al}_1$). In the critical current density (CCD) tests, the Li-Al alloy electrodes exhibited consistent kinetic behavior, with $\text{Li}_{0.25}\text{Al}_1$ and $\text{Li}_{0.5}\text{Al}_1$ failing at a current density of 0.1 mA cm^{-2} and 1.6 mA cm^{-2} , respectively, whereas $\text{Li}_{0.75}\text{Al}_1$ can be operated up to 12 mA cm^{-2} (**Supplementary Fig. 5**). On the other hand, the Li symmetric cell short-circuited at 0.7 mA cm^{-2} resulting from Li dendrite growth, which was not observed in the Li-Al alloy cells.⁴⁶ The significant overpotential development in the $\text{Li}_{0.25}\text{Al}_1$ and $\text{Li}_{0.5}\text{Al}_1$ without short-circuiting implies that insufficient β -phase channels result in limited Li diffusion through the α -phase grown at the surface when the alloyed Li ratio is low in both the working and counter electrodes (**Supplementary Fig. 6**). Therefore, $\text{Li}_{0.75}\text{Al}_1$ demonstrated stable long-term cycling performance compared to $\text{Li}_{0.5}\text{Al}_1$ which failed after a few cycles with large overpotential during lithiation and delithiation at 1 mA cm^{-2} up to 0.5 mAh cm^{-2} (**Supplementary Fig. 7**)....”

Supplementary Figure 5. Critical-current-density test of symmetric cells with Li, $\text{Li}_{0.25}\text{Al}_1$, $\text{Li}_{0.5}\text{Al}_1$ and $\text{Li}_{0.75}\text{Al}_1$ electrodes. Lithiation and delithiation were conducted up to 0.25 mAh cm^{-2} at each current density. Stack pressure: 10 MPa for Li and 75 MPa for Li-Al alloy electrodes.

Supplementary Figure 7. Long-term cyclability of symmetric cells with $\text{Li}_{0.5}\text{Al}_1$ and $\text{Li}_{0.75}\text{Al}_1$ anodes. The inset shows magnified voltage profiles of the $\text{Li}_{0.5}\text{Al}_1$ symmetric cell during the first 5 hours. Lithiation and delithiation were conducted at 1 mA cm^{-2} up to 0.5 mAh cm^{-2} . Stack pressure: 75 MPa.

5. In the CCD test (Figure 4a), the authors should explain the reason for the large overpotential of $\text{Li}_{0.25}\text{Al}_1$.

Response.

The reviewer raised an insightful question. The large overpotential of $\text{Li}_{0.25}\text{Al}_1$, observed not only in the long-term cyclability (Fig. 3a) but also in the CCD test (Supplementary Fig. 5) as shown in the previous response, originated from the α -phase-dominant surface, preventing Li^+ from being delithiated. The corresponding discussion has been further clarified in the main text with Supplementary Fig. 6, and the related DRT characterization in Fig. 3c is highlighted again as follows:

(Main text, page 9)

“The significant overpotential development in the $\text{Li}_{0.25}\text{Al}_1$ and $\text{Li}_{0.5}\text{Al}_1$ without short-circuiting implies that insufficient β -phase channels result in limited Li diffusion through the α -phase grown at the surface when the alloyed Li ratio is low in both the working and counter electrodes (Supplementary Fig. 6).”

Supplementary Figure 6. Schematics of the phase development during cycling of symmetric cells with Li-Al alloy electrodes. a $\text{Li}_{0.25}\text{Al}_1$. b $\text{Li}_{0.5}\text{Al}_1$. c $\text{Li}_{0.75}\text{Al}_1$.

(Main text, page 9)

“On the contrary, a large increase of both R_{ct} and R_{diff} in the $\text{Li}_{0.25}\text{Al}_1$ case was confirmed, indicating dominant growth of α -phase, which prevents Li^+ from moving out in the first delithiation step (Bottom in Fig. 3c).”

Fig. 3b and c. Electrochemical impedance measurements on pristine and 200 h-cycled Li-Al electrodes. **b** Nyquist plot of impedance. **c** Distribution of relaxation time (DRT) spectra.

6. The NCM/LPSC/Li cell and Li/LPSC/Li symmetric cell need to be assembled and tested as control group to highlight the improvement of as-designed Li-Al anode.

Response.

The authors appreciate the comment. With the rate performance of the Li||Li symmetric cell in **Supplementary Fig. 5**, as already discussed in comment Q4, the rate performance of Li||NCM811 full cell was tested to highlight the advantages of using Li-Al alloy anodes and added as **Supplementary Fig. 12** as follows:

(Main text, page 10)

“The rate capability of the Al-based electrodes was then investigated in an NCM811-based full-cell configuration, where the performance is limited by the Al-based anode, in contrast to the symmetric cell system limited by both the working and counter electrodes (Fig. 3d-f). The Li-Al alloy anodes with higher Li ratio exhibited superiority in rate capability tests up to 7 mA cm^{-2} (Fig. 3d, Supplementary Fig. 11). However, the Li||NCM811 cell short-circuited at a current density of 1 mA cm^{-2} , since the CCD of Li is 0.7 mA cm^{-2} as confirmed in the symmetric cell tests (Supplementary Fig. 12).”

Supplementary Figure 12. Rate capability of Li||NCM811 full cell. Stack pressure: 10 MPa.

7. From the Al-Li/LPSC/Al-Li symmetric cell, Li_{0.75}Al₁ seems to be a better experimental group with smaller impedance and change, but why the Li_{0.5}Al₁ was selected as optimum group for the following full-cell test? How did the author determine the optimum ratio of Li and Al?

Response.

The authors thank the reviewer for the insightful question. Although Li_{0.75}Al₁ exhibited superior symmetric-cell performance to Li_{0.5}Al₁, the cell-level energy densities should be evaluated based on the specific capacity of Li-Al alloy anodes. Due to the additional Li used for prelithiating Al, Li_{0.5}Al₁ and Li_{0.75}Al₁ have specific capacities of 440 mAh g⁻¹ and 208 mAh g⁻¹, respectively, as shown in **Supplementary Fig. 17**. Given that the effective capacity of NCM811-based full cells was saturated at ~170 mAh g⁻¹ from the Li_{0.5}Al₁ anode, Li_{0.5}Al₁ achieved higher energy densities (755 Wh L⁻¹ and 282 Wh kg⁻¹) compared to Li_{0.75}Al₁ (642 Wh L⁻¹ and 217 Wh kg⁻¹). Therefore, Li_{0.5}Al₁ is determined to be the optimal anode design for the high-loading full-cell applications by considering both the diffusion channel of β-LiAl and active host of α-Al. We would like to emphasize this discussion presented in the main text.

(Main text, page 10)

“Through the systematic parameter study on the phase control, it can be concluded that the capacity utilization is highly dependent on the Li⁺ diffusion kinetics in the Li-Al alloy. In order to construct a practical cell-level design, the effective capacity of the Al-based full cells should thus be considered rather than only using the theoretical capacity of the active materials (Fig. 3f). Consequently, using the obtained effective full-cell capacities, the Li_{0.5}Al₁-based full-cell system was projected to provide optimal volumetric and gravimetric energy densities (755 Wh L⁻¹ and 282 Wh kg⁻¹) compared to other Al-based anodes (Fig. 3g, Supplementary Fig. 16 and Table S2).”

Fig. 3g. Volumetric and gravimetric energy density (Wh kg⁻¹) of ASSB with the configuration of pure Al and Li-Al alloy-based anodes and NCM811 cathode.

Supplementary Figure 16. Theoretical specific capacity of Li-Al alloy anode according to Li concentration, x .

8. In Figure 3, the lithiation behavior and process were assumed, which seems to be reasonable, but I still suggest to provide some experimental evidence if possible (such as TOF-SIMS, cryo-TEM or XRD to determine the inner Li-Al component).

Response.

The authors thank the reviewer for constructive suggestion. As the reviewer suggested, the phase change in crystalline structure and morphology was investigated, through XRD analysis of $\text{Li}_{0.5}\text{Al}_1$ and PFIB-SEM cross-sectional analysis of pure Al and $\text{Li}_{0.5}\text{Al}_1$ (**Supplementary Fig. 24** and **Fig. R1**).

In the XRD patterns of the $\text{Li}_{0.5}\text{Al}_1$ anode, the α phase was clearly observed to significantly decrease after lithiation and fully recover after delithiation (**Supplementary Fig. 24**). Additionally, the phase change within the inner part of pure Al and $\text{Li}_{0.5}\text{Al}_1$ was visualized after charge (lithiation) and discharge (delithiation) (**Fig. R1**). The $\text{Li}_{0.5}\text{Al}_1$ anode showed uniform distribution of β phase (dark gray) in the charged state, which was subsequently surrounded by thin α -phase layer (light gray) after discharge (**Fig. R1b**). In contrast, the charged Al still retained α -phase islands within the β phase, which, in this case, was subsequently enclosed by thick α phase layer after discharge (**Fig. R1a**). The corresponding discussion on the phase change through the additional XRD and cross-sectional characterization has been added to the main text as follows:

(Main text, page 12)

“In the BSE mode, the phase distribution across the electrode could be distinguished by contrast: light gray = α -Al; dark gray = β -LiAl; darkest gray = Li (**Right in Fig. 4e-g**). The pristine $\text{Li}_{0.5}\text{Al}_1$ electrode showed localized Li_1Al_1 even though the electrode was prepared by pressing a thoroughly mixed powder composite of Li and Al (dashed orange box on the **Right in Fig. 4e**). This is interpreted as the Li-Al alloying reaction spreading from the darkest spot where pristine Li powder was initially located. After the first charge, interestingly, the contrast of cross-section turned to be homogeneous with a few nanometer-thick α layers, representing uniform distribution of the β -LiAl and complete consumption of Li metal (**Right in Fig. 4f**). The charged $\text{Li}_{0.5}\text{Al}_1$ is regarded as a Li^+ conductor with highly developed Li^+ transport channels throughout the electrode. After discharging, the electrode consisted of pulverized particles surrounded by thin α -phase layer (**Right in Fig. 4g**). XRD characterization of the $\text{Li}_{0.5}\text{Al}_1$ anode clearly revealed this phase development in which the α phase significantly decreased after charging and fully recovered after discharging (**Supplementary Fig. 24**). More notably, the β -LiAl was observed to be relatively concentrated on the SE side far from the copper current collector. When Li^+ moves toward the SE during discharging, it could be inferred that the Li concentration gradient, generated at the interface between the Li-absent vacancy and the Li-present β -LiAl lattice, may possibly drive Li^+ diffusion toward the SE layer³⁴. Thus, highly developed β -LiAl phase in the Li-Al alloy anodes with a higher Li ratio could facilitate Li^+ diffusion to the SE side, serving as efficient Li^+ transport channels.”

Supplementary Figure 24. XRD pattern of $\text{Li}_{0.5}\text{Al}_1$ anode in pristine, lithiated and delithiated states. *: LPSCI residue on the surface of the delithiated $\text{Li}_{0.5}\text{Al}_1$ anode.

Fig. R1. Cross-sections of pure Al and $\text{Li}_{0.5}\text{Al}_1$ alloy anodes in the pristine, 1st charged, and 1st discharged states. a Pure Al. b $\text{Li}_{0.5}\text{Al}_1$. SE and BSE indicate secondary electron mode and backscattered electron mode, respectively. The area within the yellow box was enlarged for high-magnification images in BSE mode. c Schematics of discharged pure Al and $\text{Li}_{0.5}\text{Al}_1$ alloy anodes.

9. For the LPSC at anode-side interface, the reduction decomposition to form SEI is a crucial aspect. Can the Li-Al make impact on the component of SEI compared with the LPSC/Li-derived SEI? XPS characterization on the LPSC/Al-Li interface need to be conducted.

Response.

The reviewer pointed out an insightful concern. XPS characterization of the anode surface was conducted to investigate the decomposed products formed through the reactions with LPSCl (**Fig. R2**). The $\text{Li}_{0.5}\text{Al}_1$ and Li electrodes for XPS measurement were prepared by reacting with the LPSCl layer for 6 hours under a stack pressure of 75 MPa. Given the sufficient electronic conductivity of Li-Al alloy anodes to induce the chemical decomposition of LPSCl (**Supplementary Fig. 4**), no significant difference was observed in the SEI components. However, it should be noted that the major advantage of the alloy anode is the better-maintained interface compared to the Li metal anode, leading to the enhanced critical current density and cycling stability as demonstrated in **Supplementary Figure 5** and **7**.

Fig. R2. S 2p XPS. **a** $\text{Li}_{0.5}\text{Al}_1$ **b** Li.

Supplementary Figure 4. Electronic conductivity measurement of Al, $Li_{0.25}Al_1$, $Li_{0.5}Al_1$ and $Li_{0.75}Al_1$ anodes. Direct current (DC) polarization curve was observed at an applied voltage of 50 mV.

Supplementary Figure 5. Critical-current-density test of symmetric cells with Li, $Li_{0.25}Al_1$, $Li_{0.5}Al_1$ and $Li_{0.75}Al_1$ electrodes. Lithiation and delithiation were conducted up to $0.25\ mAh\ cm^{-2}$ at each current density. Stack pressure: 10 MPa for Li and 75 MPa for Li-Al alloy electrodes.

Supplementary Figure 7. Long-term cyclability of symmetric cells with $\text{Li}_{0.5}\text{Al}_1$ and $\text{Li}_{0.75}\text{Al}_1$ anodes. The inset shows magnified voltage profiles of the $\text{Li}_{0.5}\text{Al}_1$ symmetric cell during the first 5 hours. Lithiation and delithiation were conducted at 1 mA cm^{-2} up to 0.5 mAh cm^{-2} . Stack pressure: 75 MPa.

10. The authors characterized the volume expansion of $\text{Li}_{0.5}\text{Al}_1$ anode and what about other Li-Al anodes? The morphology change of various Li-Al anodes should be observed by AFM and SEM.

Response.

We thank the reviewer for this suggestion. It is crucial to fully understand not only the volume change but also the morphology change of various Li-Al.

The volume and morphology changes of pure Al, $\text{Li}_{0.25}\text{Al}_1$ and $\text{Li}_{0.75}\text{Al}_1$ anodes after the first charge and discharge have been further observed via cross-sectional characterization using PFIB-SEM, as shown in **Supplementary Fig. 21**. The electrode density was shown to improve with increasing the Li ratio, attributed to the enhanced particle pulverization (**Supplementary Fig. 22 and 25**). Furthermore, we investigated the 2000-cycled $\text{Li}_{0.5}\text{Al}_1$ anode to compare its pulverized morphology after long-term cycling. Interestingly, the highly dense structure composed of sub-micrometer-sized particles was visualized in **Supplementary Fig. 23**. The corresponding discussion has been added to the main text as follows:

(Main text, page 12)

“Therefore, close and intimate contact between the expanded Li-Al alloy particles was observed after charging, (**Middle in Fig. 4f**) which was further enhanced by the particles pulverized to a few micrometers in size during discharge (**Middle in Fig. 4g**). The pulverized particles, which could densify the electrode structure, would influence the improvement in interface and charge transfer resistances during the early cycles (**Fig. 4c**). Additional cross-sectional characterization of the pure Al and other Li-Al alloy anodes also confirmed that the electrode density could be improved by increasing the Li ratio, due to enhanced particle pulverization (**Supplementary Fig. 21**). Correspondingly, the porosity measured in the 1st discharged state was significantly reduced in the Li-Al alloy anodes compared to pure Al (**Supplementary Fig. 22**). Interestingly, even the 2000-cycled $\text{Li}_{0.5}\text{Al}_1$ anode maintained a highly dense structure by the pulverized particles continuing to reduce to sub-micrometer size during repetitive volume changes (**Supplementary Fig. 23**). ...”

(Main text, page 13)

“When Li^+ moves toward the SE during discharging, it could be inferred that the Li concentration gradient, generated at the interface between the Li-absent vacancy and the Li-present β -LiAl lattice, may possibly drive Li^+ diffusion toward the SE layer³⁴. Thus, highly developed β -LiAl phase in the Li-Al alloy anodes with a higher Li ratio could facilitate Li^+ diffusion to the SE side, serving as efficient Li^+ transport channels. The insight could be further supported by observing the cross-sections of discharged pure Al and other Li-Al alloy anodes in BSE mode (**Supplementary Fig. 21 and 25**). The $\text{Li}_{0.5}\text{Al}_1$ and $\text{Li}_{0.75}\text{Al}_1$ anodes with a higher Li ratio showed that pulverized particles composed of small β phase domains ($\text{Li}_{0.5}\text{Al}_1$: 2.326 μm ; $\text{Li}_{0.75}\text{Al}_1$: 1.607 μm) surrounded by thin α -phase layer, and the dark β -LiAl layer is mostly distributed at the SE-anode interface. Meanwhile, the light

gray α -Al layer was clearly observed at the interface of the pure Al and $\text{Li}_{0.25}\text{Al}_1$ anodes with β -phase island (pure Al: 4.877 μm ; $\text{Li}_{0.25}\text{Al}_1$: 2.783 μm) enclosed by thick α -phase layer.”

Supplementary Figure 21. Cross-sections of pure Al and Li-Al alloy anodes in the pristine, 1st charged, and 1st discharged states. **a** Pure Al. **b** $\text{Li}_{0.25}\text{Al}_1$. **c** $\text{Li}_{0.5}\text{Al}_1$. **d** $\text{Li}_{0.75}\text{Al}_1$. SE and BSE indicate secondary electron mode and backscattered electron mode, respectively. The area of yellow box was enlarged to high-magnification images at the rightmost row. **e** Schematics of the corresponding discharged Li-Al alloy anodes.

Supplementary Figure 22. Porosity of pure Al and Li-Al alloy anodes in the 1st discharged state. **a** Pure Al. **b** $\text{Li}_{0.25}\text{Al}_1$. **c** $\text{Li}_{0.5}\text{Al}_1$. **d** $\text{Li}_{0.75}\text{Al}_1$.

Supplementary Figure 23. Cross-sections of discharged $\text{Li}_{0.5}\text{Al}_1$ after 2000 cycles. **a** Secondary electron mode. **b** Backscattered electron mode. **c** High-magnification image of the yellow region boxed in b.

Supplementary Figure 25. β -phase domains distribution within the 1st discharged state. **a** Pure Al. **b** $\text{Li}_{0.25}\text{Al}_1$. **c** $\text{Li}_{0.5}\text{Al}_1$. **d** $\text{Li}_{0.75}\text{Al}_1$.

Reviewer 2

This paper presents the impact of prelithiation on the behavior of aluminum anodes in solid state batteries with sulfide electrolytes at high stack pressure. The findings are generally interesting and the battery performance is good. The authors should carefully consider the clarity of the manuscript – the main point of the paper (that the prelithiation helps diffusivity and enables improved capacity and reversibility) is clear after reading the paper, but the first few figures seem to provide a lot of excess information that delay the reader from getting to this point.

Response: The authors appreciate the reviewer’s thorough reviews. We have carefully addressed each point and supplemented additional characterizations for further clarification. Each point has been responded one by one below, and the corresponding discussions have been incorporated into the main text. In addition, for your reference, we’ve reordered the main figures, moving original Fig. 3 to Fig. 5 to make the proposed mechanism clearer, as noted in the corresponding responses. We believe that these revisions have significantly improved the quality of this work.

1. Delithiation and lithiation labels are switched on Fig 1d.

Response.

We thank the reviewer for pointing out this mistake. The labels in **Fig. 1d** and **e** have been revised as follows:

Fig. 1 | Li⁺ diffusion kinetics in Li-Al alloy. **a** Density functional theory (DFT)-based calculation on Li⁺ diffusivity in α and β phases. **b and c** Calculated migration energy of Al and Li. **b** α phase. **c** β phase. **d and e** Galvanostatic intermittent titration technique on lithiation and delithiation of pure Al in solid-state Li metal half cell (Li|LPSCI|Al). **d** Voltage profiles of galvanostatic pulses and open-circuited state. **e** Voltage polarization at each open-circuited state. Galvanostatic titration at 0.1 mA cm⁻² for 30 mins; Relaxation at open circuit for 2 hours; Stack pressure at 50 MPa; Room temperature.

2. Fig 3 – the schematics seem to show concentration gradients between the beta LiAl phase and alpha. This should be a sharp phase boundary - please correct this.

Response.

The authors appreciate this concern. We would like to kindly clarify that Li diffusion inhomogeneously proceeds through the Al grain boundary, leading to a gradient in local Li concentration. The cross-sectional visualization by HAADF-STEM demonstrated that strictly distinguishing the interphases between α and β is not straightforward (**Fig. R1**).^{1,2}

Fig. R1. **a** Cross-section HAADF-STEM image of $\text{Li}_{0.5}\text{Al}_1$ particles. **b** Enlarged area within the yellow-dotted box in Figure Scale bar: 1 μm for a and b.

[1] Zhang, L., *et al.* Electrochemical Grain Refinement Enables High-Performance Lithium–Aluminum-Anode-Based All-Solid-State Batteries. *ACS Energy Lett.* **10**, 898-906 (2025).

[2] Li, H., *et al.* Circumventing huge volume strain in alloy anodes of lithium batteries. *Nat. Commun.* **11**, 1584 (2020).

However, following the reviewer's feedback, we have minimized the Li concentration gradient in the corresponding figures (**Fig. 1** and **5**, **Supplementary Fig. 6** and **24**). For your reference, we also note that the original **Fig. 3** has been moved to **Fig. 5** with a detailed discussion is provided on the next page.

Fig. 1 | Li^+ diffusion kinetics in Li-Al alloy. **a** Density functional theory (DFT)-based calculation on Li^+ diffusivity in α and β phases. **b** and **c** Calculated migration energy of Al and Li. **b** α phase. **c** β phase. **d** and **e** Galvanostatic intermittent titration technique on lithiation and delithiation of pure Al in solid-state Li metal half cell (Li|LPSCI|Al). **d** Voltage profiles of galvanostatic pulses and open-circuited state. **e** Voltage polarization at each open-circuited state. Galvanostatic titration at 0.1 mA cm^{-2} for 30 mins; Relaxation at open circuit for 2 hours; Stack pressure at 50 MPa; Room temperature.

Fig. 5 | Phase development of Li-Al anodes in ASSB system. **a** ASSB configuration based on pure Al and Li-Al alloy anodes with NCM811-based cathode and LPSCI SSE layer. **b and c** Mechanism of Li^+ diffusion and phase development in pure Al and Li-Al alloy anode during Li alloying and dealloying. **b** Pure Al. **c** Li-Al alloy. (i) = first Li alloying; (ii) = first Li dealloying; (iii) = second Li alloying. **The schematics represent the average lithiation state of the electrode at each stage.**

Supplementary Figure 6. Schematics of the phase development during cycling of symmetric cells with Li-Al alloy electrodes. a $\text{Li}_{0.25}\text{Al}_1$. **b** $\text{Li}_{0.5}\text{Al}_1$. **c** $\text{Li}_{0.75}\text{Al}_1$.

Supplementary Figure 24. XRD pattern of $\text{Li}_{0.5}\text{Al}_1$ anode in pristine, lithiated and delithiated states. *:LPSCI residue on the surface of the delithiated $\text{Li}_{0.5}\text{Al}_1$ anode.

3. The schematics and description of the governing mechanism in fig 3 are confusing. It is stated that “wide distribution of the internal β phase facilitates the Li^+ move out...”. This is overly vague – why exactly does partial prelithiation help reversibility? Trapping should still be active based on the proposed mechanism, since the delithiation and alpha phase nucleation should start at the particle surface.

Response.

Prelithiation of the inner part of Al to the β phase reduces the depth of the α -phase growth at the particle surface, reducing the diffusion distance for Li^+ and enhancing the chemical potential difference between the phases. Cross-sectional characterization using FIB-SEM in BSE mode was conducted to visualize the phase development in the pure Al and $\text{Li}_{0.5}\text{Al}$ anodes after charge (lithiation) and discharge (delithiation) (**Fig. R2**). In the charged state, the $\text{Li}_{0.5}\text{Al}$ anode showed a relatively uniform distribution of β phase (dark gray), which was subsequently surrounded by sub-micrometer-thick α -phase layer (light gray) after discharge (**Fig. R2b**). In contrast, the charged Al retained α -phase islands within the β phase, which, in this case, was subsequently enclosed by a few-micrometer-thick α phase layer after discharge (**Fig. R2a**). While α -phase growth at the particle surface is inevitable, its depth can be controlled by the inner pre-developed β phase, which is pre-trapped but electrochemically reversible. In addition, to improve understanding and avoid the confusion, the discussion of the mechanism previously shown in **Fig. 3** has been moved to **Fig. 5**, following the discussion of the DFT studies and physicochemical, electrochemical and morphological characterizations on Li-Al alloy anodes. The corresponding sentence has also been revised for better clarity.

(Main text, page 15)

“Contrarily, the Li-Al alloy with pre-developed β phase is expected to facilitate the diffusion when accepting Li from the cathode (**Fig. 5c-i**). During dealloying, the Li^+ diffusion is facilitated through the reduced α phase distance with a higher driving force induced by the predominantly developed internal β phase.^{26, 35} Then, the reconnected β phase between the interior and the surface during the next alloying process serves as good conductive channels for the stored Li (**Fig. 5c-iii**).”

Fig. R2. Cross-sections of pure Al and $\text{Li}_{0.5}\text{Al}_1$ alloy anodes in the pristine, 1st charged, and 1st discharged states. **a** Pure Al. **b** $\text{Li}_{0.5}\text{Al}_1$. SE and BSE indicate secondary electron mode and backscattered electron mode, respectively. The area within the yellow box was enlarged for high-magnification images in BSE mode. **c** Schematics of discharged pure Al and $\text{Li}_{0.5}\text{Al}_1$ alloy anodes.

4. Please specify stack pressure in figure captions, since stack pressure is arguably as important as current density in solid-state battery measurements.

Response.

We thank the reviewer for pointing this out. The conditions of stack pressure and N/P ratios used have been added to the figure caption. The revised captions are as follows:

(Main text, page 11)

Fig. 3 | Electrochemical kinetics behaviors of Li-Al alloy electrodes. **a** Long-term cyclability in symmetric cell with $\text{Li}_{0.25}\text{Al}_1$, $\text{Li}_{0.5}\text{Al}_1$ and $\text{Li}_{0.75}\text{Al}_1$ anodes at 0.25 mA cm^{-2} to 0.25 mAh cm^{-2} . **b and c** Electrochemical impedance measurements on the pristine and 200 h-cycled Li-Al electrodes. **b** Nyquist plot of impedance. **c** Distribution of relaxation time (DRT) spectra. **d** Rate capability test of NCM811-based full cells with pure Al, $\text{Li}_{0.25}\text{Al}_1$, $\text{Li}_{0.5}\text{Al}_1$ and $\text{Li}_{0.75}\text{Al}_1$ anodes. **e** ICE of pure Al, $\text{Li}_{0.25}\text{Al}_1$, $\text{Li}_{0.5}\text{Al}_1$ and $\text{Li}_{0.75}\text{Al}_1$ anodes. **f** Specific capacities comparison at each current density. **g** Volumetric energy density (Wh L^{-1}) and specific energy (Wh kg^{-1}) of ASSB with the configuration of pure Al and Li-Al alloy-based anodes and NCM811 cathode. Stack pressure for the Li-Al symmetric cell and NCM811||Li-Al full-cell tests: 75 MPa. N/P ratio for full cells: 2.

(Main text, page 14)

Fig. 4 | Long-term cycling performance and morphological change of $\text{Li}_{0.5}\text{Al}_1$ anode with high-loading NCM811 cathode. **a-d** Long-term capacity retention test and voltage profiles at corresponding cycle. **a** Long-term cyclability at 0.33C ($1\text{C} = 5 \text{ mA cm}^{-2}$). **b** Voltage profiles at 1st, 100th, 200th and 300th cycle. **c** Long-term cyclability at 0.8C ($1\text{C} = 5 \text{ mA cm}^{-2}$). All the data points were selected at every 25 cycles. Stack pressure: 75 MPa. N/P ratio: 2. **d** EIS analysis at 100th and 500th cycle. **e-g** PFIB-prepared cross-sections of $\text{Li}_{0.5}\text{Al}_1$ anode. **e** Pristine. **f** After 1st charge. **g** After 1st discharge. The area of yellow box was enlarged to high-magnification images at the middle row. The first-row images in backscattered mode were shown at the rightmost row. **h** Schematics of $\text{Li}_{0.5}\text{Al}_1$ anode representing morphological change, Li^+ diffusion pathway and phase development and during charging and discharging.

(Supplementary information)

Supplementary Figure 4. Electronic conductivity measurement of Al, $\text{Li}_{0.25}\text{Al}_1$, $\text{Li}_{0.5}\text{Al}_1$ and $\text{Li}_{0.75}\text{Al}_1$ anodes. Direct current (DC) polarization curve was observed at an applied voltage of 50 mV and a stack pressure of 75 MPa.

Supplementary Figure 8. Electrochemical impedance spectra of 200 h-cycled $\text{Li}_{0.25}\text{Al}_1$ alloy anode at stack pressure of 75 MPa.

Supplementary Figure 11. Rate capability test for the kinetics improvement of Li_xAl_1 alloy electrodes along with prelithiation degree increase in NCM811-based full cells. **a** Specific capacity comparison among Li_xAl_1 alloy electrodes at each current density. **b** Pure Al. **c** $\text{Li}_{0.25}\text{Al}_1$. **d** $\text{Li}_{0.4}\text{Al}_1$. **e** $\text{Li}_{0.5}\text{Al}_1$. **f** $\text{Li}_{0.75}\text{Al}_1$. **g** $\text{Li}_{0.9}\text{Al}_1$. **h** $\text{Li}_{0.95}\text{Al}_1$. Stack pressure: 75 MPa. N/P ratio: 2.

Supplementary Figure 13. Rate capability test for the kinetics improvement of Li_xAl_1 alloy electrodes in LCO-based full cells. **a-d** Voltage profiles **a** Pure Al. **b** $\text{Li}_{0.25}\text{Al}_1$. **c** $\text{Li}_{0.5}\text{Al}_1$. **d** $\text{Li}_{0.75}\text{Al}_1$. **e** Specific capacity comparison at each current density. **f** Initial coulombic efficiency. Stack pressure: 75 MPa. N/P ratio: 2.

Supplementary Figure 14. Rate capability test for the kinetics improvement along with prelithiation degree and NP ratio. **a-c** $\text{Li}_{0.5}\text{Al}_1$ alloy anode at different NP ratios. **a** NP2. **b** NP1.5. **c** NP1.2. **d-f** $\text{Li}_{0.25}\text{Al}_1$ alloy anode at different NP ratios. **d** NP2. **e** NP1.5. **f** NP1.2. **g-i** Pure Al anode at different NP ratios. **g** NP2. **h** NP1.5. **i** NP1.2. Stack pressure: 75 MPa.

Supplementary Figure 15. Comparison of initial coulombic efficiency and specific capacities along with prelithiation degree and NP ratio. **a** Initial coulombic efficiency comparison among $\text{Li}_{0.5}\text{Al}_1$, $\text{Li}_{0.25}\text{Al}_1$ and pure Al anodes at NP ratio of 2, 1.5 and 1.2. **b-d** Specific capacity comparison among NP ratios of 2, 1.5 and 1.2. **b** $\text{Li}_{0.5}\text{Al}_1$. **c** $\text{Li}_{0.25}\text{Al}_1$. **d** Pure Al. Stack pressure: 75 MPa.

Supplementary Figure 17. Rate capability of $\text{Li}_{0.5}\text{Al}_1$ and pure Al with high-loading NCM811 cathode at room temperature ($1\text{C} = 5 \text{ mA cm}^{-2}$). **a** Areal capacities at various C-rates. **b and c** Voltage profiles at each C-rate. **b** $\text{Li}_{0.5}\text{Al}_1$. **c** Pure Al. Stack pressure: 75 MPa. N/P ratio: 2.

5. In reference to the point above, it should be made clear to the reader in a general-interest publication such as Nature Communications that such high stack pressures are commonly used for testing in small scale cells but are impractical for practical cells.

Response.

We appreciate the reviewer for the comment. It was noted that this study focused on the Li diffusion behavior in the Li-Al alloy anode, supported by a high stack pressure of 75 MPa, which was applied to the small plunger cells in symmetric and full-cell configuration to maintain intact particle contacts in both the cathode and Li-Al alloy anode. Corresponding discussion has been added to the main text as follows:

(Main text, page 10)

“... At higher current densities, a relatively higher proportion of Li ($x \geq 0.5$) in the Li-Al alloy anode led to higher specific capacity through further improved Li⁺ diffusion (**Fig. 3f**). It is noted that the stack pressure of 75 MPa was applied in both symmetric and full-cell tests to maintain intact particle contacts, particularly in the cathode composite as well as the Li-Al alloy anode, to solely investigate the β -phase-dependent Li diffusion mechanism.³⁵ Exploring advanced interface strategies will be key to enabling practical large-scale solid-state batteries with Li-Al alloy anodes operating under low stack pressure.”

6. Fig 4b – which spectra are before and after?

Response.

The authors appreciate the comment for improving the details. The spectra with the darker color correspond to measurements conducted after the cycling. **Fig. 3b** and **c** have been revised as follows:

Fig. 3 | Electrochemical kinetics behaviors of Li-Al alloy electrodes. a Long-term cyclability in symmetric cell with $\text{Li}_{0.25}\text{Al}_1$, $\text{Li}_{0.5}\text{Al}_1$ and $\text{Li}_{0.75}\text{Al}_1$ anodes at 0.25 mA cm^{-2} to 0.25 mAh cm^{-2} . **b and c** Electrochemical impedance measurements on the pristine and 200 h-cycled Li-Al electrodes. **b** Nyquist plot of impedance. **c** Distribution of relaxation time (DRT) spectra. **d** Rate capability test of NCM811-based full cells with pure Al, $\text{Li}_{0.25}\text{Al}_1$, $\text{Li}_{0.5}\text{Al}_1$ and $\text{Li}_{0.75}\text{Al}_1$ anodes. **e** ICE of pure Al, $\text{Li}_{0.25}\text{Al}_1$, $\text{Li}_{0.5}\text{Al}_1$ and $\text{Li}_{0.75}\text{Al}_1$ anodes. **f** Specific capacities comparison at each current density. **g** Volumetric energy density (Wh L^{-1}) and specific energy (Wh kg^{-1}) of ASSB with the configuration of pure Al and Li-Al alloy-based anodes and NCM811 cathode. Stack pressure for the Li-Al symmetric cell and NCM811||Li-Al full-cell tests: 75 MPa. N/P ratio for full cells: 2.

7. The impedance feature in Fig 4b is referred to as charge transfer resistance in the text. This seems like an oversimplification. Proper fitting (if appropriate) would be helpful.

Response.

Thank you for your detailed comments. All the EIS spectra were fitted, and the charge transfer resistances have been revised accordingly as follows:

(Main text, page 9)

“The changes in impedance elements over the cycles support what the voltage development represents for (Fig. 3b and c, Supplementary Fig. 8 and 9). The charge transfer resistance of pristine anodes (R_a) decreases significantly from 29.4 Ω for $\text{Li}_{0.25}\text{Al}_1$ to 12.6 Ω for both $\text{Li}_{0.5}\text{Al}_1$ and $\text{Li}_{0.75}\text{Al}_1$ On the other hand, Li-Al alloy electrodes with higher Li ratio alleviated the increase in charge transfer resistance after repeated cycles: 27.5 Ω for $\text{Li}_{0.5}\text{Al}_1$ vs. 17.5 Ω for $\text{Li}_{0.75}\text{Al}_1$.”

Supplementary Figure 9. a Equivalent circuit used for fitting EIS spectra in Fig. 4b. b Fitting results.

8. The authors use the terminology CCD to refer to the full cell tests at different currents. This is inconsistent with the generally accepted use of this terminology, which is for Li/Li symmetric cells. It is suggested the authors use a more specific term to avoid confusion here.

Response.

We thank the reviewer for correcting this. The term “critical current density (CCD)” in the main text and caption has been changed to “rate capability test” as follows:

(Main text, page 2)

Abstract

“... Consequently, a remarkably **high-rate capability** of 7 mA cm^{-2} was attained in LiNi_{0.8}Co_{0.1}Mn_{0.1} (NCM811)-based full-cell operation. ...”

(Main text, page 3)

Introduction

“...Based on this insight, the phase-controlled **Li_x≥0.5Al₁** alloy anodes demonstrated significant improvements in rate capability compared to the pure Al (7 mA cm^{-2} for Li-Al alloy vs. 4 mA cm^{-2} for pure Al). ...”

(Main text, page 10)

Electrochemical kinetics of Al-based electrodes

“The **rate capability** of the Al-based electrodes was then investigated in **an NCM811-based full-cell configuration, where the performance is limited by the Al-based anode, in contrast to the symmetric cell system limited by both the working and counter electrodes** (Fig. 3d-f). The Li-Al alloy anodes with higher Li ratio exhibited superiority in **rate capability** tests up to 7 mA cm^{-2} (Fig. 3d, **Supplementary Fig. 11**). ... The **rate capability** tests on various N/P ratios revealed that the kinetics of Li_{0.25}Al₁ anode and pure Al at N/P ratio of 1.2 were much superior to other N/P ratios of 2 and 1.5 as shown in the improvement of ICE and discharged specific capacities (**Supplementary Fig. 15a, c and d**)....”

(Main text, page 11)

Fig. 3 | Electrochemical kinetics behaviors of Li-Al alloy electrodes. **a** Long-term cyclability in symmetric cell with $\text{Li}_{0.25}\text{Al}_1$, $\text{Li}_{0.5}\text{Al}_1$ and $\text{Li}_{0.75}\text{Al}_1$ anodes at 0.25 mA cm^{-2} to 0.25 mAh cm^{-2} . **b and c** Electrochemical impedance measurements on the pristine and 200 h-cycled Li-Al electrodes. **b** Nyquist plot of impedance. **c** Distribution of relaxation time (DRT) spectra. **d** Rate capability test of NCM811-based full cells with pure Al, $\text{Li}_{0.25}\text{Al}_1$, $\text{Li}_{0.5}\text{Al}_1$ and $\text{Li}_{0.75}\text{Al}_1$ anodes. **e** ICE of pure Al, $\text{Li}_{0.25}\text{Al}_1$, $\text{Li}_{0.5}\text{Al}_1$ and $\text{Li}_{0.75}\text{Al}_1$ anodes. **f** Specific capacities comparison at each current density. **g** Volumetric energy density (Wh L^{-1}) and specific energy (Wh kg^{-1}) of ASSB with the configuration of pure Al and Li-Al alloy-based anodes and NCM811 cathode. Stack pressure for the Li-Al symmetric cell and NCM811||Li-Al full-cell tests: 75 MPa. N/P ratio for full cells: 2.

(Main text, page 16)

Conclusion

“... This improvement also led to enhanced ICE and higher capacities in the rate capability tests of full cells incorporated with NCM811 cathode. ...”

(Main text, page 18)

Methods - Electrochemical measurements

“... For the rate capability test, the current density was ramped up from 0.25 mA cm^{-2} to 7 mA cm^{-2} at the stack pressures from 75 MPa to 5 MPa with NP ratios of 1.2, 1.5 and 2. ...”

(Supplementary information)

Supplementary Figure 11. Rate capability test for the kinetics improvement of Li_xAl_1 alloy electrodes along with prelithiation degree increase in NCM811-based full cells. **a** Specific capacity comparison among Li_xAl_1 alloy electrodes at each current density. **b** Pure Al. **c** $\text{Li}_{0.25}\text{Al}_1$. **d** $\text{Li}_{0.4}\text{Al}_1$. **e** $\text{Li}_{0.5}\text{Al}_1$. **f** $\text{Li}_{0.75}\text{Al}_1$. **g** $\text{Li}_{0.9}\text{Al}_1$. **h** $\text{Li}_{0.95}\text{Al}_1$. Stack pressure: 75 MPa. N/P ratio: 2.

Supplementary Figure 13. Rate capability test for the kinetics improvement of Li_xAl_1 alloy electrodes in LCO-based full cells. **a-d** Voltage profiles **a** Pure Al. **b** $\text{Li}_{0.25}\text{Al}_1$. **c** $\text{Li}_{0.5}\text{Al}_1$. **d** $\text{Li}_{0.75}\text{Al}_1$. **e** Specific capacity comparison at each current density. **f** Initial coulombic efficiency. Stack pressure: 75 MPa. N/P ratio: 2.

Supplementary Figure 14. Rate capability test for the kinetics improvement along with prelithiation degree and NP ratio. a-c $\text{Li}_{0.5}\text{Al}_1$ alloy anode at different NP ratios. **a** NP2. **b** NP1.5. **c** NP1.2. **d-f** $\text{Li}_{0.25}\text{Al}_1$ alloy anode at different NP ratios. **d** NP2. **e** NP1.5. **f** NP1.2. **g-i** Pure Al anode at different NP ratios. **g** NP2. **h** NP1.5. **i** NP1.2. Stack pressure: 75 MPa.

9. In reference to the point above, at high current densities, the rate capability of cathodes could limit the critical current density for Li-Al alloys. The reviewer suggests the critical-current-density tests in symmetrical cells for the Li-Al alloys.

Response.

We thank the reviewer for the constructive suggestion. The critical current density test was evaluated using a symmetric cell configuration with the corresponding discussion added to the main text as follows:

(Main text, page 9)

“...In the critical current density (CCD) tests, the Li-Al alloy electrodes exhibited consistent kinetic behavior, with $\text{Li}_{0.25}\text{Al}_1$ and $\text{Li}_{0.5}\text{Al}_1$ failing at a current density of 0.1 mA cm^{-2} and 1.6 mA cm^{-2} , respectively, whereas $\text{Li}_{0.75}\text{Al}_1$ can be operated up to 12 mA cm^{-2} (Supplementary Fig. 5). On the other hand, the Li symmetric cell short-circuited at 0.7 mA cm^{-2} resulting from Li dendrite growth, which was not observed in the Li-Al alloy cells.⁴⁶...”

Supplementary Figure 5. Critical-current-density test of symmetric cells with Li, $\text{Li}_{0.25}\text{Al}_1$, $\text{Li}_{0.5}\text{Al}_1$ and $\text{Li}_{0.75}\text{Al}_1$ electrodes. Lithiation and delithiation were conducted up to 0.25 mAh cm^{-2} at each current density. Stack pressure: 10 MPa for Li and 75 MPa for Li-Al alloy electrodes.

10. It is noted that data availability will be at reasonable request. It is suggested that the authors make all raw data available in the SI or as an included spreadsheet, as this is the trend in the field and is helpful for the community.

Response.

The authors appreciate the comment. All the raw data used in this study have been provided as supplementary information in a spreadsheet.

11. The SEM images of the cross sections need to be described as FIB-prepared cross sections in the main text.

Response.

The authors appreciate the reviewer for pointing this out. The information on PFIB-SEM has been added to the main text and the caption of **Fig. 4** as follows:

(Main text, page 12)

“...The cross-sectional morphology and phase development of the $\text{Li}_{0.5}\text{Al}_1$ anode at pristine, charged, and discharged states were then visualized in secondary electron (SE) and backscattered electron (BSE) modes using plasma focused ion beam scanning electron microscopy (PFIB-SEM) (**Fig. 4e-h**)....”

(Main text, page 14)

Fig. 4 | Long-term cycling performance and morphological change of $\text{Li}_{0.5}\text{Al}_1$ anode with high-loading NCM811 cathode. **a-d** Long-term capacity retention test and voltage profiles at corresponding cycle. **a** Long-term cyclability at 0.33C ($1\text{C} = 5 \text{ mA cm}^{-2}$). **b** Voltage profiles at 1st, 100th, 200th and 300th cycle. **c** Long-term cyclability at 0.8C ($1\text{C} = 5 \text{ mA cm}^{-2}$). All the data points were selected at every 25 cycles. Stack pressure: 75 MPa. N/P ratio: 2. **d** EIS analysis at 100th and 500th cycle. **e-g** PFIB-prepared cross-sections of $\text{Li}_{0.5}\text{Al}_1$ anode. **e** Pristine. **f** After 1st charge. **g** After 1st discharge. The area of yellow box was enlarged to high-magnification images at the middle row. The first-row images in backscattered mode were shown at the rightmost row. **h** Schematics of $\text{Li}_{0.5}\text{Al}_1$ anode representing morphological change, Li^+ diffusion pathway and phase development and during charging and discharging.

12. The Li-Al alloys were fabricated by mixing Al and Li powders and pressing the composite under 125 MPa. Figures 2 and 3 show that the Li-Al alloy particle has a Li concentration gradient from the surficial Li-rich layer to the inner pure Al core. The reviewer thinks that it is hard to fabricate Li-Al alloys with this structure by the direct pressing method, since the Al particle size is $\sim 15 \mu\text{m}$ and Li particle size is $\sim 80 \mu\text{m}$ (5 times larger, as shown in Supplementary Figure 3). Indeed, the pristine Li-Al anode in Fig. 5e shows Li-rich, Li-poor, and pristine Li powder regions, rather than the Li-rich surficial layer and pure Al core for each Li-Al particle. Thus, the mechanism proposed in Fig. 3 should be revised based on Fig. 5e.

Response.

The authors appreciate the opportunity to refine the schematic details of the mechanism. It is well recognized that the phase distribution within the pristine electrodes is not ideally homogenous due to the 220-times more number of Al particles compared to Li (calculated based on the particle size from SEM image). As a result, the schematics in **Fig. 5** do not perfectly represent individual alloyed particle but illustrate the averaged lithiation states of the electrodes to conceptualize phase development based on different prelithiated Li ratios. To prevent any confusion, the corresponding statement has been added to the **Fig. 5** caption as follows:

(Main text, page 15)

Fig. 5 | Phase development of Li-Al anodes in ASSB system. **a** ASSB configuration based on pure Al and Li-Al alloy anodes with NCM811-based cathode and LPSCI SSE layer. **b and c** Mechanism of Li^+ diffusion and phase development in pure Al and Li-Al alloy anode during Li alloying and dealloying. **b** Pure Al. **c** Li-Al alloy. (i) = first Li alloying; (ii) = first Li dealloying; (iii) = second Li alloying. **The schematics represent the average lithiation state of the electrode at each stage.**

Additionally, to further support the proposed mechanism and study phase development across different Li ratios, the cross-sectional characterizations of pure Al, $\text{Li}_{0.25}\text{Al}_1$ and $\text{Li}_{0.75}\text{Al}_1$ anodes in pristine, charged, and discharged states have been supplemented with schematics of discharged electrodes, as shown in **Supplementary Fig. 21**. We would like to highlight that the Li powder regions in the pristine electrode were completely redistributed after the charging of all Li-Al alloy anodes. The changes have been made in the main text accordingly:

(Main text, page 12)

“...Therefore, close and intimate contact between the expanded Li-Al alloy particles was observed after charging, (**Middle in Fig. 4f**) which was further enhanced by the particles pulverized to a few micrometers in size during discharge (**Middle in Fig. 4g**). **The pulverized particles, which could densify the electrode structure, would influence the improvement in interface and charge transfer resistances during the early cycles (Fig. 4c).** **Additional cross-sectional characterization of the pure Al and other Li-Al alloy anodes also confirmed that the electrode density could be improved by increasing the Li ratio, due to enhanced particle pulverization (Supplementary Fig.**

21). Correspondingly, the porosity measured in the 1st discharged state was significantly reduced in the Li-Al alloy anodes compared to pure Al (Supplementary Fig. 22). Interestingly, even the 2000-cycled Li_{0.5}Al₁ anode maintained a highly dense structure by the pulverized particles continuing to reduce to sub-micrometer size during repetitive volume changes (Supplementary Fig. 23)....”

(Main text, page 12)

“...The charged Li_{0.5}Al₁ is regarded as a Li⁺ conductor with highly developed Li⁺ transport channels throughout the electrode. After discharging, the electrode consisted of pulverized particles surrounded by thin α -phase layer (Right in Fig. 4g). XRD characterization of the Li_{0.5}Al₁ anode clearly revealed this phase development in which the α phase significantly decreased after charging and fully recovered after discharging (Supplementary Fig. 24). More notably, the β -LiAl was observed to be relatively concentrated on the SE side far from the copper current collector. When Li⁺ moves toward the SE during discharging, it could be inferred that the Li concentration gradient, generated at the interface between the Li-absent vacancy and the Li-present β -LiAl lattice, may possibly drive Li⁺ diffusion toward the SE layer³⁴. Thus, highly developed β -LiAl phase in the Li-Al alloy anodes with a higher Li ratio could facilitate Li⁺ diffusion to the SE side, serving as efficient Li⁺ transport channels. The insight could be further supported by observing the cross-sections of discharged pure Al and other Li-Al alloy anodes in BSE mode (Supplementary Fig. 21 and 25). The Li_{0.5}Al₁ and Li_{0.75}Al₁ anodes with a higher Li ratio showed that pulverized particles composed of small β phase domains (Li_{0.5}Al₁: 2.326 μm ; Li_{0.75}Al₁: 1.607 μm) surrounded by thin α -phase layer, and the dark β -LiAl layer is mostly distributed at the SE-anode interface. Meanwhile, the light gray α -Al layer was clearly observed at the interface of the pure Al and Li_{0.25}Al₁ anodes with β -phase island (pure Al: 4.877 μm ; Li_{0.25}Al₁: 2.783 μm) enclosed by thick α -phase layer.”

Supplementary Figure 21. Cross-sections of pure Al and Li-Al alloy anodes in the pristine, 1st charged, and 1st discharged states. **a** Pure Al. **b** $\text{Li}_{0.25}\text{Al}_1$. **c** $\text{Li}_{0.5}\text{Al}_1$. **d** $\text{Li}_{0.75}\text{Al}_1$. SE and BSE indicate secondary electron mode and backscattered electron mode, respectively. The area of yellow box was enlarged to high-magnification images at the rightmost row. **e** Schematics of discharged Li-Al alloy anodes.

13. How were the Li-Al particles for cryo-TEM were prepared from the pressed Li-Al composite? The details should be clarified in Methods, because the Li-Al alloy (Fig. 5e) clearly has different compositional regions (Li-rich, Li-poor, and pristine Li powder).

Response.

The authors thank the reviewer for pointing this out. It should be noted that the main purpose of cryo-TEM characterization was to visualize the inner structure of Li-Al alloy and understand the chemical state of Al according to the lithiated state despite the presence of residual Li powder regions after electrode fabrication. Nevertheless, as discussed in the previous comment, the remaining Li in the pristine electrodes was fully consumed after charging. The preparation method for $\text{Li}_{0.5}\text{Al}_1$ particles was added as follows:

(Main text, page 17)

Electrodes preparation

“*Li-Al powder composite.* The Li and Al powder was mixed to prepare Li-Al composite at the atomic ratio of x to 1 in Li_xAl_1 ($x = 0.25, 0.4, 0.5, 0.75, 0.9$ and 0.95) for 10 min at 1900 rpm using a vortex mixer (MI0101002D, Four E’s Scientific). The pristine Li-Al alloy electrodes used for XRD, cryo-TEM and XPS characterizations were prepared by pressing the Li-Al powder composite at a corresponding Li atomic ratio under 125 MPa for 10 s, followed by 6 h resting at 75 MPa.”

(Main text, page 19)

Cryo-transmission electron microscopy (cryo-TEM) characterization

“The $\text{Li}_{0.5}\text{Al}_1$ particles for cryo-TEM characterization were prepared by hand grinding the pressed $\text{Li}_{0.5}\text{Al}_1$ pellet into particles using mortar and pestle. The ground $\text{Li}_{0.5}\text{Al}_1$ alloy particle clusters were directly dispersed onto a Cu grid and loaded to a Gatan 636 cooling holder for cryo-TEM analysis at liquid-nitrogen temperatures. All sample preparation and loading were conducted inside a glovebox. To minimize side reactions, the sample was sealed with an airtight cover prior to brief air exposure during insertion into the TEM column. TEM images and electron diffraction patterns were captured with a JEOL JEM-2100F TEM at 200 kV, equipped with a Gatan K3 direct electron camera and a Gatan OneView camera. To minimize electron beam damage to the Li-Al alloy crystal structure, the electron dose was kept below $150 \text{ e}^-/\text{\AA}^2 \text{ s}$. High-resolution transmission electron microscopy (HRTEM) images along with fast Fourier transform (FFT) pattern and FFT-filtering were analyzed using Gatan Microscopy Suite software. For further elemental analysis, Li-Al alloy particles were thinned using a Tescan GAIA3 SEM-FIB on a Cu FIB grid. The characterization of electron-dispersive X-ray spectroscopy (EDS) or electron energy-loss spectroscopy (EELS) in scanning transmission electron microscopy (STEM) mode was conducted using a JEOL JEM ARM300F Grand ARM TEM operated at 300 kV and equipped with aberration correctors, Gatan K2 Summit camera, and dual 100 mm^2 Si drift detectors. A probe convergence semi-angle of 25.7 mrad , a spectrometer collection semi-angle of approximately 35 mrad , and an energy dispersion of 0.05 eV

per channel were used for acquiring high-angle annular dark-field (HAADF)-STEM images, Al and O EDS elemental mapping, and Li K-edge and Al L-edge EELS spectra. All TEM and STEM imaging as well as EDS and EELS analyses were conducted at cryogenic conditions.”

14. In line 145, “(Supplementary Fig. 4)” should be “(Supplementary Fig. 5)”..

Response.

We thank the reviewer for this comment. Due to the reordering of figures during the revision process, the original **Supplementary Fig. 5** has been changed to **Supplementary Fig. 4** as follows:

(Main text, page 9)

Electrochemical kinetics of Al-based electrodes

“...The enhancement of the charge transfer before the reaction is largely related to the electronically conductive environment, which is consistent with the previous results showing that $\text{Li}_{0.5}\text{Al}_1$ and $\text{Li}_{0.75}\text{Al}_1$ provide improved electronic conductivity than $\text{Li}_{0.25}\text{Al}_1$ (**Supplementary Fig. 4**)....”

15. In line 327, Fig. 5b caption shows “voltage profiles at 1st, 100th, 200th, and 300th cycles”, and Fig. 5b legend shows voltage profiles at 1st, 100th, 200th, and 400th cycles, but Fig. 5b only has three sets of voltage curves.

Response.

We apologize for this inconsistency, and we have modified the figures and caption. The revised **Fig. 5b** and its corresponding caption are as follows:

(Main text, page 14)

Fig. 4 | Long-term cycling performance and morphological change of $\text{Li}_{0.5}\text{Al}_1$ anode with high-loading NCM811 cathode. **a-d** Long-term capacity retention test and voltage profiles at corresponding cycle. **a** Long-term cyclability at 0.33C (1C = 5 mA cm⁻²). **b** Voltage profiles at 1st, 100th, 200th and 300th cycle. **c** Long-term cyclability at 0.8C (1C = 5 mA cm⁻²). All the data points were selected at every 25 cycles. Stack pressure: 75 MPa. N/P ratio: 2. **d** EIS analysis at 100th and 500th cycle. **e-g** PFIB-prepared cross-sections of $\text{Li}_{0.5}\text{Al}_1$ anode. **e** Pristine. **f** After 1st charge. **g** After 1st discharge. The area of yellow box was enlarged to high-magnification images at the middle row. The first-row images in backscattered mode were shown at the rightmost row. **h** Schematics of $\text{Li}_{0.5}\text{Al}_1$ anode representing morphological change, Li^+ diffusion pathway and phase development during charging and discharging.

16. It looks that the initial coulombic efficiency of the cycling test in Fig. 5a-b is higher than 90%, which is contradictory to the results in Fig. 4e. This should be explained. And the voltage profiles for the cycling test in Fig. 5c should be included in supplementary information.

Response.

The reviewer raised an insightful concern. The high-loading NCM811-based full cells were precycled by gradually ramping up the C-rate to the long-term cycling rate in the sequence of 0.05C, 0.1C, 0.2C, 0.4C, and 0.6C.

The initial coulombic efficiency of the full cells for long-term cycling at 0.33C (Fig. 4a) and 0.8C (Fig. 4c) is 79.5% and 79.4%, respectively, which is consistent with the ICEs in Fig. 4e (**Fig. R3**). The methods for precycling have been added to the experimental section, and the voltage profiles for long-term cycling at 0.8C (Fig. 4c) have also been included in the supplementary information as follows:

(Main text, page 18)

Electrochemical measurements

“...Used conditions of the stack pressure and NP ratio were noted in corresponding figures. The high-loading NCM811-based full cells were precycled by gradually ramping up the C-rate to the long-term cycling rate in the order of 0.05C, 0.1C, 0.2C, 0.4C, and 0.6C...”

Fig. R3. 1st precycle of $\text{Li}_{0.5}\text{Al}_1$ with high-loading NCM811 cathode at room temperature ($1\text{C} = 5 \text{ mA cm}^{-2}$). 1st voltage profile at 0.05C of precycling prior to the main long-term cycling at a 0.33C and b 0.8C. Stack pressure: 75 MPa. N/P ratio: 2.

17. In line 316, “After discharging, the β -LiAl layer was formed on the SE side far from the copper current collector”, but the BSE image in Fig. 5g does not show a clear darker layer of β -LiAl near the SSE or a clear brighter layer of α -Al near the copper current collector. Further evidence needs to be added to support the proposed mechanism in Fig. 5h.

Response.

We thank the reviewer for pointing this out. The contrast and brightness of the cross-sectional images were adjusted to enhance the phase differentiation. Additionally, to further support the mechanism, the cross-sections of pure Al, $\text{Li}_{0.25}\text{Al}_1$ and $\text{Li}_{0.75}\text{Al}_1$ anodes after the first charge and discharge were observed using PFIB-SEM. The $\text{Li}_{0.5}\text{Al}_1$ and $\text{Li}_{0.75}\text{Al}_1$ anodes with a higher Li ratio showed that the dark β -LiAl layer is distributed at the SE-anode interface, while the light gray α -Al layer is clearly observed at the interface of the pure Al and $\text{Li}_{0.25}\text{Al}_1$ anodes. The corresponding discussion has been added to the main text as follows:

(Main text, page 12)

“...The charged $\text{Li}_{0.5}\text{Al}_1$ is regarded as a Li^+ conductor with highly developed Li^+ transport channels throughout the electrode. After discharging, the electrode consisted of pulverized particles surrounded by thin α -phase layer (Right in Fig. 4g). XRD characterization of the $\text{Li}_{0.5}\text{Al}_1$ anode clearly revealed this phase development in which the α phase significantly decreased after charging and fully recovered after discharging (Supplementary Fig. 24). More notably, the β -LiAl was observed to be relatively concentrated on the SE side far from the copper current collector. When Li^+ moves toward the SE during discharging, it could be inferred that the Li concentration gradient, generated at the interface between the Li-absent vacancy and the Li-present β -LiAl lattice, may possibly drive Li^+ diffusion toward the SE layer³⁴. Thus, highly developed β -LiAl phase in the Li-Al alloy anodes with a higher Li ratio could facilitate Li^+ diffusion to the SE side, serving as efficient Li^+ transport channels. The insight could be further supported by observing the cross-sections of discharged pure Al and other Li-Al alloy anodes in BSE mode (Supplementary Fig. 21 and 25). The $\text{Li}_{0.5}\text{Al}_1$ and $\text{Li}_{0.75}\text{Al}_1$ anodes with a higher Li ratio showed that pulverized particles composed of small β phase domains ($\text{Li}_{0.5}\text{Al}_1$: 2.326 μm ; $\text{Li}_{0.75}\text{Al}_1$: 1.607 μm) surrounded by thin α -phase layer, and the dark β -LiAl layer is mostly distributed at the SE-anode interface. Meanwhile, the light gray α -Al layer was clearly observed at the interface of the pure Al and $\text{Li}_{0.25}\text{Al}_1$ anodes with β -phase island (pure Al: 4.877 μm ; $\text{Li}_{0.25}\text{Al}_1$: 2.783 μm) enclosed by thick α -phase layer.”

Supplementary Figure 21. Cross-sections of pure Al and Li-Al alloy anodes in the pristine, 1st charged, and 1st discharged states. **a** Pure Al. **b** $\text{Li}_{0.25}\text{Al}_1$. **c** $\text{Li}_{0.5}\text{Al}_1$. **c** $\text{Li}_{0.5}\text{Al}_1$. **d** $\text{Li}_{0.75}\text{Al}_1$. SE and BSE indicate secondary electron mode and backscattered electron mode, respectively. The area of yellow box was enlarged to high-magnification images at the rightmost row. **e** Schematics of discharged Li-Al alloy anodes.

Reviewer 4

This paper explores a Li-Al alloy anode and investigates variations in Li diffusivity depending on the Li content within the alloy. It highlights a high-performance all-solid-state battery that leverages the Li-rich β phase, known for its exceptional diffusivity. However, the purported advantages of superior diffusivity and the *in situ* pressurized prelithiation method discussed in the paper have already been addressed in several prior studies. Furthermore, several shortcomings in formatting and detailed analysis prevent us from endorsing the paper in its current form. A summary of the necessary revisions is provided below.

Response: The authors appreciate the reviewer's insightful comments. We have carefully addressed each point and supplemented additional characterizations to reinforce the novelty of this work and demonstrate the consistency of the anodes under various conditions. Each point has been responded one by one below, and the corresponding discussions have been incorporated into the main text. For your reference, we've reordered the main figures, moving original Fig. 3 to Fig. 5 to make the proposed mechanism clearer, as noted in the corresponding responses. We believe that these revisions have significantly improved the quality of this work.

1. The superior conductivity of the Li-Al beta phase has been established in several previous studies (doi.org/10.1038/s41467-023-39685-x, doi.org/10.1038/s41467-024-48585-7, doi.org/10.1126/sciadv.abn4372). The authors need to clearly specify their novel findings, rather than merely restating known information.

Response.

The authors appreciate the comment on improving the introduction section by specifying our findings. The novel point of this study is 'the introduction of the correlation between β -LiAl phase development and Li^+ diffusion kinetics in Li-Al alloy anodes at various Li ratios'. We have placed an emphasis on studying how different proportions of α and β phases influence the Li-Al alloy anodes in terms of rate performance and electrode-level phase changes, which were characterized with comprehensive electrochemical analyses and PFIB-SEM. To the authors' knowledge, such systematic study on phase development at various Li ratios is unprecedented, including the studies referenced by the reviewers and other recent publications. Nevertheless, we fully agree that further specification of our findings in this study is necessary in the introduction section, and we have significantly supplemented and emphasized the cross-sectional characterization of Li-Al alloy anodes in the main text, which is specifically discussed in Q5. The introduction section has been revised as follows:

(Main text, page 4)

"Herein, we present a comprehensive study of the correlation between β -LiAl phase distribution and Li^+ kinetics in the Li-Al alloy by controlling the Li ratio via *in situ* prelithiation during cell assembly. We found that the reversibility of the Li-Al alloy anode improved significantly at Li ratios higher than 0.5 through predominantly developed β -phase structure. The underlying mechanism of Li^+ diffusion within the Li-poor α and Li-rich β phases

was investigated using density functional theory (DFT), revealing ten orders of magnitude difference in Li^+ diffusivity between the two phases. Based on this insight, the phase-controlled $\text{Li}_{x \geq 0.5}\text{Al}_1$ alloy anodes demonstrated significant improvements in rate capability compared to the pure Al (7 mA cm^{-2} for Li-Al alloy vs. 4 mA cm^{-2} for pure Al). The improved charge/discharge reversibility of $\text{Li}_{0.5}\text{Al}_1$ alloy anode extended the cycle life to 2000 cycles with capacity retention of 83% in $\text{LiNi}_{0.8}\text{Co}_{0.1}\text{Mn}_{0.1}$ (NCM811)-based solid-state cells at high areal capacity loading of 5 mAh cm^{-2} . Cross-sectional characterization corroborated the improved reversibility by visualizing distinct changes in phase development according to the Li ratios, in which the β -phase was predominantly developed in the inner core of the Al particles with a thinner α -phase surface layer at higher Li ratios. In addition, a dense and intact structure of Li-Al alloy anode independent of volume change after Li alloying and dealloying was also confirmed. Consequently, this study is believed to provide an in-depth understanding of the phase-dependent Li^+ diffusion behavior for future alloy anode design.”

2. The authors should discuss whether in-situ prelithiation using high-pressure conditions, as opposed to electrochemical charging, leads to the formation of a single Li-Al phase rather than a two-phase system. The paper should include a comparison of the phase differences between electrochemical charging of Al foils and physical Al-Li alloying under pressure.

Response.

The authors appreciate the comment. We would like to kindly clarify that Li-Al alloys have no intermediate single Li-Al alloy phase due to the immiscibility of α and β phases within the Li concentration from 0 (α -Al) to 50 at. % (β -LiAl) based on the phase diagram (**Supplementary Fig. 1**). Consequently, the Li-Al alloy anodes used in this study ($\text{Li}_{0.25}\text{Al}_1$, $\text{Li}_{0.5}\text{Al}_1$ and $\text{Li}_{0.75}\text{Al}_1$) cannot achieve single-phase system.

Supplementary Figure 1. Binary Li-Al phase diagram.

Nevertheless, the proportion of predeveloped β phase is enhanced with the prelithiation Li ratio of Li-Al alloy anodes, and the effect can be compared by investigating the electrochemical kinetics relative to electrochemically charged Al. To evaluate this effect, *in situ* EIS combined with GITT characterizations was conducted on pure Al to compare the EIS of pristine Li-Al alloy anodes prepared through *in situ* prelithiation (**Fig. R1**). The pure Al was observed to maintain its resistances in the lithiated state from 24% to 78%. However, the pristine Li-Al alloy anodes prepared by pressure-derived prelithiation showed significant decrease in resistance as the prelithiation ratio increased from $\text{Li}_{0.25}\text{Al}_1$ to $\text{Li}_{0.75}\text{Al}_1$, representing a relatively higher development of β -LiAl at higher Li ratios.

Fig. R1. Comparison of resistance between pure Al and pristine Li-Al alloy anodes at corresponding lithiated states. **a** GITT analysis of pure Al with voltage polarization (V_p) across lithiation states. **b** *In situ* EIS measurements of pure Al at 24%, 48% and 78% lithiation. **c** EIS measurements of pristine Li_{0.25}Al₁, Li_{0.5}Al₁ and Li_{0.75}Al₁ alloy anodes. **d** Total resistance extracted from the EIS semicircles in Fig. R1b and c. GITT was conducted with galvanostatic titration at 0.1 mA cm⁻² for 2 hours, followed by relaxation at open circuit for 1 hour. EIS was immediately measured after each titration within a frequency range from 5 MHz to 0.5 MHz. Stack pressure: 50 MPa. Room temperature.

Furthermore, the effect of predeveloped β phase on the Li⁺ diffusion kinetics in all Li-Al alloy anodes was fully compared to pure Al across the lithiated states via GITT analysis with *in-situ* EIS and DRT characterizations (**Supplementary Fig. 10**). During lithiation, the Li-Al alloy anodes with higher Li ratios showed lower voltage polarization and resistances compared to pure Al, representing facilitated Li⁺ transport through the more developed β phase. The corresponding discussion and figures have been added as follows:

(Main text, page 9)

Electrochemical kinetics of Al-based electrodes

“...On the contrary, a large increase of both R_{ct} and R_{diff} in the Li_{0.25}Al₁ case was confirmed, indicating dominant growth of α -phase, which prevents Li⁺ from moving out in the first delithiation step (**Bottom in Fig. 3c**). The GITT and *in-situ* EIS analysis across the lithiated state further confirmed the gradual change of Li⁺ diffusion kinetics in Li-Al alloy anodes compared to pure Al (**Supplementary Figure 10**). It was consistently observed that the Li-Al alloy anodes with predominant β phase, developed at higher Li ratio, exhibited lower voltage polarization and resistances during lithiation.”

Supplementary Figure 10. Analysis of GITT, *in situ* EIS, and DRT on lithiation of pure Al and Li-Al alloy anodes. a - c Pure Al. d - f $\text{Li}_{0.25}\text{Al}_1$. g - i $\text{Li}_{0.5}\text{Al}_1$. j - l $\text{Li}_{0.75}\text{Al}_1$. GITT was conducted with galvanostatic titration at 0.1 mA cm^{-2} for 2 hours, followed by relaxation at open circuit for 1 hour. EIS was measured immediately after each titration within a frequency range from 5 MHz to 0.5 MHz. The EIS corresponding to the red, orange, yellow and green circles were used for b, c, e, f, h, I, k and l, excluding the measurements in the unstable range of the normalized capacity below 0.1 and above 0.9. Stack pressure: 50 MPa. Room temperature.

3. The concept of in-situ prelithiation under pressure for all-solid-state battery fabrication has already been extensively applied, not only for Al (doi.org/10.1002/adma.202407128) but also for Si (doi.org/10.1038/s41467-024-47352-y), reducing the novelty of this approach.

Response.

The authors agree that the *in-situ* prelithiation strategy has been widely used for solid-state batteries in recent years. But we would like to emphasize that the novelty of this study is not the *in-situ* prelithiation itself but ‘the introduction of the correlation between β -LiAl phase development and Li⁺ diffusion kinetics in Li-Al alloy anodes at various Li ratios’, as discussed in the first response above. The reason for employing *in-situ* prelithiation of Li and Al powder is that it was considered the most efficient method to fabricate the various Li-Al alloy anodes in terms of processibility and reaction homogeneity.

4. Since GITT analysis is based on Fick's law, it is technically challenging to calculate the exact diffusion coefficient for a two-phase reaction. This limitation should be clearly stated in the paper, or alternative methods should be employed to obtain quantitative values.

Response.

We thank the reviewer for pointing this out. Because of this limitation of GITT method, we did not calculate the exact diffusion coefficient of Li-Al system. To clearly inform the reader, the limitation of GITT has been added to the main text, as follows:

(Main text, page 5)

“...This diffusion-dependent kinetics was also confirmed through the delithiation where the voltage polarization decreased back to 68% of Li after the vacancy formation, followed by rapid increase due to the development of α -dominant phase at the Al surface. It should be noted that GITT provides Li diffusivity at each lithiated state of a single-phase reaction material based on Fick's law. In other words, a two-phase reaction material, such as Al, is not an ideal model for Li diffusivity calculations via GITT, which was therefore not applied in this study.^{42, 44}”

5. In addition to the cross-sectional image of the $\text{Li}_{0.5}\text{Al}$ electrode after the first charge-discharge cycle, additional cross-sectional analysis of the following additional samples is also required.

- Pure Al foil after charge and discharge

- $\text{Li}_{0.5}\text{Al}$ anode and its interface after 100+ cycles

Response.

We thank the reviewer for the constructive suggestion. It is crucial to fully understand not only the volume change but also the morphology change of various Li-Al. The volume and morphology changes of pure Al, $\text{Li}_{0.25}\text{Al}_1$ and $\text{Li}_{0.75}\text{Al}_1$ anodes after the first charge and discharge have been further observed via cross-sectional characterization using PFIB-SEM, as shown in **Supplementary Fig. 21**. The electrode density was shown to improve with increasing the Li ratio, attributed to enhanced particle pulverization (**Supplementary Fig. 22 and 25**). Furthermore, we investigated the 2000-cycled $\text{Li}_{0.5}\text{Al}_1$ anode to compare its pulverized morphology after long-term cycling. Interestingly, the highly dense structure composed of sub-micrometer-sized particles was visualized in **Supplementary Fig. 23**. The corresponding discussion has been added to the main text as follows:

(Main text, page 12)

“...Therefore, close and intimate contact between the expanded Li-Al alloy particles was observed after charging, (**Middle in Fig. 4f**) which was further enhanced by the particles pulverized to a few micrometers in size during discharge (**Middle in Fig. 4g**). The pulverized particles, which could densify the electrode structure, would influence the improvement in interface and charge transfer resistances during the early cycles in **Fig. 4c**. Additional cross-sectional characterization of the pure Al and other Li-Al alloy anodes also confirmed that the electrode density could be improved by increasing the Li ratio, due to enhanced particle pulverization (**Supplementary Fig. 21**). Correspondingly, the porosity measured in the 1st discharged state was significantly reduced in the Li-Al alloy anodes compared to pure Al (**Supplementary Fig. 22**). Interestingly, even the 2000-cycled $\text{Li}_{0.5}\text{Al}_1$ anode maintained a highly dense structure by the pulverized particles continuing to reduce to sub-micrometer size during repetitive volume changes (**Supplementary Fig. 23**)...”

(Main text, page 12)

“...The charged $\text{Li}_{0.5}\text{Al}_1$ is regarded as a Li^+ conductor with highly developed Li^+ transport channels throughout the electrode. After discharging, the electrode consisted of pulverized particles surrounded by thin α -phase layer (**Right in Fig. 4g**). XRD characterization of the $\text{Li}_{0.5}\text{Al}_1$ anode clearly revealed this phase development in which the α phase significantly decreased after charging and fully recovered after discharging (**Supplementary Fig. 24**). More notably, the β -LiAl was observed to be relatively concentrated on the SE side far from the copper current collector. When Li^+ moves toward the SE during discharging, it could be inferred that the Li concentration gradient, generated at the interface between the Li-absent vacancy and the Li-present β -LiAl lattice, may possibly drive Li^+ diffusion toward the SE layer³⁴. Thus, highly developed β -LiAl phase³⁴ in the Li-Al alloy anodes with a

higher Li ratio could facilitate Li^+ diffusion to the SE side, serving as efficient Li^+ transport channels. The insight could be further supported by observing the cross-sections of discharged pure Al and other Li-Al alloy anodes in BSE mode (Supplementary Fig. 21 and 25). The $\text{Li}_{0.5}\text{Al}_1$ and $\text{Li}_{0.75}\text{Al}_1$ anodes with a higher Li ratio showed that pulverized particles composed of small β phase domains ($\text{Li}_{0.5}\text{Al}_1$: 2.326 μm ; $\text{Li}_{0.75}\text{Al}_1$: 1.607 μm) surrounded by thin α -phase layer, and the dark β -LiAl layer is mostly distributed at the SE-anode interface. Meanwhile, the light gray α -Al layer was clearly observed at the interface of the pure Al and $\text{Li}_{0.25}\text{Al}_1$ anodes with β -phase island (pure Al: 4.877 μm ; $\text{Li}_{0.25}\text{Al}_1$: 2.783 μm) enclosed by thick α -phase layer.”

Supplementary Figure 21. Cross-sections of pure Al and Li-Al alloy anodes in the pristine, 1st charged, and 1st discharged states. a Pure Al. b $\text{Li}_{0.25}\text{Al}_1$. c $\text{Li}_{0.5}\text{Al}_1$. d $\text{Li}_{0.75}\text{Al}_1$. SE and BSE indicate secondary electron

mode and backscattered electron mode, respectively. The area of yellow box was enlarged to high-magnification images at the rightmost row. **e** Schematics of discharged Li-Al alloy anodes.

Supplementary Figure 22. Porosity of pure Al and Li-Al alloy anodes in the 1st discharged state. **a** Pure Al. **b** Li_{0.25}Al₁. **c** Li_{0.5}Al₁. **d** Li_{0.75}Al₁.

Supplementary Figure 23. Cross-sections of discharged Li_{0.5}Al₁ after 2000 cycles. **a** Secondary electron mode. **b** Backscattered electron mode. **c** High-magnification image of the yellow region boxed in **b**.

Supplementary Figure 25. β -phase domains distribution within the 1st discharged state. **a** Pure Al. **b** $\text{Li}_{0.25}\text{Al}_1$. **c** $\text{Li}_{0.5}\text{Al}_1$. **d** $\text{Li}_{0.75}\text{Al}_1$.

6. The following errors in the paper must be addressed, and a thorough review is necessary:

Response.

The authors appreciate the detailed comments for the revision. The following items have been revised accordingly, one by one.

6-1. In the case of Li half-cell, cycling begins with the lithiation of Li-Al alloy. An explanation is needed for how the delithiation capacity is higher than lithiation capacity in Fig. 1d, e.

Response: We apologize for this labeling error in the initial draft. The voltage profile of delithiation was reversed to start from the end of the lithiation. To clarify, the normalized capacity for delithiation has been added at the top with a reversed direction, and arrows indicating the reaction direction have been included as follows:

Fig. 1 | Li^+ diffusion kinetics in Li-Al alloy. **a** Density functional theory (DFT)-based calculation on Li^+ diffusivity in α and β phases. **b and c** Calculated migration energy of Al and Li. **b** α phase. **c** β phase. **d and e** Galvanostatic intermittent titration technique on lithiation and delithiation of pure Al in solid-state Li metal half cell ($\text{Li}|\text{LPSCl}|\text{Al}$). **d** Voltage profiles of galvanostatic pulses and open-circuited state. **e** Voltage polarization at each open-circuited state. Galvanostatic titration at 0.1 mA cm^{-2} for 30 mins; Relaxation at open circuit for 2 hours; Stack pressure at 50 MPa; Room temperature.

6-2. Fig. 4b impedance graphs require clear internal labeling to identify each component, as done in Fig. 5d.

Response: The spectra with the darker color correspond to measurements conducted after the cycling. **Fig. 3b** and **c** have been revised as follows:

Fig. 3 | Electrochemical kinetics behaviors of Li-Al alloy electrodes. a Long-term cyclability in symmetric cell with $\text{Li}_{0.25}\text{Al}_1$, $\text{Li}_{0.5}\text{Al}_1$ and $\text{Li}_{0.75}\text{Al}_1$ anodes at 0.25 mA cm^{-2} to 0.25 mAh cm^{-2} . **b and c** Electrochemical impedance measurements on the pristine and 200 h-cycled Li-Al electrodes. **b** Nyquist plot of impedance. **c** Distribution of relaxation time (DRT) spectra. **d** Rate capability test of NCM811-based full cells with pure Al, $\text{Li}_{0.25}\text{Al}_1$, $\text{Li}_{0.5}\text{Al}_1$ and $\text{Li}_{0.75}\text{Al}_1$ anodes. **e** ICE of pure Al, $\text{Li}_{0.25}\text{Al}_1$, $\text{Li}_{0.5}\text{Al}_1$ and $\text{Li}_{0.75}\text{Al}_1$ anodes. **f** Specific capacities comparison at each current density. **g** Volumetric energy density (Wh L^{-1}) and specific energy (Wh kg^{-1}) of ASSB with the configuration of pure Al and Li-Al alloy-based anodes and NCM811 cathode. Stack pressure for the Li-Al symmetric cell and NCM811||Li-Al full-cell tests: 75 MPa. N/P ratio for full cells: 2.

6-3. The overall readability of the figures, including Fig. 5b is compromised. The visual distinction of the lines should be enhanced by modifying the colors and line thicknesses for better reader comprehension.

6-4. In Fig. 5b, although four datasets are listed in the caption, only three are shown on the graph.

Response: In Fig. 4b, the colors, line thickness, and datasets have been revised based on comments 6-3 and 6-4 as follows:

Fig. 4 | Long-term cycling performance and morphological change of $\text{Li}_{0.5}\text{Al}_1$ anode with high-loading NCM811 cathode. a-d Long-term capacity retention test and voltage profiles at corresponding cycle. a Long-term cyclability at 0.33C ($1\text{C} = 5\text{ mA cm}^{-2}$). b Voltage profiles at 1st, 100th, 200th and 300th cycle. c Long-term cyclability at 0.8C ($1\text{C} = 5\text{ mA cm}^{-2}$). All the data points were selected at every 25 cycles. Stack pressure: 75 MPa. N/P ratio: 2. d EIS analysis at 100th and 500th cycle. e-g PFIB-prepared cross-sections of $\text{Li}_{0.5}\text{Al}_1$ anode. e Pristine. f After 1st charge. g After 1st discharge. The area of yellow box was enlarged to high-magnification images at the middle row. The first-row images in backscattered mode were shown at the rightmost row. h Schematics of $\text{Li}_{0.5}\text{Al}_1$ anode representing morphological change, Li^+ diffusion pathway and phase development and during charging and discharging.

6-5. In Fig. 5c, while the cycle life extends to 2000 cycles, the number of plotted data points is insufficient. Please plot all 2000 data points accordingly.

Response: Fig. 4c displays the data points for every 25 cycles. Although it is possible to include all data points for 2000 cycles, the authors considered that including all 2000 data points would make the graph too complicated, as shown in **Supplementary Figure 18**.

Supplementary Figure 18. Long-term cyclability at 0.8C (1C = 5 mA cm⁻²). All the data points were included. Stack pressure: 75 MPa. N/P ratio: 2.

6-6. In Fig. 5c, the capacity continues to increase for approximately 100 cycles; an explanation for this prolonged initial capacity increase is needed.

Response: The capacity of the high-loading Li_{0.5}Al₁||NCM811 full cell increased due to the enhanced utilization of Li_{0.5}Al₁ anode, facilitated by improved Li⁺ diffusion through reduced diffusion length within pulverized Li-Al particles. Particularly at the beginning of cycling, the utilization of both the anode and cathode was restricted at high current density (1st discharge capacity: 1.65 mAh cm⁻² at 0.8C) compared to low current density (1st discharge capacity: 3.47 mAh cm⁻² at 0.33C) leading to gradual anode activation toward the optimal diffusion status. EIS measurements after 100 cycles confirmed that the R_{anode} remained higher at 0.8C than that at 0.33C (37.3 Ω vs. 30.4 Ω), whereas R_{cathode} showed lower resistance at 0.8C than that at 0.33C (37.3 Ω vs. 30.4 Ω), representing the limited utilization of both anode and cathode at 0.8C (**Fig. R2**).

Fig. R2. **a** EIS measurement on $\text{Li}_{0.5}\text{Al}_1\|\text{NCM811}$ full cell at the 100th cycle of long-term cycling at 0.33C and 0.8C. **b** Equivalent circuit used for fitting EIS spectra in Fig. R2a. **c** Fitting results.

6-7. Please cite and introduce recent research trends in alloy-based anode strategies in the introduction.

doi.org/10.1021/acsenergylett.4c00915, doi.org/10.1002/aenm.202301508,

doi.org/10.1002/adv.202301381, doi.org/10.1021/acsami.2c09013

Response: The authors agree to introduce recent studies on alloy-based anodes for general understanding. The suggested works, including additional cases on pouch cells operating under low stack pressure, have been discussed in the introduction as follows:

(Main text, page 3)

“...For example, a recent work on micron-silicon (μSi) demonstrated a high specific capacity (2890 mAh g^{-1}) with a high Li^+ diffusivity in the lithiated state ($\sim 10^{-9} \text{ cm}^2 \text{ s}^{-1}$) without adding SEs in the anode, leading to outstanding high-capacity performance at room temperature¹⁴. The prelithiation strategy was applied to alloy anodes to facilitate uniform ion diffusion for mitigating chemo-mechanical degradation and suppressing Li dendrite formation, using Li-Si and Li-Ag alloys, respectively.^{18,19} To further achieve specific energy density (Wh kg^{-1}) to the ultimate limit, an anode-less design was developed with a thin Mg film as an effective seeding layer for reversible lithium plating and stripping, supported by a $\text{Ti}_3\text{C}_2\text{T}_x$ MXene mechanical buffer layer underneath.²⁰ For practical applications, however, recent studies at the forefront have been challenging the large volume changes in alloy anodes during Li (de)alloying reactions by improving chemo-mechanical stability within large-scale pouch cells operating under low stack pressure.^{21,22}”

6-8. The cell assembly used in Supplementary Figure 7 should be added to the caption, as indicated in Supplementary Figure 8.

Response: The cell configuration has been added to the caption of **Supplementary Fig. 11**, which was updated from Supplementary Figure 7 during the revision as follows:

Supplementary Figure 11. Rate capability test for the kinetics improvement of Li_xAl_1 alloy electrodes along with prelithiation degree increase in NCM811-based full cells. a Specific capacity comparison among Li_xAl_1 alloy electrodes at each current density. b Pure Al. c $\text{Li}_{0.25}\text{Al}_1$. d $\text{Li}_{0.4}\text{Al}_1$. e $\text{Li}_{0.5}\text{Al}_1$. f $\text{Li}_{0.75}\text{Al}_1$. g $\text{Li}_{0.9}\text{Al}_1$. h $\text{Li}_{0.95}\text{Al}_1$. Stack pressure: 75 MPa. N/P ratio: 2.

Reviewer 5

In this manuscript, the authors present a Li-Al alloy as an anode for sulfide-based all-solid-state batteries. This anode demonstrates a high critical current density, extended cycling life, and stable structural integrity during cycling. While the use of a Li-Al alloy as an anode is not a novel concept (references: *Sci. Adv.* 2022, 8, eabn4372; *J. Mater. Chem. A* 2022, 10, 12350; *Small* 2022, 18, e2204037; *Nat. Commun.* 2023, 14, 3975; *Adv. Mater.* 2024, 2407128), previous studies have also confirmed its compatibility with sulfide solid electrolytes. However, this report introduces distinctive features, particularly in its comprehensive analysis of the correlation between crystal phase and Li⁺ kinetics. After significant revision, this manuscript could be suitable for publication in *Nature Communications*.

Response: The authors appreciate the reviewer's thoughtful and positive comments. We have carefully addressed each point and supplemented additional characterizations to reinforce the novelty of this work and demonstrate the consistency of the anodes under various conditions. Previous works on Li-Al alloys, including those suggested by the reviewer, have been introduced into the main text, and each suggested point has been responded one by one below. Lastly, for your reference, we've reordered the main figures, moving original Fig. 3 to Fig. 5 to make the proposed mechanism clearer. We believe that these revisions have significantly improved the quality of this work.

1. In Fig. 2a, the XRD patterns show that the peak of the Li-Al alloy shifts compared to the standard reference, indicating lattice distortion in the prepared Li-Al alloy, suggesting that Li or Al atoms are integrating into the lattice. However, the authors state that "there is no intermediate Li-Al alloy compound due to the immiscibility of the two phases (Line 89)." The authors should clarify the interpretation of this peak shift.

Response.

The authors appreciate the opportunity to revise the XRD results. It was confirmed that the positive shift occurred due to a Z-axis misalignment between the surface of the XRD characterization area and the thick specimen (**Fig. R1**).¹ Since the Li-Al alloy anodes used in the NCM811 full-cell study were prepared, the electrode surface was positioned higher than that of the XRD specimen holder, resulting in a positive shift even for the thin Li_{0.5}Al₁ case ($\Delta_2 > \Delta_1$ as shown in **Fig. R1b**). Therefore, all XRD peaks were recalibrated based on the strong Al (111) peak, which eventually aligned well with the reference peaks, as shown in the revised **Fig. 2a** below.

Fig. R1. **a** A schematic of XRD specimen preparation with specimen height indicated. **b** XRD patterns of $\text{Li}_{0.5}\text{Al}_1$ anodes with different thicknesses. The XRD patterns of thick $\text{Li}_{0.5}\text{Al}_1$ anode shifted more positively when the characterization surface was positioned higher in the same sample holder.

Fig. 2a. XRD pattern of $\text{Li}_{0.25}\text{Al}_1$, $\text{Li}_{0.5}\text{Al}_1$ and $\text{Li}_{0.75}\text{Al}_1$ anodes.

[1] Kaduk, J. A., *et al.* Powder diffraction. *Nat. Rev. Methods Primers* **1**, 77 (2021).

2. In Fig. 2d, the fitted curve does not align well with the experimental data and should be re-fitted.

Response.

The authors appreciate the reviewers for pointing this out, which has led to improved data quality. The Al 2p XPS spectra have been carefully refitted (**Fig. 2d** and **Table R1**), and the corresponding discussion on the binding energy was revised as follows:

(Main text, page 7)

“The lithiated state of Li-Al alloy anode was estimated by X-ray photoelectron spectroscopy (XPS) depth profiling on Al 2p (**Fig. 2d**). The deconvoluted XPS Al 2p spectra show Al_2O_3 at 75.7 eV, Li_xAlO_y at 74.8 eV, $\text{Al}(\text{OH})_3$ at 73.6 eV, Al at 72.6 eV and LiAl at 71.7 eV.⁴⁵...”

Fig. 2d. Al 2p X-ray photoelectron spectra (XPS) of $\text{Li}_{0.5}\text{Al}_1$ anode according to the depth.

Table R1 Fitting results of the XPS Al 2p spectra in Fig. 2d.

	Binding energy (eV)			FWHM		
	Surface	15s milling	45s milling	Surface	15s milling	45s milling
Al_2O_3	75.67	75.67	75.67	0.83	0.83	0.83
Li_xAlO_y	74.81	74.81	74.81	1.09	1.16	1.24

Al(OH)₃	73.64	73.64	76.64	1.24	1.20	1.17
Al	-	72.60	72.60	-	0.87	0.82
LiAl	71.74	71.74	71.74	0.72	0.71	0.73

3. The Li⁺ diffusion coefficient should be calculated using the GITT method.

Response.

We appreciate the reviewer for this comment. As also pointed out by Reviewer #4 in Q4, the authors would like to kindly clarify that the galvanostatic intermittent titration technique (GITT) is applicable only to materials undergoing single-phase reactions, as the theoretical conditions of GITT is based on Fick's law, considering only diffusion. Therefore, materials with two-phase reaction behavior, such as aluminum, are not ideal for Li diffusivity analysis using GITT, since the additional factor of phase boundary propagation should be considered. Nevertheless, this explanation should be clearly stated in the main text. The limitation of GITT has been added to the main text, as follows:

(Main text, page 5)

Li diffusion kinetics in Li-Al alloy

“...This diffusion-dependent kinetics was also confirmed through the delithiation where the voltage polarization decreased back to 68% of Li after the vacancy formation, followed by rapid increase due to the development of α -dominant phase at the Al surface. It should be noted that GITT provides Li diffusivity at each lithiated state of a single-phase reaction material based on Fick's law. In other words, a two-phase reaction material, such as Al, is not an ideal model for Li diffusivity calculations via GITT, which was therefore not applied in this study.^{42, 44}”

4. In Figs. 3b and 3c, after the dealloying of the alloyed Al/Li-Al (step ii), it is unclear why the α -phase Al core transitions to a β -phase LiAl core. The authors should provide additional characterization data to substantiate this observation.

Response.

We thank the reviewer for insightful comment. The potential difference resulted from the concentration gradient between the outer β -phase LiAl and the inner α -phase Al drives Li interdiffusion, leading to Li trapping into the Al core.^{1,2} To visualize the phase development in the pure Al and $\text{Li}_{0.5}\text{Al}_1$ after charge (lithiation) and discharge (delithiation), cross-sectional characterization was conducted using FIB-SEM in BSE mode (**Fig. R2**). The $\text{Li}_{0.5}\text{Al}_1$ anode showed uniform distribution of β phase (dark gray) in the charged state, which was subsequently surrounded by thin α -phase layer (light gray) after discharge (**Fig. R2b**). In contrast, the charged Al retained α -phase islands within the β phase, which, in this case, was subsequently enclosed by thick α phase layer after discharge (**Fig. R2a**). Additionally, to further support the proposed mechanism and study phase development across different Li ratios, the cross-sections of pure Al, $\text{Li}_{0.25}\text{Al}_1$, $\text{Li}_{0.5}\text{Al}_1$ and $\text{Li}_{0.75}\text{Al}_1$ anodes in pristine, charged and discharged states have been added as **Supplementary Fig. 21**.

[1] Rehnlund, D., *et al.* Lithium trapping in alloy forming electrodes and current collectors for lithium based batteries. *Energy & Environ. Sci.* **10**, 1350-1357 (2017).

[2] Li, H., *et al.* Circumventing huge volume strain in alloy anodes of lithium batteries. *Nat. Commun.* **11**, 1584 (2020).

Fig. R2. Cross-sections of pure Al and $\text{Li}_{0.5}\text{Al}_1$ alloy anodes in the pristine, 1st charged, and 1st discharged states. a Pure Al. b $\text{Li}_{0.5}\text{Al}_1$. SE and BSE indicate secondary electron mode and backscattered electron mode, respectively. The area within the yellow box was enlarged for high-magnification images in BSE mode. c Schematics of discharged pure Al and $\text{Li}_{0.5}\text{Al}_1$ alloy anodes.

Supplementary Figure 21. Cross-sections of pure Al and Li-Al alloy anodes in the pristine, 1st charged, and 1st discharged states. **a** Pure Al. **b** $\text{Li}_{0.25}\text{Al}_1$. **c** $\text{Li}_{0.5}\text{Al}_1$. **c** $\text{Li}_{0.5}\text{Al}_1$. **d** $\text{Li}_{0.75}\text{Al}_1$. SE and BSE indicate secondary electron mode and backscattered electron mode, respectively. The area of yellow box was enlarged to high-magnification images at the rightmost row. **e** Schematics of discharged Li-Al alloy anodes.

5. The current density and areal capacity used in the cycling tests of symmetric cells are too low to adequately demonstrate the high-rate performance of Li-Al alloy anodes.
 6. The authors should consider conducting a critical current density test in symmetric cells to determine the maximum current density the anode can tolerate. Evaluating this in a full battery would also be influenced by the dynamics of the positive electrode.
-

Response.

The reviewer raised an important concern. Based on the comments 5 and 6, the critical current density (CCD) test and cycling test with current density of 1 mA cm^{-2} and areal capacity of 0.5 mAh cm^{-2} were evaluated using a symmetric cell configuration. In the CCD tests, $\text{Li}_{0.75}\text{Al}_1$ showed stable operation up to a current density of 12 mA cm^{-2} , while $\text{Li}_{0.25}\text{Al}_1$ and $\text{Li}_{0.5}\text{Al}_1$ failed at 0.1 mA cm^{-2} and 1.6 mA cm^{-2} , respectively. Additionally, in the long-term cyclability tests, only $\text{Li}_{0.75}\text{Al}_1$ exhibited stable performance. This failure was due to insufficient $\beta\text{-LiAl}$ to provide Li to the counterpart which is a critical limiting factor in symmetric cell configuration. Therefore, for evaluating the Li-Al alloy anodes, the authors would like to highlight the full-cell rate performance with an areal capacity of 2.5 mAh cm^{-2} , as shown in **Fig. 3d-f**, where the cell performance was limited by anode rate capability (Pure Al: 4 mA cm^{-2} , $\text{Li}_{0.25}\text{Al}_1$: 4 mA cm^{-2} , $\text{Li}_{0.5}\text{Al}_1$: 7 mA cm^{-2} and $\text{Li}_{0.75}\text{Al}_1$: 7 mA cm^{-2}). The corresponding discussion on CCD and long-term cycling tests has been added to the main text as follows:

(Main text, page 9)

Electrochemical kinetics of Al-based electrodes

“The Li^+ diffusion kinetics was investigated in symmetric cells employing Li-Al alloy electrodes with different Li to Al ratios of 0.25, 0.5, and 0.75 (**Fig. 3a-c**). Overpotentials were compared during delithiation and lithiation at 0.25 mA cm^{-2} to 0.25 mAh cm^{-2} (**Fig. 3a**). While the voltage of $\text{Li}_{0.25}\text{Al}_1$ already spiked dramatically prior to the first lithiation, Li-Al alloy electrodes with higher Li ratio experienced less overpotential (0.26 V for $\text{Li}_{0.5}\text{Al}_1$ and 0.23 V for $\text{Li}_{0.75}\text{Al}_1$). In the critical current density (CCD) tests, the Li-Al alloy electrodes exhibited consistent kinetic behavior, with $\text{Li}_{0.25}\text{Al}_1$ and $\text{Li}_{0.5}\text{Al}_1$ failing at a current density of 0.1 mA cm^{-2} and 1.6 mA cm^{-2} , respectively, whereas $\text{Li}_{0.75}\text{Al}_1$ can be operated up to 12 mA cm^{-2} (**Supplementary Fig. 5**). On the other hand, the Li symmetric cell short-circuited at 0.7 mA cm^{-2} resulting from Li dendrite growth, which was not observed in the Li-Al alloy cells.⁴⁶ ... Therefore, $\text{Li}_{0.75}\text{Al}_1$ demonstrated stable long-term cycling performance compared to $\text{Li}_{0.5}\text{Al}_1$ which failed after a few cycles with large overpotential during lithiation and delithiation at 1 mA cm^{-2} up to 0.5 mAh cm^{-2} (**Supplementary Fig. 7**)...”

Supplementary Figure 5. Critical-current-density test of symmetric cells with Li, $\text{Li}_{0.25}\text{Al}_1$, $\text{Li}_{0.5}\text{Al}_1$ and $\text{Li}_{0.75}\text{Al}_1$ electrodes. Lithiation and delithiation were conducted up to 0.25 mAh cm^{-2} at each current density. Stack pressure: 10 MPa for Li and 75 MPa for Li-Al alloy electrodes.

Supplementary Figure 7. Long-term cyclability of symmetric cells with $\text{Li}_{0.5}\text{Al}_1$ and $\text{Li}_{0.75}\text{Al}_1$ anodes. The inset shows magnified voltage profiles of the $\text{Li}_{0.5}\text{Al}_1$ symmetric cell during the first 5 hours. Lithiation and delithiation were conducted at 1 mA cm^{-2} up to 0.5 mAh cm^{-2} . Stack pressure: 75 MPa.

7. Regarding the NP ratio, there are several concerns: Is NP calculated based on the theoretical specific capacity of the Li-Al alloy? The theoretical specific capacity of the Li-Al alloy in Fig. S11 is based on Li_1Al_1 , but Li_1Al_1 can be further lithiated to Li_3Al_2 and Li_9Al_4 . The theoretical NP ratio may not be appropriate due to the different rate capabilities of the anode and cathode; that is, at the same current density, the delivered capacity/theoretical capacity ratio of the anode and cathode differs. Additionally, in Line 260, the statement “Decreasing the N/P ratio reduces the capacity of Li-poor α -phase Al” needs further clarification.

Response.

Yes. The N/P ratio was calculated based on the theoretical capacity of the Li_xAl_1 alloy (700, 440 and 208 mAh/g for $\text{Li}_{0.25}\text{Al}_1$, $\text{Li}_{0.5}\text{Al}_1$ and $\text{Li}_{0.75}\text{Al}_1$, respectively), since Al can be stably lithiated up to 1 mol of Li at room temperature without undergoing further lithiation,¹ as confirmed by XRD analysis (**Fig. 2a**). In addition, to clarify the statement, the relationship between the N/P ratio and phase utilization during charge and discharge was illustrated using a $\text{Li}_{0.25}\text{Al}_1\|\text{NCM811}$ full-cell configuration (**Fig. R3**). In a cell design with a fixed cathode capacity at an N/P ratio of 1.2, more of the β phase developed while the proportion of the α phase decreased compared to an N/P ratio of 2. Consequently, as discussed in **Supplementary Fig. 14** and **15**, initially α -phase-dominant $\text{Li}_{0.25}\text{Al}_1$ and pure Al exhibited significant improvements at the N/P ratio 1.2, where the transition to a β -phase-dominant state provided enhanced Li^+ conductive channels, improving reversibility.

Fig. 2a. XRD pattern of $\text{Li}_{0.25}\text{Al}_1$, $\text{Li}_{0.5}\text{Al}_1$ and $\text{Li}_{0.75}\text{Al}_1$ anodes.

Fig. R3. Schematics of phase utilization within $\text{Li}_{0.25}\text{Al}_1$ anode based at different N/P ratios. **a** N/P ratio = 2 and **b** N/P ratio = 1.2.

[1] Zheng, T., Zhang, J., Jin, W., Boles, S. T. Utilization of Li-rich phases in aluminum anodes for improved cycling performance through strategic thermal control. *ACS Appl. Energy Mater.* **6**, 1845-1852 (2023).

8. The cutoff voltage of the full cells is questionable, as the polarization of the charging curves sharply increases above 4.0 V. In previous Li-Al alloy-based studies, the cutoff voltage for full cells does not exceed 4.0 V.

Response.

The upper cutoff voltage of 4.25 V for the Li-Al||NCM811 cell was determined to fully obtain capacity from the NCM cathode. Additionally, previous studies on alloy anodes, including Li-Al alloy anode, have employed NCM cathodes for full-cell tests with upper cutoff voltages higher than 4.0 V, some of which are introduced below (Cutoff voltage of Ref. 1: 4.2 V; Ref. 2: 4.1 V; Ref. 3: 4.1 V; Ref. 4: 4.3 V).

[1] Zhang, L., *et al.* Electrochemical grain refinement enables high-performance lithium–aluminum-anode-based all-solid-state batteries. *ACS Energy Letters* **10**, 898-906 (2025).

[2] Jeong, W. J., Wang, C., Yoon, S. G., Liu, Y., Chen, T., McDowell, M. T. Electrochemical behavior of elemental alloy anodes in solid-state batteries. *ACS Energy Letters*, 2554-2563 (2024).

[3] Fan, Z., *et al.* Long-cycling all-solid-state batteries achieved by 2D interface between prelithiated aluminum foil anode and sulfide electrolyte. *Small* **18**, 2204037 (2022).

[4] Ham, S. Y., *et al.* Overcoming low initial coulombic efficiencies of Si anodes through prelithiation in all-solid-state batteries. *Nat Commun* **15**, 2991 (2024).

9. In Fig. S13, the intermediate frequency region of the EIS spectra is attributed to the resistance of the cathode interface, but the authors have designated it as the anode interface resistance. This should be corrected.

Response.

We appreciate the reviewer for the comment. The fitted $R_{\text{interface}}$ of the cycled $\text{Li}_{0.5}\text{Al}_1\|\text{NCM811}$ full cell contains information from both the anode and cathode.¹ EIS and DRT spectra of $\text{Li}_{0.5}\text{Al}_1\|\text{NCM811}$ were investigated to compare the relaxation time (τ) region for $R_{\text{interface}}$ with that of the $\text{Li}_{0.5}\text{Al}_1\|\text{Li}_{0.5}\text{Al}_1$ symmetric cell (**Fig. R4**). After the first charge and discharge of $\text{Li}_{0.5}\text{Al}_1\|\text{NCM811}$, the second DRT peaks within the τ region from 3×10^{-6} s to 3×10^{-5} s remained stable, coinciding with the interface region of $\text{Li}_{0.5}\text{Al}_1$ (**Fig. R4b**). Meanwhile, given the low resistances of $\text{Li}_{0.5}\text{Al}_1$ across the τ range, the intermediate region from $\sim 10^{-4}$ s exhibited significant peak change depending on the lithiated state of NCM811.

Fig. R4. **a** EIS measurement of 200 h-cycled $\text{Li}_{0.5}\text{Al}_1\|\text{Li}_{0.5}\text{Al}_1$ cell and $\text{Li}_{0.5}\text{Al}_1\|\text{NCM811}$ full cell after 1st charge and discharge at 0.1C. The equivalent circuit used for fitting is shown as insets. **b** DRT spectra converted from Fig. R4a. **c** Fitting results.

[1] Chen, Y.-T., *et al.* Enabling uniform and accurate control of cycling pressure for all-solid-state batteries. *Advanced Energy Materials* **14**, 2304327 (2024).

10. How was the Li-Al electrode prepared? Was it prepared similarly to the Al electrode using the casting method? If not, why?

Response.

All Li-Al electrodes were prepared by mixing Li and Al powder and spreading the Li-Al powder composite onto the LPSCl separator layer during cell assembly, as described in the experimental section. The pure Al electrode, however, was prepared by slurry casting, as it was difficult to control the Al loading density of 2.5 mg cm^{-2} using powder spreading.

(Main text, page 17)

Electrodes preparation

“*Pure Al electrode.* The pure Al electrode slurry was prepared by mixing Al with 0.1 wt. % of SEBS in xylene solvent for 15 minutes. The slurry was cast on the non-polished side of Cu foil by a doctor blade and dried at $80 \text{ }^\circ\text{C}$ for 2 hours. The Al loading density was 2.5 mg cm^{-2} equivalent to 2.5 mAh cm^{-2} .”

Li-Al powder composite. The Li and Al powder was mixed to prepare Li-Al composite at the atomic ratio of x to 1 in Li_xAl_1 ($x = 0.25, 0.4, 0.5, 0.75, 0.9$ and 0.95) for 10 min at 1900 rpm using a vortex mixer (MI0101002D, Four E’s Scientific).”

(Main text, page 17)

Solid-state cell assembly

“The solid-state cells were assembled with two titanium plungers and one polyetheretherketone (PEEK) holder with diameter of 10 mm in the glove box.

Half cell. Li metal half cells with pure Al and Li-Al alloy anodes were used for GITT and *in situ* EIS analyses. For a pure Al half cell, A pure Al electrode disc (radius = 5 mm) was prepared as the working electrode, and a Li metal foil (radius = 5 mm) was used as the counter electrode. 70 mg of LPSCl was spread on a titanium plunger in a PEEK holder and pressed with the other plunger at 310 MPa for 10 s to first prepare a separator layer. The pure Al electrode disc was then placed on one side of the separator layer and pressed at 375 MPa for 3 min, followed by insertion of the Li metal foil on the opposite side. For half cells with Li-Al alloy anodes, 70 mg of LPSCl was spread and pressed between two titanium plungers at 375 MPa for 3 min. The Li-Al powder composite for Li_xAl_1 anode ($x = 0.25, 0.5$ and 0.75) was spread on one side of the separator layer and pressed at 125 MPa

for 10 s, followed by insertion of the Li metal foil on the opposite side. The assembled half cells were rested under a stack pressure of 55 MPa for 6 h before testing.”

Symmetric cell. 70 mg of LPSCl was spread and pressed between two titanium plungers at 375 MPa for 3 min. The Li-Al powder composite for Li_xAl_1 anode ($x = 0.25, 0.5$ and 0.75) was spread on both sides of the separator layer and pressed at 125 MPa for 10 s. The assembled cells were rested under a stack pressure of 75 MPa for 6 h before testing.

Full cell. A SE layer was first prepared by pressing 70 mg of LPSCl at 310 MPa for 10 s. NCM powder composite or NCM dry film was placed on one side of the layer and pressed at 375 MPa for 3 min, followed by Li_xAl_1 ($x = 0.25, 0.4, 0.5, 0.75, 0.9$ and 0.95) anode preparation on the opposite side by pressing at 125 MPa for 10 s,…”

11. Were the cells tested under 75 MPa? This pressure is significantly higher than what has been reported in the literature. The authors should justify this choice.

Response.

A stack pressure of 75 MPa was applied to the symmetric and full-cells to solely investigate the β -phase-dependent Li diffusion in the Li-Al alloy anodes by maintaining intact particle contacts, particularly in the cathode as well as in Li-Al alloy anode.⁶ Additionally, 75 MPa is considered to be within the range of stack pressures employed in ‘high-loading’ and ‘high-rate’ full-cell tests based on the previous works (**Table R2**). Corresponding discussion for the use of 75 MPa has been noted to the main text as follows:

	This work	Ref. 1	Ref. 2	Ref. 3	Ref. 4	Ref. 5	Ref. 6	Ref. 7
Cell configuration	Li _{0.5} Al ₁ NCM811	Li-Al S	MP-Al-H LCO	Al-In NCM	Li ₁ Al ₁ NCM811	Al NCM622	Li _{0.4} Al ₁ SCNCM	In-coated Al SCNCM622
Cathode loading (mAh cm ⁻²)	5	10	3.24	8.3	2.55	5	4.60	5
Test current density (mA cm ⁻²)	4	2	0.81 or 1.62	6.5	2.55	0.50	4.60	1 (Ch)/0.2 (dCh)
Temperature (°C)	25	25 or 60	25 or 50	25	25	25	25	25
Stack pressure (MPa)	75	70	100	50	15	8	100	2 or 5

- [1] Huang, Y., Shao, B., Han, F. Li alloy anodes for high-rate and high-areal-capacity solid-state batteries. *J. Mater. Chem. A* **10**, 12350-12358 (2022).
- [2] Fan, Z., *et al.* Long-Cycling All-solid-state batteries achieved by 2D interface between prelithiated aluminum foil anode and sulfide electrolyte. *Small* **18**, e2204037 (2022).
- [2] Liu Y, *et al.* Aluminum foil negative electrodes with multiphase microstructure for all-solid-state Li-ion batteries. *Nat. Commun.* **14**, 3975 (2023).
- [4] Zhu, J., *et al.* A Porous Li–Al alloy anode toward high-performance sulfide-based all-solid-state lithium batteries. *Adv. Mater.* **36**, 2407128 (2024).
- [5] Jeong, W.J., Wang, C., Yoon, S.G., Liu, Y., Chen, T., McDowell, M.T. Electrochemical behavior of elemental alloy anodes in solid-state batteries. *ACS Energy Lett.* **9**, 2554-2563 (2024).
- [6] Zhang, L., *et al.* Electrochemical grain refinement enables high-performance lithium–aluminum-anode-

based all-solid-state batteries. *ACS Energy Lett.* **10**, 898-906 (2025).

- [7] Wang, C., *et al.* The influence of pressure on lithium dealloying in solid-state and liquid electrolyte batteries. *Nat. Mater.* **24**, 907-916 (2025).

(Main text, page 10)

Electrochemical kinetics of Al-based electrodes

“...At higher current densities, a relatively higher proportion of Li ($x \geq 0.5$) in the Li-Al alloy anode led to higher specific capacity through further improved Li⁺ diffusion (Fig. 3f). It is noted that the stack pressure of 75 MPa was applied in both symmetric and full-cell tests to maintain intact particle contacts, particularly in the cathode composite as well as the Li-Al alloy anode, to solely investigate the β -phase-dependent Li diffusion mechanism.³⁵ Exploring advanced interface strategies will be key to enabling practical large-scale solid-state batteries with Li-Al alloy anodes operating under low stack pressure.”

12. Other minor errors include:

Line 20: "Lithium" should be lowercase as "lithium".

Line 145: "Supplementary Fig. 4" should be corrected to "Supplementary Fig. 5".

The preparation process of the LCO cathode should be provided.

The voltage profiles of the high-loading full cell under different rates and cycles should also be included.

Response.

We appreciate the reviewer for correcting our mistakes. The errors have been revised as follows:

12-1. Line 20: "Lithium" should be lowercase as "lithium".

(Main text, page 2)

Abstract

“Metal alloy anodes are promising candidates for lithium (Li) all-solid-state batteries due to their high specific capacity and low cost...”

12-2. Line 145: "Supplementary Fig. 4" should be corrected to "Supplementary Fig. 5".

Due to the reordering of figures during the revision process, the original **Supplementary Fig. 5** has been changed to **Supplementary Fig. 4** as follows:

(Main text, page 10)

Electrochemical kinetics of Al-based electrodes

“...The enhancement of the charge transfer before the reaction is largely related to the electronically conductive environment, which is consistent with the previous results showing that $\text{Li}_{0.5}\text{Al}_1$ and $\text{Li}_{0.75}\text{Al}_1$ provide improved electronic conductivity than $\text{Li}_{0.25}\text{Al}_1$ (**Supplementary Fig. 4**)...”

12-3. The preparation process of the LCO cathode should be provided.

(Main text, page 17)

Electrodes preparation

“Cathode powder composite. The NCM cathode powder composite was prepared by hand mixing NCM811, ball-milled LPSCI and VGCF in mortar and pestle at 66 to 31 to 3 in mass ratio. NCM811 and ball-milled LPSCI powder were first mixed for 20 mins, and VGCF was added to the composite followed by 10 min mixing. **The LCO cathode powder composite was prepared using the same process and composition as NCM811.”**

12-4. The voltage profiles of the high-loading full cell under different rates and cycles should also be included.

The high-loading full-cell voltage profiles for the rate capability (**Supplementary Fig. 17**) and the long-term cyclability at 0.8C (Fig. 5c) have been added to **Supplementary Fig. 17** and **19**, respectively.

Supplementary Figure 17. Rate capability of Li_{0.5}Al₁ and pure Al with high-loading NCM811 cathode at room temperature (1C = 5 mA cm⁻²). a Areal capacities at various C-rates. b and c Voltage profiles at each C-rate. b Li_{0.5}Al₁. c Pure Al. Stack pressure: 75 MPa. N/P ratio: 2.

Supplementary Figure 19. Voltage profiles at 100th, 500th, 1000th and 2000th cycle of long-term cyclability at 0.33C in Fig. 5c. Stack pressure: 75 MPa. N/P ratio: 2.